# FcγR engagement reprograms neutrophils into antigen cross-presenting cells that elicit acquired anti-tumor immunity

Vijayashree Mysore[1], Xavier Cullere[1], Joseph Mears[2,3,4], Florencia Rosetti[5], Koshu Okubo[1,10], Pei X. Liew [1,10], Fan Zhang [2,3,4], Iris Madera-Salcedo [5], Frank Rosenbauer [6], Richard M. Stone[7], Jon C. Aster[1], Ulrich H. von Andrian [8], Andrew H. Lichtman[1], Soumya Raychaudhuri[2,3,4,9] & Tanya N. Mayadas [1✉]

Classical dendritic cells (cDC) are professional antigen-presenting cells (APC) that regulate immunity and tolerance. Neutrophil-derived cells with properties of DCs (nAPC) are observed in human diseases and after culture of neutrophils with cytokines. Here we show that FcγR-mediated endocytosis of antibody-antigen complexes or an anti-FcγRIIIB-antigen conjugate converts neutrophils into nAPCs that, in contrast to those generated with cytokines alone, activate T cells to levels observed with cDCs and elicit CD8+ T cell-dependent anti-tumor immunity in mice. Single cell transcript analyses and validation studies implicate the transcription factor PU.1 in neutrophil to nAPC conversion. In humans, blood nAPC frequency in lupus patients correlates with disease. Moreover, anti-FcγRIIIB-antigen conjugate treatment induces nAPCs that can activate autologous T cells when using neutrophils from individuals with myeloid neoplasms that harbor neoantigens or those vaccinated against bacterial toxins. Thus, anti-FcγRIIIB-antigen conjugate-induced conversion of neutrophils to immunogenic nAPCs may represent a possible immunotherapy for cancer and infectious diseases.

[1] Department of Pathology, Brigham and Women's Hospital and Harvard Medical School, Boston, MA, USA. [2] Center for Data Sciences, Brigham and Women's Hospital, Boston, MA, USA. [3] Division of Genetics, Department of Medicine, Brigham and Women's Hospital and Harvard Medical School, Boston, MA, USA. [4] Division of Rheumatology, Immunology, Allergy, Brigham and Women's Hospital and Harvard Medical School, Boston, MA, USA. [5] Departamento de Inmunología y Reumatología, Instituto Nacional de Ciencias Médicas y Nutrición Salvador Zubirán, Mexico City, Mexico. [6] Institute of Molecular Tumor Biology, University of Muenster, Muenster, Germany. [7] Medical Oncology, Dana-Farber Cancer Institute and Harvard Medical School, Boston, MA, USA. [8] Department of Microbiology and Immunobiology, Harvard Medical School, Boston, MA, USA. [9] Arthritis Research UK Centre for Genetics and Genomics, Centre for Musculoskeletal Research, The University of Manchester, Manchester, UK. [10] These authors contributed equally: Koshu Okubo, Pei X. Liew. ✉email: tmayadas@rics.bwh.harvard.edu

Classical DCs (cDCs) are highly effective at presenting peptides derived from internalized extracellular antigens on MHC class II molecules (MHCII) to CD4[+] T cells and on MHC class I molecules (MHCI) to CD8[+] T cells. The latter process referred to as cross-presentation, is essential for anti-tumor immunity and eradication of pathogens[1]. Among antigen-presenting cells (APC), only cDCs are specialized for cross-presentation[2] and robust activation of immunologically naive CD4[+] and CD8[+] T cells[3], which have a higher activation threshold than memory T cells[4]. However, cDC-based cancer therapy is difficult in practice due to the low abundance of DCs that cross-present antigens[5–9], and the need to administer short-lived, toxic TLR agonists[10,11] to convert cDC populations into immunogenic APCs.

Neutrophils are widely considered terminally differentiated, short-lived innate immune effector cells[12]. However, neutrophils with DC markers are present in patients with cancer[13], infections[14–16], and autoimmune diseases[17–20], and several studies show that neutrophils upregulate the cDC surface molecules CD11c, MHCII, and T cell co-stimulatory molecules when cultured with cytokines (e.g., GM-CSF) or autologous T cells, while retaining select neutrophil markers and functions[21,22]. These cells were described as transcriptionally similar to monocyte-derived DCs[23] and their generation required the downregulation of the transcription factor Ikaros[13]. Although in vitro studies demonstrate that cytokine-generated neutrophil-derived APCs (nAPC) can promote CD4[+] T cell responses[15,24–26] it is debated whether they efficiently cross-present antigen or if mature, differentiated neutrophils can acquire DC-like features[21,27]. Furthermore, the molecular drivers of conversion are poorly understood and whether nAPCs are pathogenic or can be harnessed for immunotherapy remain largely unexplored.

IgG antibody–antigen immune complexes (ICs) trigger several neutrophil effector functions by binding to activating Fcγ receptors (FcγRs). Mouse neutrophils express activating FcγRs, FcγRI, FcγRIII, and FcγRIV, each associated with a Fcγ-chain that is required for their expression and function. Notably, Fcγ-chain-deficient ($\gamma^{-/-}$) mice are protected from IgG-mediated tissue injury[28,29]. Human neutrophils, by contrast, express FcγRI, FcγRIIA (which promotes cytotoxic functions), and FcγRIIIB, a glycosylphosphatidylinositol (GPI)-linked neutrophil-specific receptor of uncertain function[28–30]. We have explored the in vivo functions of human FcγRs[31] by expressing human FcγRIIA and/or FcγRIIIB selectively on neutrophils of mice that lack the Fcγ-chain (FcγR/$\gamma^{-/-}$) and therefore their endogenous activating FcγRs. Our studies revealed that FcγRIIA and FcγRIIIB both promote IgG-mediated neutrophil accumulation in tissues[31], while only FcγRIIA induces tissue injury[31,32].

Here, we investigate whether neutrophils can convert to cells with the antigen-presenting capabilities of cDCs following engagement of their FcγRs and explore the therapeutic potential of this conversion. Our data demonstrate that neutrophil endocytosis of antibody–antigen complexes via FcγRs or FcγRIIIB engagement with an anti-FcγRIIIB-antigen conjugate rapidly converts them into fully immunogenic nAPCs, and the number of nAPCs in lupus patient blood correlates with disease outcomes. We also define the transcriptome of nAPCs and the transcriptional program driving neutrophil to APC conversion downstream of GM-CSF and/or FcγRs. Of therapeutic relevance, we show that nAPCs induce anti-tumor immunity in mice and when generated from human neutrophils can re-activate autologous antigen-specific memory T cells in vitro. Thus, neutrophil FcγRs provide a direct link between innate and adaptive immunity and may be targeted as a strategy to generate a large number of immunogenic nAPCs for T cell-based immunotherapy.

## Results

**Immune complexes engage activating FcγRs on mature neutrophils to promote their conversion to nAPCs.** Murine peripheral blood mature neutrophils (Supplementary Fig. 1a) were treated with Ovalbumin (Ova)-anti-Ova immune complexes (Ova-IC), anti-Ova, or Ova for 2 h, washed and cultured with GM-CSF, known to preserve viability[33]. At day 3, Ly6G[+] neutrophils under all three conditions exhibited >65% survival and acquired DC surface markers (CD11c[+]MHCII[+]), as previously reported for neutrophils cultured in GM-CSF alone[23]. Importantly, Ova-IC treatment significantly increased the percentage of neutrophils expressing CD11c, MHCII, T cell co-stimulatory molecules (CD80 and CD86), and CCR7 (which promotes DC migration to secondary lymphoid organs)[34] as compared to the other treatments (Fig. 1a, b). We obtained similar results with bone marrow neutrophils (BMNs) from G-CSF (Supplementary Fig. 1b–d) and non-G-CSF (Supplementary Fig. 1e) treated mice and with ICs made with antibody to bovine serum albumin (BSA) or hapten-conjugated-Ova (Supplementary Fig. 1f); levels of DC markers on converted BMNs were equivalent to those of splenic and monocyte-derived cDCs (Supplementary Fig. 1g) while monocyte and macrophage markers were minimal (Supplementary Fig. 1h). Neutrophils treated with Ova-ICs may release cytokines that contribute to nAPC conversion, independent of FcγR engagement with ICs. To test this, neutrophils isolated from mice expressing CD45.1 or CD45.2 isoforms were differentially pre-treated and co-cultured with GM-CSF. We found that CD45.1[+] Ova-IC-pretreated neutrophils were unable to upregulate nAPC markers in CD45.2[+] Ova or anti-Ova pre-treated neutrophils (Fig. 1c). Thus, ICs enhance the conversion of neutrophils to nAPCs in a cell-autonomous manner.

Systemic Lupus Erythematosus is a prototypical IC-mediated disease associated with circulating ICs containing autoantibodies against double-stranded DNA and RNA-protein complexes[35]. We observed that sera from SLE patients, as well as ICs formed in vitro with IgG isolated from SLE patients and Ribonucleoprotein, RNP (SLE-IC), promoted conversion of neutrophils to nAPCs. (Fig. 1d); neutrophil survival in samples treated with human sera or ICs was high despite the absence of GM-CSF, possibly reflecting the ability of human IgG to promote neutrophil survival[36]. SLE-IC did not promote nAPC conversion of $\gamma^{-/-}$ neutrophils lacking activating FcγRs, whereas neutrophils from FcγR humanized mice expressing either human FcγRIIA or FcγRIIIB[31] yielded nAPCs in numbers comparable to wild-type neutrophils (Fig. 1e). TLR ligands such as dsDNA and RNP are present in SLE patient sera[37]. However, MyD88[−/−]TRIF[−/−] neutrophils, which lack all TLR signaling[38] converted to nAPCs following SLE-IC treatment (Supplementary Fig. 1i). Thus, SLE-IC engage activating FcγRs to promote neutrophil to nAPC conversion in the absence of GM-CSF and TLR signaling and, notably, engagement of either human FcγRIIA or FcγRIIIB triggers this conversion.

**Immune complex induced conversion of neutrophils to nAPCs is rapid and is accompanied by polymorpho- to mononuclear changes.** To assess the kinetics of neutrophil to nAPC conversion after IC treatment and potential changes in nuclear morphology that accompany this change, we imaged mature peripheral blood neutrophils obtained from mice with YFP "knocked-in" to the CD11c locus (CD11c-YFP), labeled with a nuclear dye and treated with SLE-IC (in the absence of GM-CSF). At day 2 of SLE-IC treatment, CD11c-YFP negative neutrophils acquired YFP signal, became mononuclear-like with decondensed nuclear chromatin, as observed by confocal imaging of several fields (Fig. 1f, Supplementary Fig. 2, Movie S1–S2). While in vitro treatment of band neutrophils (immature, immediate precursors of end-stage

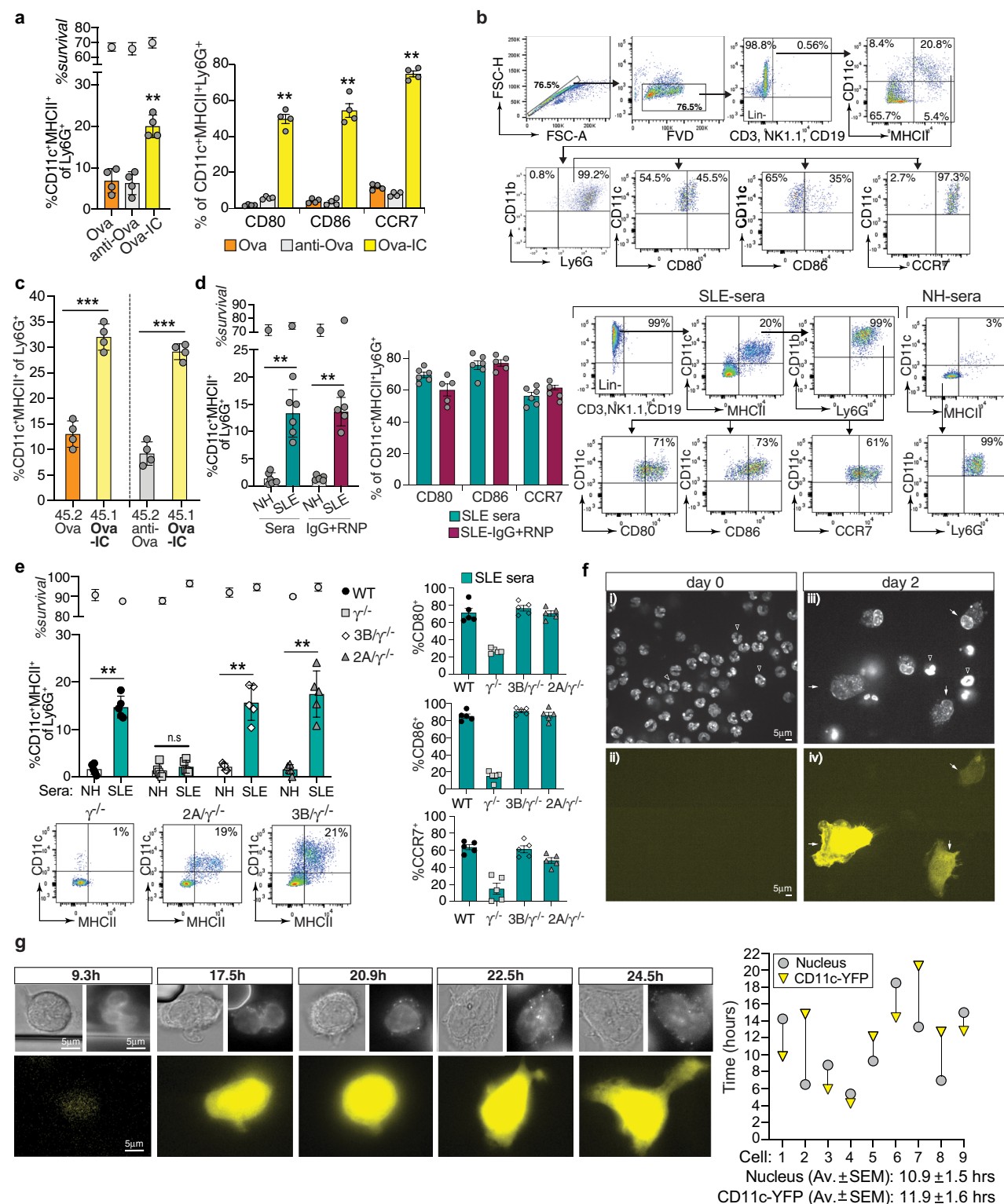

neutrophils[39]) with GM-CSF for 6–9 days has been shown to results in nuclear morphology changes[23,40], we demonstrate the capacity of fully segmented, mature blood neutrophils to undergo changes in nuclear shape. Using live cell imaging, we tracked the conversion of individual neutrophils to nAPCs to examine the kinetics of conversion and thus exclude the possibility that the observed CD11c[+] cells are derived from contaminating mononuclear cells. Within 11 h of SLE-IC stimulation, segmented blood neutrophils acquired CD11c-YFP and a mononuclear appearance

with decondensed chromatin (Fig. 1g, Supplementary Fig. 3a, b, Movie S3). In summary, SLE-IC treatment rapidly converts fully differentiated blood neutrophils into nAPCs that exhibit marked alterations in nuclear morphology.

**nAPC frequency in blood of patients with systemic lupus erythematosus correlate with clinical scores.** Given the ability of SLE-IC to promote neutrophil to nAPC conversion, we looked for

**Fig. 1 Engagement of FcγRs on mouse neutrophils generates nAPCs. a, b** Wild-type (WT) blood neutrophils treated with Ova, anti-Ova, or Ova-IC, cultured with GM-CSF and evaluated 3 days later by flow cytometry for survival and acquisition of CD11c and MHCII on Ly6G+ cells (left panel), and CD80, CD86, and CCR7 on Ly6G+CD11c+MHCII+ cells (right panel) (**a**). Representative gating strategy for Ova-IC generated nAPCs (**b**). **c** Neutrophils from CD45.1 (45.1) or CD45.2 (45.2) mice pre-treated with Ova or Ova-IC, or anti-Ova or Ova-IC, co-cultured with GM-CSF and analyzed for percent of Ly6G+ cells expressing CD11c and MHCII. **d** WT bone marrow neutrophils (BMN) pre-treated with SLE patient or normal human (NH) sera, or SLE-IgG+ RNP (SLE-ICs) or normal human sera-IgG+RNP, cultured without GM-CSF and analyzed as in (**a**). Representative FACS plots are shown. RNP alone resulted in 0.38 ± 0.11% conversion. **e** BMNs from WT, γ−/−, FcγRIIIB(3B)/γ−/− or FcγRIIA(2A)/γ−/− mice treated with SLE or NH sera and analyzed as in (**a**). **f** Blood neutrophils from CD11c-YFP (yellow fluorescent protein) reporter mice treated with SLE-ICs, labeled with a nuclear stain and imaged at day 0 and 2 of culture for nuclear changes (i, iii) and CD11c induction (YFP positive) (ii, iv). Arrows: YFP+ cells with nuclear change. Arrowheads: Unconverted neutrophils. Scale bar = 5 μm; all images are at the same magnification. **g** Live cell imaging of cells treated as in (**f**). Representative images, taken at indicated hours, of nuclear changes and CD11c-YFP acquisition in tracked cells (left) and quantitation of the same (right). Scale bar = 5 μm; all images are at the same magnification. Data are mean ± s.e.m. **a, d** One-way analysis of variance and Dunnett's multiple comparison test, **c, e** Student t-test for unpaired comparisons with Dunn-Bonferroni **\*\***p < 0.005, **\***p < 0.05.

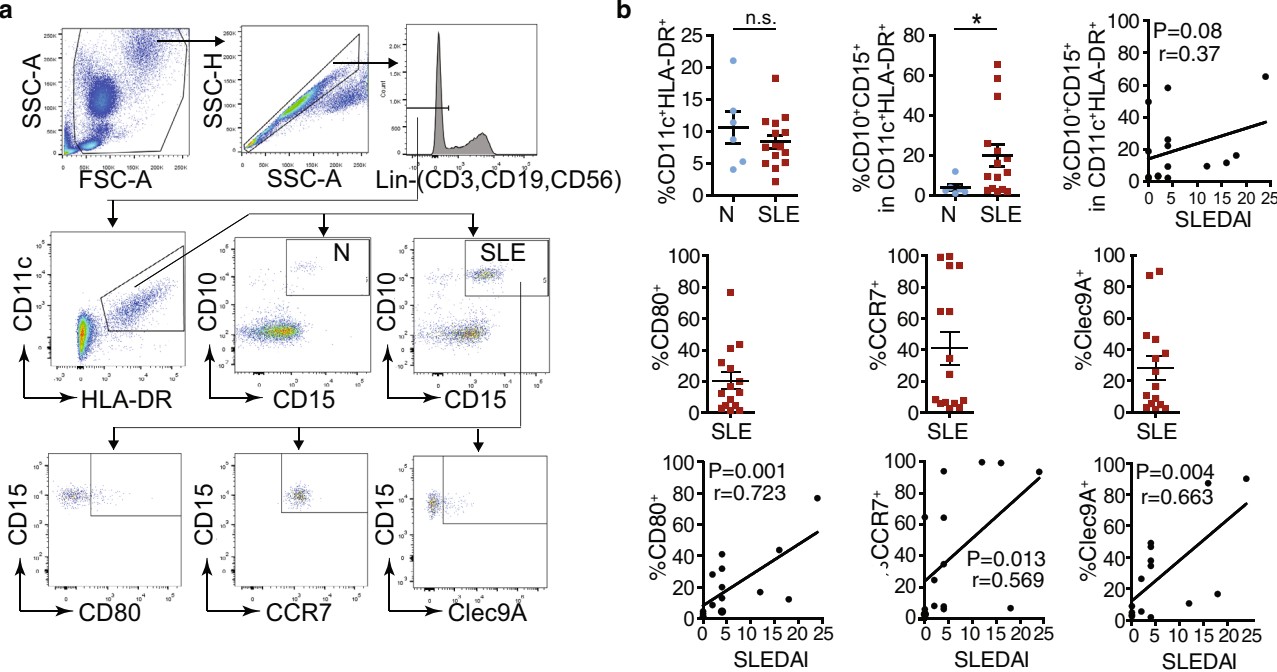

**Fig. 2 The frequency of nAPCs in lupus blood correlates with disease scores. a, b** Blood samples from normal human controls (N) and SLE patients were analyzed for CD11c and HLA-DR markers on lineage negative (Lin−) cells. CD11c-HLA-DR positive cells were examined for neutrophil markers, CD10 and CD15. In SLE samples, CD10 and CD15 positive cells were further evaluated for CD80, CCR7, and Clec9A. Gating strategy for the samples is shown (**a**). The percent of neutrophils with indicated markers was determined and correlated with SLE disease activity index (SLEDAI) scores (**b**). Data are mean ± s.e.m. Non-parametric test using Mann–Whitney analysis; correlation analyzed by a Spearman test. \*p < 0.05.

an association between the frequency of nAPCs in the blood of SLE patients (Supplementary Table 1) and clinical disease. We observed that CD11c and MHCII cells expressing neutrophil markers were more frequent in SLE patients than in normal donors, and that these cells expressed CD80, CCR7, and Clec9A, which is implicated in antigen cross-presentation[41] (Fig. 2a). Importantly, the frequency of nAPCs expressing these markers correlated with SLE clinical severity (SLEDAI) scores (Fig. 2b), suggesting that nAPCs may contribute to the pathogenesis of SLE.

**nAPCs retain the ability to phagocytose and generate reactive oxygen species.** Previous studies showed that GM-CSF treatment of immature neutrophils generates nAPC that capture *E. coli*, release DNA neutrophil extracellular traps (NETs)[23] and generate reactive oxygen species (ROS)[15] like neutrophils. We found that nAPCs generated from mature bone marrow-derived neutrophils treated with SLE-IC, Ova-IC + GM-CSF, or GM-CSF (i.e., anti-Ova alone) were equivalent to neutrophils in their ability to

phagocytose *E. coli* and IgG coated beads (Fig. 3a) and generate ROS in response to *E. coli* and Zymosan (Fig. 3b). Therefore, nAPCs retain some functions of canonical neutrophils.

**nAPCs generated with antibody–antigen complexes present antigen to naive CD4+ T cells, cross-present to CD8+ T cells and generate immunogenic cytokines.** To address whether IC-generated nAPCs can activate naive T cells and cross-present to CD8+ T cells, we assessed their ability to stimulate the proliferation of Ova-peptide specific naive CD4+ and CD8+ T cells isolated from OT-II and OT-I TCR transgenic mice, respectively[42,43]. nAPCs were generated by pretreating neutrophils with Ova-IC, soluble Ova or vehicle (−) and culturing them with GM-CSF. After 3 days, adherent cells, which were primarily nAPCs (Supplementary Fig. 4a) were harvested and co-cultured with T cells. Ova-IC generated nAPCs stimulated proliferation of greater than 80% of naive CD4+ (Fig. 3c) and CD8+ (Fig. 3d) T cells. Ova-IC-β2-microglobulin deficient nAPCs

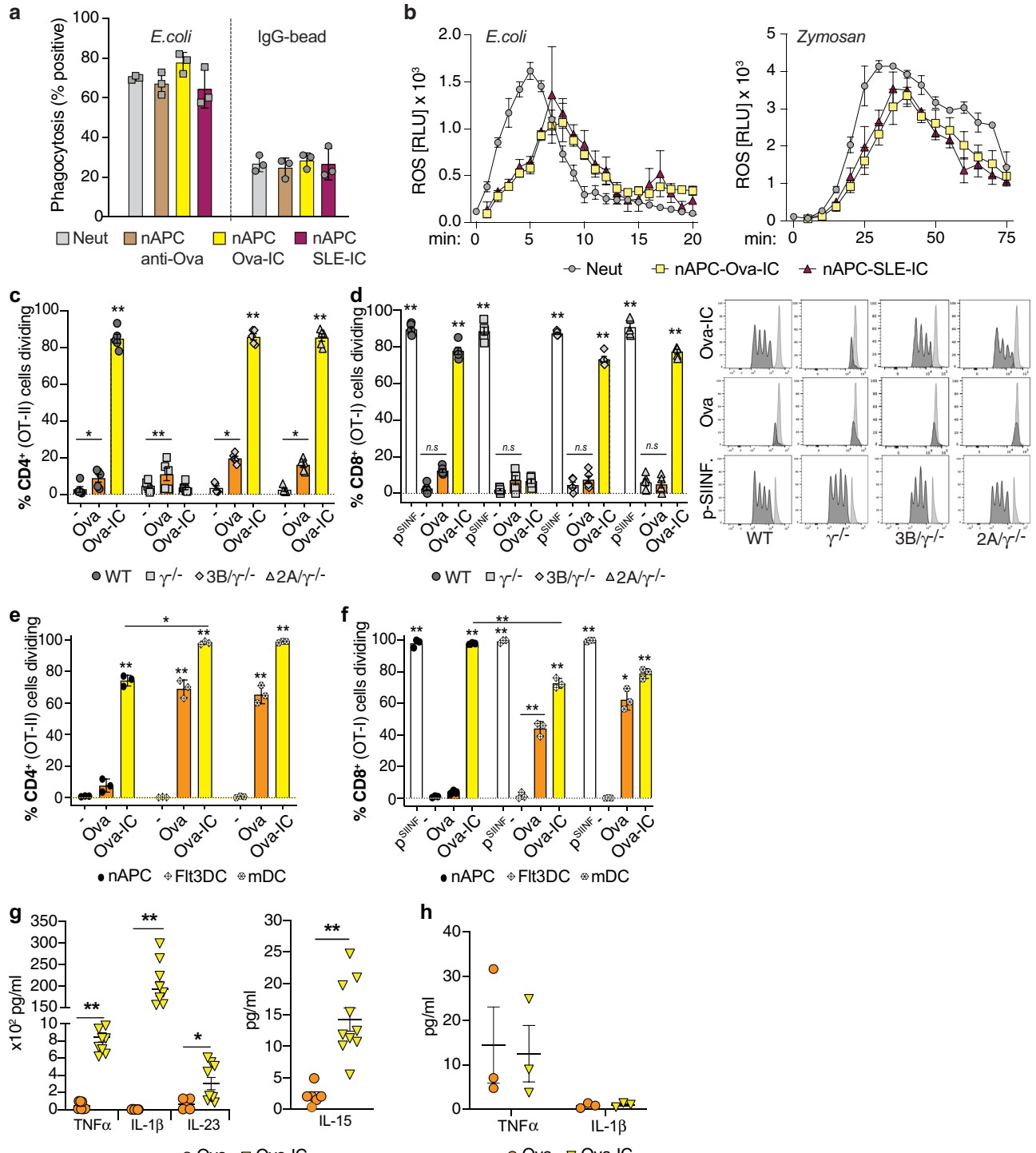

**Fig. 3 Ova-IC generated nAPCs retain neutrophil functions, promote naive CD4$^+$ and CD8$^+$ T cell proliferation and generate immunogenic cytokines. a** Freshly isolated FcgRIIIB(3B)/$\gamma^{-/-}$ BMNs (Neut) or the same treated with anti-Ova, Ova-IC or SLE-IC and cultured to generate nAPCs were incubated with inactivated FITC-*E. coli* or IgG-coated, FITC-labeled latex beads and analyzed for FITC-uptake by flow cytometry. **b** Reactive oxygen species (expressed as relative light units/sec, RLU/s) generated over time by GM-CSF-primed 3B/$\gamma^{-/-}$ Neut and Ova-IC- or SLE-IC- generated nAPCs incubated with serum opsonized *E. coli* or zymosan. **c, d** Proliferation of CellTrace Violet-labeled CD4$^+$ (OT-II) (**c**) and CD8$^+$ (OT-I) (**d**) T cells after co-culture with Ova, Ova-IC or vehicle (−, GM-CSF alone) generated nAPCs of indicated genotypes assessed by CellTrace Violet dilution. In **d**, vehicle generated nAPCs pulsed with Ova SIINFEKL-peptide (pSIINF) is a positive control. Representative profiles for CD8$^+$ T cells (**d**) are shown. **e, f** CellTrace Violet-labeled CD4$^+$ (**e**) or CD8$^+$ (**f**) T cells co-cultured with Ova- or Ova-IC generated nAPC, and Ova or Ova-IC treated splenic monocyte-derived (mDC) or Flt3L-induced splenic DCs, and analyzed as in (**c, d**). **g, h** Cytokine concentrations in supernatant of Ova- and Ova-IC generated nAPCs (**g**) and splenic cDCs (**h**). Data are mean ± s.e.m. For **c**–**f** one-way analysis of variance and Dunnett's multiple comparison test; **g, h** Multiple *t* test between pairs of samples. *$p < 0.05$, **$p < 0.005$.

lacking MHCI failed to stimulate CD8[+] T cells, confirming the dependence of this response on MHCI-restricted cross-presentation of Ova, while presentation to CD4[+] T cells was normal as expected (Supplementary Fig. 4b). TLR agonists, known to enhance antigen presentation by DCs[44,45] were not required, as T cell proliferation was unimpaired in MyD88/TRIF-deficient Ova-IC-nAPCs (Supplementary Fig. 4c). By contrast to Ova-IC-nAPCs, Ova-nAPCs stimulated the proliferation of less than 20% of T cells (Fig. 3c, d), none of which were CD8[+] (Fig. 3d). As a positive control, nAPCs generated in all conditions promoted CD8[+] T cell proliferation when pulsed with high concentrations of Ova-SIINFEKL peptide (Fig. 3d), which directly binds MHC I and bypasses the need for antigen processing and co-stimulatory molecules[46]. Consistent with a requirement for IC engagement of FcγRs to generate immmunogenic nAPCs, Ova-IC-nAPC derived from γ[−/−] neutrophils were impaired in their ability to activate T cells; levels of activation were similar to nAPCs generated with soluble Ova. On the other hand, Ova-IC-nAPCs generated from neutrophils expressing FcγRIIA or FcγRIIIB and WT neutrophils robustly activated T cells (Fig. 3c, d). Differences in antigen internalization cannot account for the significantly greater T cell stimulation by Ova-IC-nAPCs versus Ova-nAPCs as neutrophil uptake of Ova-IC was only twice that of soluble Ova (Supplementary Fig. 4d). The degree of CD4[+] and CD8[+] T cell proliferation supported by Ova-IC nAPCs was comparable to that observed with Ova-IC stimulated Flt3-induced splenic DCs[47] and monocyte-derived cDCs[48] (Fig. 3e, f). Thus, nAPCs generated by engaging FcγRs with Ova-IC are far superior to nAPCs made with soluble Ova in stimulating CD4[+] T cells, and only Ova-IC-nAPCs with functional FcγR cross-present antigen to CD8[+] T cells. Moreover, levels of T cell proliferation were comparable in Ova-IC-nAPCs and Ova-IC treated cDCs.

Cytokines secreted by cDCs promote T cell priming, differentiation and polarization[49,50]. Notably, levels of the key T cell immunomodulatory cytokines IL-1β, TNFα, IL-15, and IL-23, were 5- to 3500-fold higher in supernatants from cultures of Ova-IC versus Ova generated nAPCs (Fig. 3g). In comparison to Ova-IC-nAPCs, cultures of Ova-IC treated splenic cDCs also had significantly less IL-1β, and TNFα (Fig. 3h) and undetectable levels of IL-15 and IL-23. Other cytokines and chemokines examined were present at similar levels in culture supernatants of Ova- and Ova-IC- generated nAPCs and, for the most part, cDCs (Supplementary Fig. 4e, f).

**Adoptively transferred nAPCs generated with antibody–antigen complexes migrate to draining lymph nodes, support a delayed-type hypersensitivity response and are anti-tumorigenic.** Tissue-resident DCs traffic to draining lymph nodes to activate T cells[34]. We observed by intravital microscopy that Ova-IC-nAPCs injected into the mouse footpad accumulated in the draining popliteal lymph node, where they displayed dendritic extensions and made prolonged contacts with Ova-peptide specific CD8[+] T cells. Only a few T cells were present in areas without nAPCs (Fig. 4a, Movies S4 and S5). We also observed that adoptively transferred Ova-IC-nAPCs, but not Ova-nAPCs, elicited a delayed-type hypersensitivity (DTH) response, which requires an effector CD4[+] Th1 T cell-driven recall response to antigen[51] (Fig. 4b). cDCs can acquire pre-formed peptide-MHC complexes from the surface of antigen-loaded APCs, referred to as cross-dressing[52]. However, cross-dressing is unlikely to explain the stimulatory in vivo effects of Ova-IC-nAPCs, as splenic cDCs from CD45.2[+] mice that received adoptively transferred CD45.1[+] Ova-IC-nAPCs failed to promote CD4[+] and CD8[+] T cell proliferation ex vivo, while CD45.1[+] Ova-IC-nAPCs retrieved from the same organ did

(Fig. 4c). In parallel, as a control, we show that in vitro Ova-IC-loaded cDCs and Ova-IC-generated nAPCs comparatively promoted T cell proliferation (Fig. 4c).

We next assessed whether Ova-IC-nAPCs can promote CD8[+] T cell-dependent immunity to B16F10 melanoma cells expressing Ova (B16F10-Ova)[53]. In vitro generated nAPCs were injected intravenously into WT mice prior to the subcutaneous implantation of B16F10-Ova cells. Mice given Ova-IC-nAPCs showed no tumor outgrowth, whereas tumor growth in Ova-nAPC immunized mice was comparable to that of mice given nAPCs without loaded antigen (Fig. 4d). Tumor immunity in Ova-IC-nAPC treated mice was associated with higher accumulation of Ova-specific endogenous CD8[+] T cells in draining lymph node and spleen compared to Ova-nAPC immunized mice (Fig. 4d). Ova-IC-generated nAPC-anti-tumor immunity was CD8[+] T cell-dependent (Fig. 4e). Anti-tumor immunity was intact in mice given Ova-IC-generated MyD88[−/−]TRIF[−/−] nAPCs (Fig. 4d), which excludes a role for TLR ligands, potentially present in experimental reagents and known to elicit anti-tumor responses[54], in the observed results. To examine nAPC trafficking, labeled Ova-IC nAPCs were adoptively transferred into mice implanted with B16F10-Ova tumor cells 6 days prior; 24 h after transfer, Ova-IC-nAPCs were found mostly within the tumor bed, whereas by 96 h these cells were present in draining lymph node and spleen (Fig. 4f). Thus, it is possible that, like cDCs[55], nAPCs facilitate T cell recruitment to tumors and migrate to draining lymph nodes to cross-present endocytosed tumor antigens to CD8[+] T cells.

**An anti-FcγRIII-antigen conjugate promotes neutrophil to nAPC conversion.** FcγRIIIB is sufficient for IC-induced conversion of neutrophils to nAPCs (Fig. 1e) and for antigen presentation (Fig. 3c, d) in vitro. We therefore next asked if binding of a FcγRIIIB antibody conjugated to Ova antigen reproduced the effects of ICs as this could have therapeutic potential. A murine IgG1 antibody specific for FcγRIII, 3G8[56], was conjugated to FITC-labeled Ovalbumin (fOva) and is hereafter referred to as 3G8-fOva. We observed that 3G8-fOva treated humanized neutrophils expressing both FcγRIIA and FcγRIIIB and cultured with GM-CSF acquired DC markers while similar treatment of γ[−/−] neutrophils failed to do so (Fig. 5a). The effect of 38-fOva was seen consistently with several independent conjugate preparations. 3G8 isotype control had no effect (Fig. 5a). Conversion also occurred in humanized neutrophils expressing only FcγRIIIB (FcγRIIIB/γ[−/−]), while it did not occur in γ[−/−] or FcγRIIA/γ[−/−] neutrophils (Fig. 5b), indicating that 3G8-fOva specifically binds FcγRIIIB to promote neutrophil to nAPC conversion.

To explore how the GPI-linked FcγRIIIB triggers neutrophil conversion, we first evaluated whether monomeric 3G8 alone is sufficient or whether 3G8-fOva, a heterogeneous species (generated by conjugation of antibody to fOva) is required. 3G8-fOva treatment of FcγRIIIB/γ[−/−] neutrophils was accompanied by receptor internalization (Fig. 5c) and acquisition of CD11c and MHCII (Fig. 5d), which were not observed with monomeric 3G8 or isotype control (Fig. 5c, d). The endocytic machinery is known to promote receptor signaling and gene transcription[57–60] and our prior work demonstrated that FcγRIIIB and FcγRIIA internalize ICs via a lipid-raft, actin, and cdc42 regulated pathway[61]. Cytochalasin D and MβCD, which disrupt the actin cytoskeleton and lipid rafts, respectively, both abrogated nAPC generation following 3G8-fOva treatment (Fig. 5d) without affecting binding of 3G8 (Supplementary Fig. 5a). Interestingly, these treatments also disrupted FcγRIIIB internalization (Fig. 5e) and nAPC generation (Fig. 5f) induced by Ova-IC or SLE-IC, while generation of nAPCs by GM-CSF alone (in presence of

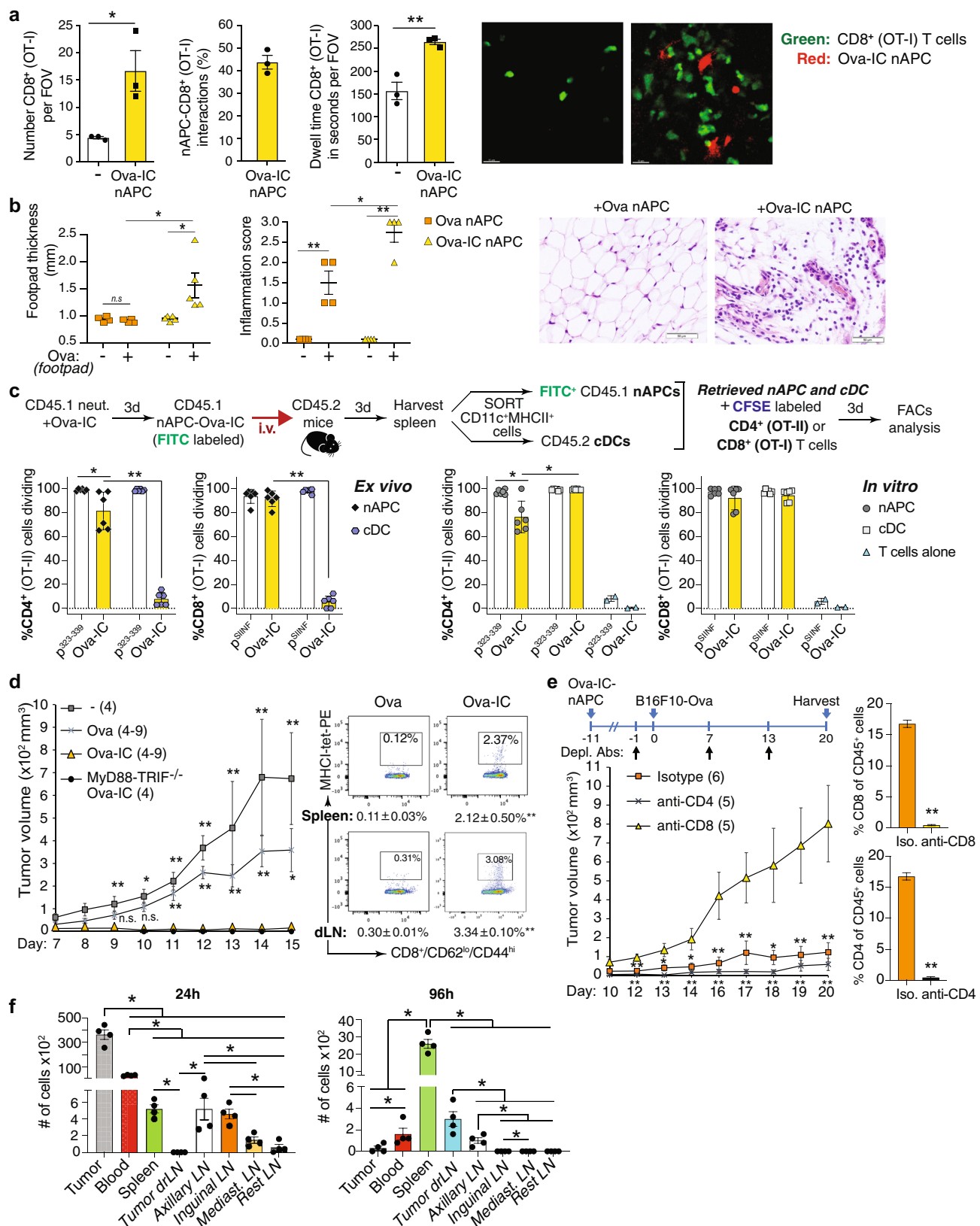

soluble Ova control) was unaffected (Fig. 5f). Thus, with the caveat that we cannot exclude endocytosis independent effects of chemical inhibitors, these findings suggest that FcγRIIIB endocytosis is a common proximal mechanism in 3G8-fOva- and IC-induced neutrophil to nAPC conversion.

**Anti-FcγRIII-antigen conjugate induces FcγR internalization and human neutrophil conversion to nAPCs.** We next evaluated the ability of the 3G8-fOva conjugate to convert human peripheral blood neutrophils from normal volunteers to nAPCs. Ova, a model protein antigen conjugated to 3G8 for mouse models, is irrelevant

**Fig. 4 nAPCs migrate to draining lymph nodes, elicit a delayed-type hypersensitivity (DTH) response, promote anti-tumor immunity, and accumulate in the tumor and draining lymph nodes. a** Ova-IC generated RFP+nAPCs injected in the footpad and GFP-CD8+ T cells injected intravenously detected in the draining lymph node at day 3. Dwell time and number of CD8+ cells in field of view (FOV) and interacting with RFP+nAPC are given. Data are average of three FOV per mouse. Representative images of OT-I cells (green) distribution in areas without (left) and clustering around (red) (right) nAPCs. Scale bar: 15 μm. **b** DTH response in mice given Ova or Ova-IC generated nAPC in footpad and re-challenged with Ova or vehicle. Footpad swelling and inflammation score are given. Representative images of Ova challenged mice immunized with indicated nAPCs. **c** Experimental scheme for analysis of "cross-dressing". Endogenous cDC and Ova-IC loaded nAPCs (injected 3 days prior) were retrieved from spleen and co-cultured with Cell Trace Violet labeled CD4+ (OT-II) and CD8+ (OT-I) T cells ex vivo. In parallel, CD4+ and CD8+ T cell proliferation on in vitro generated Ova-IC-nAPCs and Ova-IC loaded cDCs was assessed. Direct loading of nAPCs and cDCs with Ova-peptide for CD4+ (p323–339) or CD8+ (pSIINFEKL) T cells were controls. Proliferation of CD4+ or CD8+ T cells for ex-vivo (left panels) and in-vitro (right panels) samples are shown. **d** Tumor volumes in wild-type (WT) mice immunized with no Ova- (−), Ova- or Ova-IC- generated WT nAPCs or Ova-IC-MyD88/TRIF−/− nAPCs 7 days prior to s.c. B16F10-Ova tumor cells (left panel). Number of mice per group is in parenthesis. Flow cytometric plots for MHCI-tetramer-PE+, Ova-specific CD8+ T cells from spleen and draining lymph nodes (dLN) and gated on CD8+CD62loCD44hi population (right panels). **e** Experimental scheme for examining the effect of CD8+ or CD4+ T cell depleting antibodies (arrows) on Ova-IC-nAPC induced reduction in tumor growth over time (left panel). The number of mice per group is in parenthesis. CD8+ and CD4+ T cell depletion quantitated by FACS analysis of blood samples (right panels). **f** Quantification of Ova-IC-nAPCs accumulated in organs 24 h or 96 h after their i.v. injection in mice injected 6 days prior with B16F10-Ova cells. Data are mean ± s.e.m. **a** unpaired student *t*-test, **b, c, d** one-way ANOVA and Dunnett's multiple comparison test, **e** Two-way ANOVA and Tukey's multiple comparison test (left panel) and Student *t*-test for unpaired comparisons with Dunn-Bonferroni (right panel), and **f** Wilcoxon signed-rank test. *$p < 0.05$; **$p < 0.005$.

for all studies with human samples. Whole blood was first treated with 3G8-fOva or isotype control, and neutrophils were isolated and cultured with GM-CSF. 3G8-fOva treatment resulted in the rapid internalization of FcγRIIIB and a slower internalization of FcγRIIA (Fig. 5g). Subsequent culturing of purified human neutrophils in GM-CSF for 2 days led to a significant increase in CD11c, HLA-DR, T cell co-stimulatory molecules, and CCR7 in 3G8-fOva versus isotype control treated samples (Fig. 5h). High purity of the starting neutrophil population (Fig. 5i, Supplementary Fig. 5b) and a representative flow cytometric profile of nAPCs at day 2 of culture are shown (Fig. 5i). Greater than 15% of neutrophils converted to nAPCs in 4 of 7 healthy volunteers, while the 3 remaining volunteers tested had lower conversion (<3% conversion) and were not included. The basis for this donor-dependent variation could reflect the presence of neutrophil subpopulations of varying sensitivity[62,63] and/or differences in neutrophil priming[64] by soluble chemokines/cytokines, which can vary 2–3 log fold[65] in blood of normal humans. To assess whether human nAPCs adopted a mononuclear morphology, as observed for murine nAPCs, we flow-sorted 3G8-fOva- and isotype-treated neutrophils based on CD11c and HLA-DR surface markers. CD11c+HLA-DR+ cells in 3G8-fOva treated samples exhibited a mononuclear appearance (Fig. 5j), were positive for CD15 and CD66b, markers of mature neutrophils[66], and lacked monocyte (CD14) and T cell, B cell, and NK cell lineage markers (Fig. 5j), whereas residual unconverted neutrophils, in both 3G8-fOva and isotype control treated samples, retained their polymorphonuclear appearance (Fig. 5j).

**Anti-FcγRIIIB-antigen conjugate generates functional nAPCs that elicit CD8+ T cell proliferation and target cell killing in vivo.** We next sought to determine whether infusion of 3G8-fOva converts neutrophils into functional nAPCs in vivo. Blood neutrophils in FcγRIIIB/γ−/− mice internalized intravenously injected 3G8-fOva (Supplementary Fig. 6a), leading to the appearance of FITC+Ly6G+CD11c+MHCII+ cells expressing CD80 and CCR7 (Fig. 6a) by 24 h. These changes were not observed in similarly treated γ−/− mice. FcγRIIIB and FcγRIIA are expressed on a subpopulation of CD11b+CD115+Ly6Chi monocytes in our humanized mice[31] (Supplementary Fig. 6b), but >97% of 3G8-fOva-positive CD11c+MHCII+ cells in the spleen expressed the neutrophil marker Ly6G and were Ly6Clo (Supplementary Fig. 6c), consistent with an origin from converted neutrophils. Moreover, FcγRIIA and FcγRIIIB were undetectable in macrophages[31] and CD11c+ splenic DCs (Supplementary Fig. 6d). Next, we examined the accumulation of nAPCs in tissues and secondary lymphoid

organs in mice given 3G8-fOva conjugate or monomeric 3G8 control in the absence or presence of GM-CSF. By 72 h, nAPCs generated in mice given 3G8-fOva accumulated in lymph nodes, spleen, lung and liver (Fig. 6b), similar to the reported tropisms of blood cDCs[67], whereas 3G8 treatment did not result in significant nAPC accumulation in any organs. Although GM-CSF treatment produced a small increase in the frequency of nAPCs in some tissues of mice given 3G8 alone (Fig. 6b), it did not in mice treated with 3G8-fOva.

We next examined the ability of 3G8-fOva-induced nAPCs generated in vivo to promote CD4+ T cell proliferation, using MHCII deficient mice to rule out effects of endogenous APCs on outcomes. These mice were infused with FcγR humanized neutrophils, treated with vehicle, 3G8 antibody, or 3G8-fOva, and then administered naive CellTrace Violet-labeled Ova-specific CD4+ T cells. Robust CD4+ T cell proliferation was observed in the spleen and lymph nodes of MHCII deficient recipient mice treated with 3G8-fOva, whereas no proliferation was observed in mice given 3G8 or vehicle control (Fig. 6c). In addition, 3G8-fOva treatment of humanized FcγRIIIB/γ−/− mice promoted the proliferation of adoptively transferred Ova-specific CD8+ T cells (Fig. 6d). Next, the function of endogenous CD8+ T cells activated by nAPCs in vivo was assessed in a target cell killing assay. For this, 3G8-fOva immunized humanized mice expressing FcγRIIIB and/or FcγRIIA, or mice lacking FcγRs (γ−/−) were injected with Ova-SIINFEKL peptide pulsed and unpulsed target splenocytes, each loaded with a different amount of fluorescent label to distinguish the populations by FACS analysis. We observed killing specifically of splenocytes pulsed with the Ova peptide only in FcγRIIIB expressing humanized mice (Fig. 6e), an effect that was abrogated by neutrophil depletion (Fig. 6f). Thus, 3G8-fOva creates nAPCs that promote the proliferation of CD4+ and CD8+ T cells and the generation of endogenous, cytotoxic CD8+ T cells.

**Anti-FcγRIIIB-Ova antigen conjugate is anti-tumorigenic in mice.** Given the anti-tumorigenic activity of Ova-IC-nAPCs generated in vitro in the presence of GM-CSF (see Fig. 4d) and the success of anti-tumor strategies involving GM-CSF[68–72], we combined a single dose of 3G8-fOva with sequential doses of GM-CSF. Tumor growth in 3G8-fOva-immunized mice expressing FcγRIIIB and FcγRIIA was significantly reduced compared to unimmunized, GM-CSF treated humanized mice (Fig. 7a, Supplementary Fig. 7a) and two additional sets of GM-CSF treated controls: Wild-type mice given fOva, which will lead to an anti-tumor immune response driven by endogenous APCs,

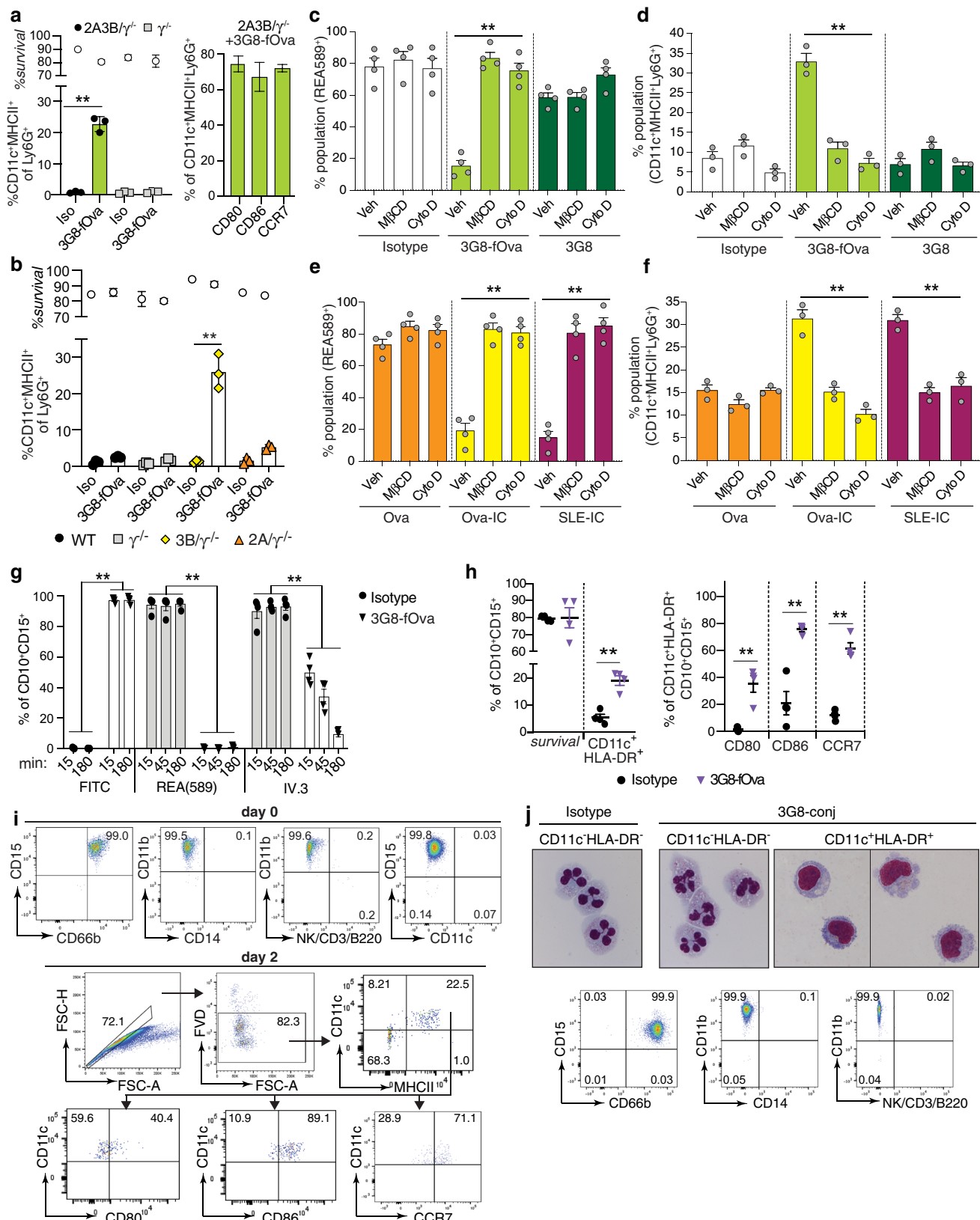

such as cDCs, and $\gamma^{-/-}$ mice given 3G8-fOva that do not make nAPCs as they lack the FcγRIIIB but nonetheless, like wild-type plus fOva, will mount an immune response to fOva (contained in the 3G8-fOva conjugate) (Fig. 7a). The reduction in tumor growth in 3G8-fOva immunized mice expressing FcγRIIIB and FcγRIIA correlated with the generation of Ova-specific CD8[+]

effector T cells and higher CD8[+] effector to Treg cell ratios in the spleens of these mice versus other groups (Fig. 7b, Supplementary Fig. 7b). In summary, a single immunization of GM-CSF treated mice with 3G8-fOva attenuates melanoma growth and results in the generation of antigen-specific CD8[+] effector T cells.

**Fig. 5 An anti-FcγRIIIB-antigen conjugate converts murine and human neutrophils to nAPCs that is dependent on FcγRIIIB endocytosis. a** Blood neutrophils from 2A3B/γ$^{-/-}$ and γ$^{-/-}$ mice incubated with isotype (Iso) or 3G8-fOva, cultured with GM-CSF and analyzed after 3 days for CD11c, MHCII on Ly6G positive cells. CD11c$^+$MHCII$^+$Ly6G$^+$ cells in 2A3B/γ$^{-/-}$ + 3G8-fOva sample further analyzed for CD80, CD86, and CCR7. **b** CD11c and MHCII acquisition after 3G8-fOva or isotype treatment of murine neutrophils from indicated mouse strains and culture with GM-CSF for 3 days. **c, d** 3B/γ$^{-/-}$ bone marrow neutrophils pretreated with vehicle, methyl-beta cyclodextrin (MβCD) or Cytochalasin D (Cyto D) and treated with isotype, 3G8-fOva or 3G8 antibody. FcγRIIIB on the cell surface evaluated using anti-FcγRIIIB, clone REA589 (**c**). Cells from **c** analyzed for the percent of cells acquiring CD11c and MHCII after 3 days in culture with GM-CSF (**d**). **e, f** Cells treated with Ova, Ova-IC, or SLE-IC and analyzed for surface FcγRIIIB (**e**) and acquisition of CD11c and MHCII on Ly6G$^+$ cells (**f**) as in (**c, d**). **g, h** Human blood neutrophils treated with 3G8-fOva or isotype control. Evaluation of surface FcγRIIIB (assessed with REA589 antibody), FcγRIIA (assessed with IV.3 antibody) and FITC (fOva) positivity over time (**g**). The percent of CD11c and HLA-DR positive cells with neutrophil markers (CD15, CD10) after 2 days in culture with GM-CSF. CD11c and HLA-DR double-positive cells further examined for CD80, CD86, and CCR7 (**h**). **i** Flow plots assessing the purity of human neutrophils at day 0 (upper panels) using markers for neutrophils (CD15, CD66b), DCs (CD11c), monocytes (CD14), B cells (B220), NK cells (NK1.1), and T cells (CD3) (Lin−). Gating strategy for nAPC markers at day 2 after 3G8-fOva treatment (lower panels). **j** Images of purified human neutrophils, treated as in (**g, h**), FACS sorted on day 1 and stained with Wright-Giemsa. FACS profiles of day 0 human blood neutrophils stained for markers of neutrophils (left), monocytes (middle) and NK, T and B cells (right). Data are mean ± s.e.m. **c–g** One-way analysis of variance and Dunnett's multiple comparison test; **a, b, h** Multiple *t*-tests between pairs of samples. \*\*$p < 0.005$.

**FcγRIIIB engagement of human neutrophils from patients with myeloid neoplasia or vaccinated normal human volunteers generate nAPCs that activate autologous memory T cells.** To explore the therapeutic relevance of our results, we tested if 3G8-fOva treatment of human neutrophils expressing endogenous tumor antigens or loaded with exogenous pathogen associated antigens could generate nAPCs that stimulate autologous T cells. As already noted, the Ova in the 3G8-fOva conjugate is irrelevant in the human samples. For tumor antigen responses, we used neutrophils isolated from patients with myeloid neoplasms, in which neutrophils are components of the malignant clone and hence express tumor neoantigens[73]. We selected two patients that harbored several driver mutations; based on high variant allele frequencies and differential cell counts (Supplementary Table 2), these mutations can be inferred to be present in the neutrophil fractions of these patients. 3G8-conjugate treatment converted neutrophils from both patients into nAPCs, which activated autologous CD3$^+$ T cells as assessed by IFNγ secretion. No IFNγ production was observed with nAPCs generated from three normal human volunteers incubated with autologous T cells indicating that the response seen in patient samples is due to neutrophils carrying neoantigens (Fig. 7c). For responses to exogenous pathogen antigens, neutrophils from normal volunteers previously immunized with the Tdap (tetanus, diptheria, and pertussis) vaccine[74] were treated with 3G8-fOva or isotype control antibody, pulsed with the vaccine antigens, and incubated with autologous T cells. IFNγ generation was observed in cultures of 3G8-fOva- but not isotype-treated neutrophils loaded with vaccine antigens (Fig. 7d). Together, these studies indicate that human nAPCs can stimulate the response of autologous memory T cells to neutrophils carrying tumor neoantigens and exogenously provided vaccine antigens.

**Transcriptional profiling reveals the uniqueness of nAPCs and transcription factor PU.1 as a potential driver of neutrophil to nAPC conversion.** We next sought to define the transitional cell states and uncover potential molecular mechanisms underlying neutrophil to nAPC conversion by conducting single-cell RNA-seq (scRNA-seq) on (i) splenic Ly6G$^+$CD11c$^+$MHCII$^+$ and Ly6G$^+$CD11c$^-$MHCII$^-$ cells (presumptive nAPCs and neutrophils, respectively) isolated from FcγR humanized mice 3 days after i.v. injection of 3G8-fOva treatment and (ii) isolated neutrophils harvested at day 0 and cultured in vitro with 3G8-fOva for 1 to 3 days. We also analyzed splenic cells from untreated wild-type mice and FcγR humanized mice as reference cell types (Supplementary Fig. 8). We identified five neutrophil subpopulations, Nt.1-Nt.5, and two CD11c$^+$MHCII$^+$ clusters,

referred to as nAPCs, which could be further divided into six sub-clusters (Fig. 8a). Importantly, similar neutrophil and nAPC cell states were identified in vitro and in vivo (Fig. 8a). These states were robust with respect to potential batch effects (Supplementary Fig. 9) despite the challenges of profiling neutrophils, which are fragile and have low RNA content[75–77]. These factors explain the observed cluster of monocytes on day 0 in vitro, as these cells (which were >99% neutrophils as assessed by flow cytometry, Supplementary Fig. 1a) are likely to be disproportionally represented in scRNA-seq analyses relative to neutrophils. Importantly, these contaminating monocytes are transcriptionally distinct from Nt and nAPC populations (Fig. 8a). We also observed DC, monocyte, macrophage, and NK clusters in splenic samples from untreated wild-type and humanized FcγR mice (Fig. 6a, Supplementary Data 1).

Next, we wanted to understand the transcriptomic features that are associated with two identified nAPC clusters that were heterogeneous and separated from each other in UMAP space, in both trajectory and correlational analyses (Fig. 8a, c, Supplementary Fig. 10a, Supplementary Data 2). A heatmap of differentially expressed genes revealed greater similarity of neutrophils and the nAPC.1 subset than of the nAPC.1 and nAPC.2 subsets (Supplementary Fig. 10b). The close relationship between neutrophils and the nAPC.1 subset was further supported by pseudo-bulk gene correlation analysis (Supplementary Fig. 10a) and the expression of neutrophil-specific chemokines by nAPC.1 cells (e.g., Cxcr2) (Supplementary Fig. 11). nAPC.1 cells differed from Nt.1 cells by showing upregulation of immune stimulatory pathways (e.g., NFκB, JAK/STAT, cytokine, and chemokine signaling) (Supplementary Fig. 12). In contrast to Nt.1 and nAPC.1 cells, nAPC.2 cells shared many differentially expressed genes with DCs, including MHCII and genes associated with antigen processing and presentation (Supplementary Figs. 11 and 12). Most nAPC.2 cells (particularly subset d) also expressed high levels of Ccl17, Ccl22, and Cxcl16 (Supplementary Fig. 11), known to stimulate effector T cell function and/or migration[78], and included a larger fraction of cells expressing Ccr2 and Ccr5, which promote DC activation and maturation[79,80]. Comparison of the nAPC.1 and nAPC.2 subsets revealed that nAPC.2 cells showed upregulation of genes within the antigen processing and presentation pathway and pathways related to protein processing and endocytosis (Supplementary Fig. 13a). Within the neutrophil compartment, Nt.1/Nt.2 cells showed high expression of canonical neutrophil markers and genes associated with cytotoxicity (e.g., Lcn2, Cybb, Camp) (Supplementary Fig. 11), while Nt.5 cells corresponded to a more mature neutrophil subset (Supplementary Fig. 13b, Supplementary Data 3) previously described in peripheral blood and mouse spleen[63].

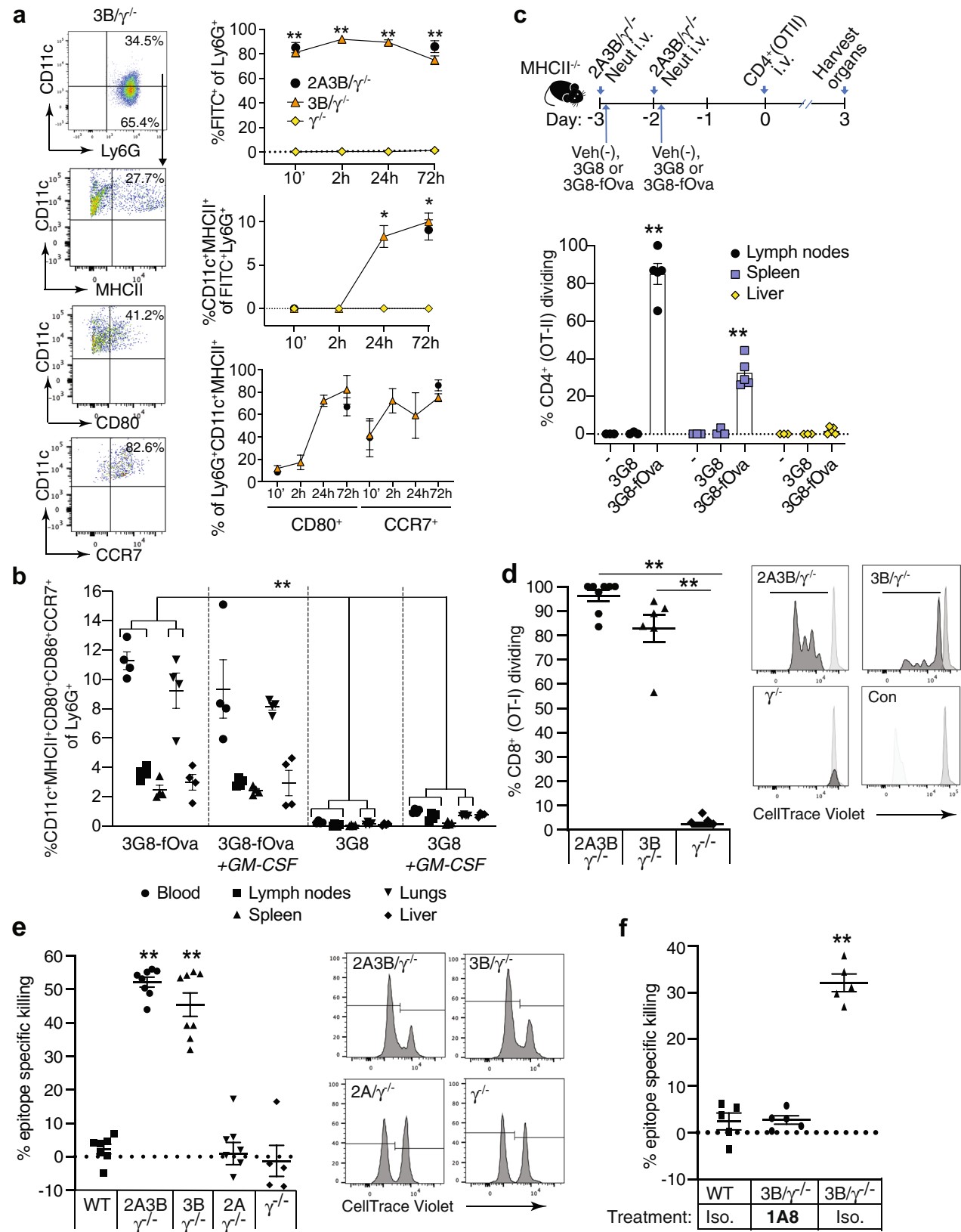

Characterization of neutrophil to nAPC transition based on in vitro time-course data suggested two lineage trajectories: Nt.1 to nAPC.1 cells; and Nt.1 through Nt.2-Nt.4 cells to nAPC.2 cells (Fig. 8b, Supplementary Fig. 13b). We further explored these trajectories by using Monocle to order cells along a pseudo-time gradient, with Nt.1 cells defined as the starting node (Fig. 8c).

Monocytes exhibited large transcriptomic differences from Nt and nAPC cells (Fig. 8c), indicating that they are unlikely nAPC progenitors. Consistent with neutrophil transition to a cDC-like state, the pseudo-time gradient of Nt.1 to nAPC.2 transition correlated with reduced Ly6G expression and increased expression of CD11c and genes related to antigen presentation (Fig. 8c, Supplementary

**Fig. 6 Intravenously injected anti-FcγRIIIB-antigen conjugate in FcγR humanized mice generates nAPCs that accumulate in secondary lymphoid organs and promote T cell proliferation and target cell killing. a** Representative flow cytometry plots for blood 24 h after i.v. injection of indicated mice with 3G8-conjugate (3G8-fOva) (left panel). Percent of 3G8-fOva uptake (FITC+) (upper panel), acquisition of CD11c and MHCII on Ly6G+ cells (middle panel) and CD80 and CCR7 on the CD11c+MHCII+Ly6G+ population (lower panel) at indicated time points. **b** Percent CD11c+MHCII+ CD80+CD86+CCR7+ of Ly6G+cells in indicated organs 3 days after i.v. injection of 3G8-fOva or 3G8 without or with GMCSF. **c** Timeline of indicated treatments to assess CD4+ (OT-II) T cell proliferation in mice lacking MHCII; percent dividing Cell Trace Violet-labeled adoptively transferred CD4+ (OT-II) T cells, as assessed by CellTrace Violet dilution, in indicated organs. **d** Proliferation of adoptively transferred CellTrace Violet-labeled CD8+ (OT-I) T cells in spleen of indicated mice immunized 10 days prior with 3G8-fOva (left panel). Representative histograms of dividing (bar) and starting (Con, light gray) population of CD8+ T cells are shown. **e** Epitope-specific, target cell killing by endogenous CD8+ T cells following injection of Ova-peptide, SIINFEKL, pulsed or unpulsed target cells in indicated mice immunized 7 days prior with 3G8-fOva. FACS plots displaying CellTrace Violet-high (right peak) represent epitope-specific target cells, while low CellTrace Violet (left peak) represent unpulsed target cells. A relative reduction in the right versus the left peak indicates epitope-specific target cell killing. **f** Epitope-specific killing assay as in (**e**) in indicated mice treated with the neutrophil-immunodepleting antibody 1A8 or an isotype control. Data are mean ± s.e.m. One-way analysis of variance and Dunnett's multiple comparison test was used for comparison of multiple groups. *p < 0.05, **p < 0.005.

Fig. 13c). We then used a linear model to identify all genes correlated with pseudo-time and constructed a trajectory gene score for all cells obtained from spleen (Supplementary Data 3). Predictions based on in vitro trajectories captured inferred differences among in vivo cells, with sorted Ly6G+CD11c−MHCII− and Ly6G+CD11c+MHCII+ cell populations, which are presumptive neutrophils and nAPCs, respectively, falling at two ends of the gene score (Fig. 8d).

Since GM-CSF alone promotes expression of DC-like markers on neutrophils, we investigated whether FcγR engagement induced a gene program distinct from that of GM-CSF by analyzing bulk RNA-seq data obtained from nAPCs generated in vitro with Ova- or Ova-IC (in the presence of GM-CSF), or SLE-IC alone. Principal component analysis revealed that PC1 separates neutrophils and nAPCs in each of the four treatment groups (Supplementary Fig. 13d), differences that were also captured by the trajectory gene score for these samples (Fig. 8e). While changes induced by these 4 distinct treatments were similar, we also identified a set of genes that were differentially regulated in Ova-IC versus Ova generated nAPCs that may account for the immunogenicity of the former (Supplementary Data 4).

**Inhibition of PU.1 prevents neutrophil to nAPC conversion.** To identify transcription factors involved in reprogramming neutrophils to nAPCs, we performed transcription factor motif enrichment analysis (using MARGE-cistrome)[81] on genes whose expression significantly correlated with pseudo-time in the single cell data set, capturing the transition to nAPC.2. STRING database[82] analysis revealed the activation of two transcriptional factor networks in nAPCs, PU.1 and NFκB/Stat/Irf8[83] (Fig. 9a). PU.1 is a myeloid-specific transcription factor with chromatin remodeling activity[84]. Two compounds (DB2313 and DB2115) that selectively inhibit PU.1 interaction with DNA[85] blocked neutrophil to nAPC conversion in response to Ova, Ova-IC and SLE-IC in vitro without affecting cell survival (Fig. 9b, Supplementary Fig. 14a). PU.1 inhibitors also abrogated 3G8-conjugate-induced conversion of human neutrophils to nAPCs (Fig. 9c). Conditional deletion of PU.1 in neutrophils in Spi1flx/flxMRP8-Cre mice results in normal numbers of neutrophils[86] that we found were capable of internalizing Ova and Ova-IC (Supplementary Fig. 14b) but were impaired in conversion to nAPCs (Fig. 9d, Supplementary Fig. 14c, d). Moreover, the few CD11c+MHCII+ cells produced from PU.1 deficient cells had reduced expression of CCR7 (Fig. 9d), suggesting a direct role for PU.1 in CCR7 expression. Importantly, Ova-IC treated PU.1-deficient cells failed to stimulate CD8+ T cell proliferation (Fig. 9e), whereas our controls, Ova SIINFEKL-peptide-pulsed WT and PU.1 deficient nAPCs both promoted CD8+ T cell proliferation (Fig. 9e). Thus, PU.1 is required for neutrophil conversion to immunogenic, IC-induced nAPCs.

## Discussion

Our work shows that binding of ICs or a human FcγRIIIB antibody–antigen conjugate rapidly converts mature neutrophils to nAPCs in a process that requires FcγR endocytosis. The nAPCs generated by engaging FcγRs acquire antigen-presenting properties and secrete immunomodulatory cytokines that far exceed those made with soluble antigen alone. They are as immunogenic as classical DCs, and accumulate in secondary lymphoid organs where they interact with T cells. The transcription factor PU.1 is required for nAPC generation following either GM-CSF stimulation and/or FcγR engagement, which suggests the existence of a common transcriptional pathway for nAPC generation. Antigen-specific nAPCs that activate T cells with anti-tumorigenic properties can be generated by administering anti-FcγRIII conjugated to a defined antigen, and clonal neutrophils harboring neoantigens from patients with myeloid neoplasms can be converted to immunogenic nAPCs that activate T cells in an antigen-agnostic manner. Both of these approaches may be deployed as T cell-based immunotherapeutic strategies for the treatment of cancer. We also provide evidence that the generated human nAPCs efficiently process vaccine bacterial toxins and activate T cells, suggesting a potential way to combat infections. Conversely, nAPC frequency in the blood of patients with SLE correlates with disease severity, which suggests that nAPCs may be pathogenic in autoimmune disorders.

We provide several complementary types of evidence that nAPCs are produced from neutrophils. Using live cell imaging, we demonstrated the transition of SLE-IC treated mature, polymorphonuclear blood neutrophils to morphologically distinct CD11c+ cells with mononuclear-like appearance. Moreover, scRNAseq analyses performed on purified neutrophils treated in vitro and splenocytes from 3G8-fOva conjugate treated mice revealed similar alterations in gene expression that are consistent with differentiation of neutrophils into two nAPC subsets, nAPC.1 and nAPC.2. Based on greater enrichment of genes in the antigen processing and presentation pathway, nAPC.2 appear to be more immunogenic than nAPC.1. We also identify PU.1 and NFκB/STATs/Irf8 transcription factor networks as being associated with the reprogramming of neutrophils treated with FcγRs and/or GM-CSF, suggesting a common gene regulatory circuitry underlying nAPC generation. PU.1 may function as a pioneer factor that increases chromatin accessibility for "primary response factors" such as NFκB/STAT, which do not require de novo synthesis[87]. Although a set of differentially regulated genes found only in Ova-IC but not Ova generated nAPCs may be responsible for the immunogenicity of Ova-IC-nAPCs, post-transcriptional or epigenetic changes may additionally contribute to their

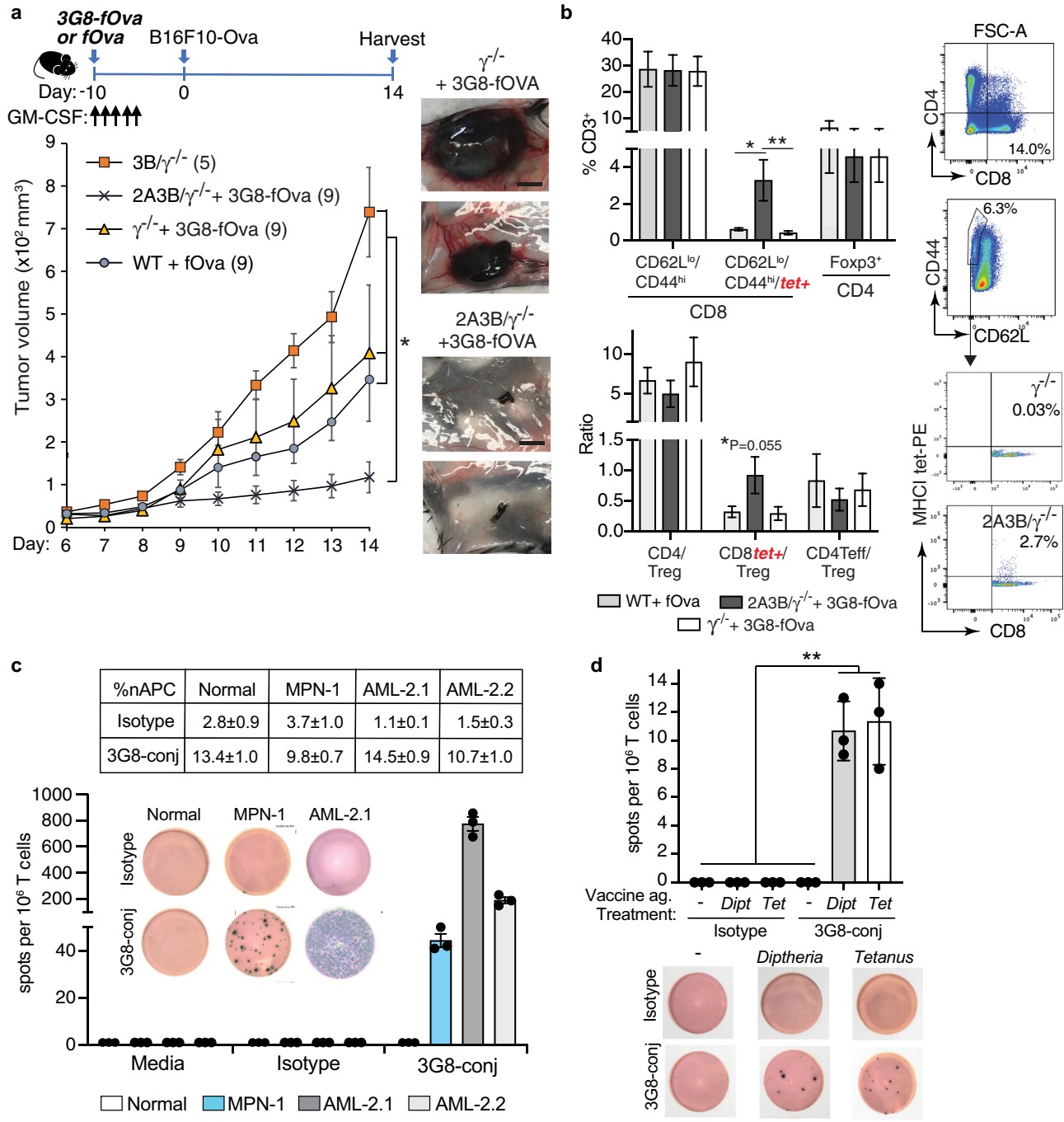

**Fig. 7 Anti-FcγRIIIB-antigen conjugate is anti-tumorigenic in FcγR humanized mice and converts human neutrophils into nAPCs that activate autologous T cells. a, b** Schematic for the timeline of indicated treatments for examining the effect of 3G8-fOva on B16F10-Ova tumor growth (top panel). Tumor volume was assessed at indicated times after s.c. injection of B16F10-Ova melanoma cells. The number of mice per group is in parenthesis (lower panel). Representative images of harvested tumors are shown, scale bar = 5 mm (**a**). Spleens of indicated mice from (**a**) at tumor harvest were analyzed for frequency of Ova-peptide specific effector CD8+ T cells (CD62L^lo CD44^hi) using MHCI-tetramers (tet+), and CD4+ and Treg (Foxp3+, CD4+) cells (**b**). **c** Neutrophils from a patient with primary myelofibrosis (a myeloproliferative neoplasm, MPN) or acute myeloid leukemia (AML) (AML-2.1, AML-2.2 representing two independent blow draws) were treated with 3G8-conjugate (3G8-fOva) or isotype, cultured for 2 days and then co-cultured with autologous T cells and analyzed for IFN-γ secretion on a human IFN-γ ELISpot plate. The percent nAPC generation is shown in the table (avg ± s.e.m.). T cell responses quantified as the number of spots per 1 × 10^6 cells on a human IFN-γ ELISpot plate is reported and representative images of the same are shown. **d** Autologous T cell responses against diphtheria toxin (Dipt) or tetanus toxoid (Tet) loaded isotype or 3G8-fOva (3G8-conj) treated neutrophils from normal human volunteers were evaluated using an IFN-γ ELISpot plate as in (**c**). Representative images are shown. Data are mean ± s.e.m. One-way analysis of variance and Dunnett's multiple comparison test was used for comparison of multiple groups except for a), for which statistical analysis is described in Supplementary Fig. 7a. *p < 0.05, **p < 0.005.

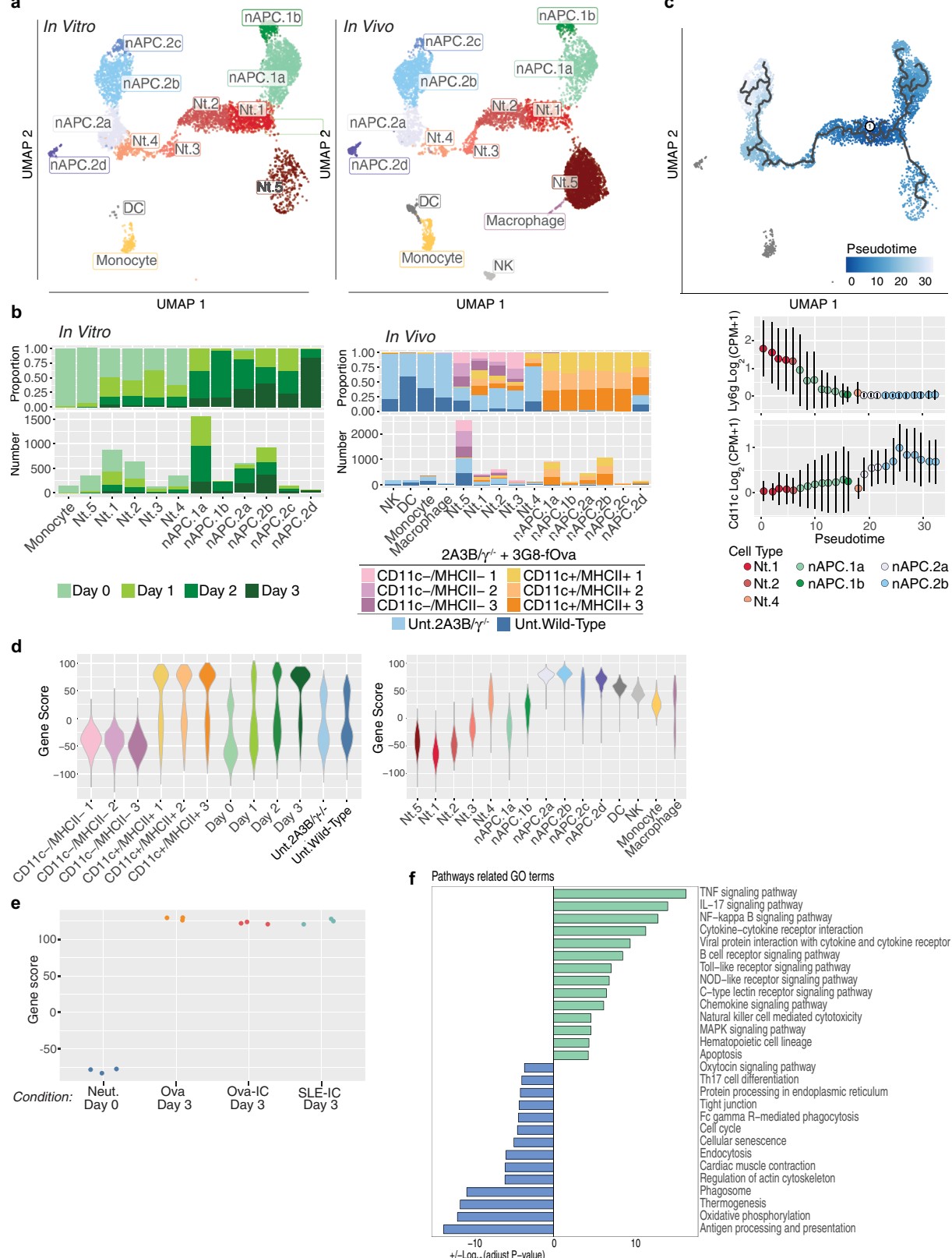

acquisition of immunogenic functions, a fruitful area for future studies.

Endocytosis of the GPI-linked FcγRIIIB following 3G8-fOva engagement preceded and was essential for nAPC conversion, consistent with evidence that endosomes are signaling hubs required for diverse processes from migration to cell fate[88]. Endocytosis was also required for IC-induced nAPC generation suggesting that the 3G8-fOva conjugate and ICs engage a similar proximal signaling pathway for nAPC generation. Antigen endocytosis that occurs in parallel with FcγRIIIB internalization

**Fig. 8 Transcriptome analysis reveals the uniqueness and heterogeneity of nAPCs. a–d** In vitro, neutrophils isolated from 2A3B/$\gamma^{-/-}$ mice were harvested at day 0, or at day 1, 2, or 3 after treatment with 3G8-fOva. In vivo, 2A3B/$\gamma^{-/-}$ mice were given 3G8-fOva i.v. and spleens were harvested 3 days later and FACS sorted on Ly6G$^+$CD11c$^+$MHCII$^+$ and Ly6G$^+$CD11c$^-$MHCII$^-$ cells. Unsorted splenic cells from an untreated 2A3B/$\gamma^{-/-}$ and wild-type mouse were also harvested. Cells were subjected to scRNA-seq. **a** UMAP projection of all in vitro and in vivo cells, colored by identified cell types. A negligible number of in vitro cells clustered with DCs ($n = 10$). **b** Proportion of each cluster by hashtag and number of cells coming from each hashtag-ed population in each cluster and separated by in vitro cells and in vivo cells. **c** UMAP projection of in vitro data with learned principal graph overlaid. Pseudo-time ordering of cells in same UMAP projections. Mean normalized expression of Ly6g and Cd11c (Itgax), binned along pseudo-time. **d** Trajectory gene score of hashtag and cell type populations generated with genes associated with pseudo-time. **e** Neutrophils from wild-type mice, treated with Ova or Ova-IC and cultured with GM-CSF, or SLE-IC and cultured without GM-CSF were harvested after 3 days and freshly isolated neutrophils (Day 0) and subjected to bulk RNA-seq. Trajectory gene score of populations generated under indicated conditions is shown. **f** Pathway analysis of top 200 DEG by pseudo-bulk counts among nAPC.1 s (green) and nAPC.2 s (blue).

is likely targeted to an endocytic pathway with low degradative potential for antigen processing, as in cDCs[89]. Indeed, the exposure of neutrophils to complexed antigen (Ova-anti-Ova ICs) or the 3G8-fOva conjugate for only 2 h was sufficient for subsequent Ova cross-presentation by newly generated nAPCs, which suggests the existence of a pathway for antigen processing in neutrophils that is distinct from degradative routes[12]. This possibility was not appreciated in previous work studying acquisition of DC-like functions by neutrophils because antigen was present, when noted, throughout the time course of the experiments[15,90,91].

Key features of dendritic cells include their ability to generate immunomodulatory cytokines, migrate to lymph nodes and capture blood borne antigens in the spleen[34]. We show that FcγR generated nAPCs released several-fold higher levels of immunogenic cytokines such as IL-1β, IL-15, and IL-23 than cytokine induced nAPCs or IC-treated cDCs. IL-1β enhances the function and memory response of T cells[92], IL-15 promotes the survival and function of CD8$^+$ T cells and lowers the threshold for TCR activation[93] and IL-23 promotes memory T cell proliferation and the differentiation of Th17 cells[94]. We show that FcγR-induced nAPCs accumulate in the spleen and can migrate from peripheral tissue (foot pad) to draining lymph nodes, where they interact with T cells. The latter demonstrates that like cDC, nAPCs are migratory and can carry antigen taken up in tissues to secondary lymphoid organs.

Despite significant advances in cancer immunotherapy[95], clinical benefits remain limited to a minority of patients and there still is an unmet need to achieve durable T cell immunity to many tumors. Targeted DC-based cancer vaccines have been studied extensively but challenges remain, including the low abundance of cross-presenting cDCs and the need for TLR agonists to induce immunogenicity[11]. Conversion of the abundant neutrophils to nAPCs in vivo with neutrophil-selective anti-FcγRIIIB-antigen conjugates could overcome these limitations by generating a large pool of immunogenic APCs without any need for TLR agonists. In vivo, nAPCs generated by engaging FcγRs are highly immunogenic. They stimulate a CD4$^+$ T cell-dependent delayed-type hypersensitivity reaction, promote robust CD4$^+$ and CD8$^+$ T cell proliferation, and elicit CD8$^+$ T cell-dependent target cell killing and anti-tumor immunity. Both pro- and anti-tumorigenic functions for neutrophils in cancer have been described[96]; FcγR agonists could potentially shift this balance by promoting the generation of highly immunogenic nAPCs. Our strategy enables the initiation of immune responses to antigens linked as cargo to FcγRIIIB-specific antibody as well as tumor neoantigens present in neutrophils from patients with myeloid neoplasms. The latter has the potential to provide an antigen-agonistic immunotherapeutic approach to treat these cancers of the innate immune system[73,97,98], which are currently largely incurable, particularly in older adults.

In summary, our studies reveal that FcγRs, including the poorly understood human FcγRIIIB receptor, transduce signals following their endocytosis that induces PU.1-dependent transcriptional changes that convert neutrophils into immunogenic nAPCs with functions previously ascribed exclusively to classical DCs. We propose that converting the abundantly present neutrophils into potent, cross-presenting, immunostimulatory nAPCs by in vivo targeting of FcγRIIIB, which is a receptor selectively expressed on neutrophils, could serve as an immunotherapeutic approach that overcomes the obstacles encountered with cDC therapy.

## Methods

**Mice.** All mice were on a C57Bl/6 background: Wild-type mice, FcRγ$^{-/-}$ [99], γ$^{-/-}$ mice expressing human FcγRs (FcγRIIA(2A)/γ$^{-/-}$, FcγRIIIB(3B)/γ$^{-/-}$ and FcγRIIA+IIIB (2A3B)/γ$^{-/-}$)[31], β2m$^{-/-}$ [100] (The Jackson Laboratory #002087), MyD88/TRIF$^{-/-}$ [38], B6.SJL-Ptprc$^a$ Pepc$^b$/BoyJ (The Jackson Laboratory #002014), CD11c-YFP (B6.Cg-Tg (Itgax-Venus)1Mnz/J, The Jackson Laboratory #008829)[101], Granulocyte-specific PU.1 conditional knock-out mice (MRP8-Cre-IRES/GFP-Spi1$^{[f/f\ 86]}$), OT-I expressing the transgenic T cell receptor recognizing Ovalbumin residues 257–264 (SIINFEKL) in the context of H2K$^b$ (The Jackson Laboratory #003831), OT-I/ βactin-GFP (obtained by crossing OT-I mice with β actin-GFP mice), OT II, transgenic mice expressing T cell receptor recognizing Ovalbumin 323–339 peptide in the context of H2K$^a$ and are β-actin RFP positive (The Jackson Laboratory, # 005884). Animals were maintained in a specific pathogen-free facility. All in vivo experiments were conducted with age and sex matched animals. The Brigham and Women's Hospital Animal Care and Use Committee approved all procedures in this study.

**Human volunteers, SLE patients and sera and myeloid neoplasia patients.** For studies with SLE patients, peripheral blood samples were obtained from healthy controls and patients with SLE who fulfilled the 1997 ACR classification criteria. The study was approved by the IRB of the Instituto Nacional de Ciencias Médicas y Nutrición Salvador Zubirán (IRE-2297) and all participants signed informed consent forms. Healthy volunteers without family history of autoimmune diseases were recruited as controls. The clinical and demographic characteristics of the study participants are summarized in Supplementary Table 1. Human SLE serum samples were obtained from the Lupus Clinic at the Beth Israel Deaconess Medical Center and Instituto Nacional de Ciencias Medicas y de la Nutricion Salvador Zubiran. For studies on normal human neutrophils, and collection of normal sera, blood samples were obtained from healthy volunteers consented to Brigham and Women's Hospital IRB protocol P001694/PHS. Blood samples from patients with myeloid neoplasia were obtained from patients consented to Dana-Farber Cancer Institute IRB protocol 01-206. The clinical and demographic characteristics of the study participants are summarized in Supplementary Table 2. All volunteer subjects gave written informed consent.

**Reagents.** Murine GM-CSF (#315-03), human GM-CSF (#300-03), murine G-CSF (#AF-250-05), were from Peprotech. BSA (#A-7970), Ova (#A5503) anti-BSA rabbit antibody (#B7276) and anti-Ova rabbit antibody (#C6534) were from Sigma Aldrich. FITC-Ova (#O23020) was from Thermofisher). NIP-Ova (11 NIPs per Ova) was from Biosearch Technologies and anti-NIP human IgG1 (chimeric antibody with lambda light mouse chains and heavy light chains) was a gift from Richard Blumberg (Brigham and Women's Hosp, Boston). H-2 K$^b$ Ova Tetramer (Ova257–264) were from the NIH Tetramer Core Facility. Anti-FcγRIIIB (3G8) (Biolegend) was conjugated to FITC-Ova (#O23020, Thermofisher) as a custom order (Biolegend). Accutase cell detachment solution (#07920) was from STEMCELL Technologies. FACS antibodies used, including catalog numbers, clones, and dilutions, are in Supplementary Tables 3–6. For depletion experiments, anti-mouse Ly6G (1A8, #BP0075-1), anti-mouse CD4 (GK1.5, #BE0003), anti-mouse CD8 (2.43, #BE0061) and rat IgG2a isotype control (2A3, #BP0089) and rat IgG2b Isotype control (LTF-2, #BE0090) were obtained from BioXCell. Ova peptide SIINFEKL, (OTI peptide), and Ova peptide (323–339) (OTII peptide) were provided by the Partners peptide/protein core facility (Boston, MA), Diphtheria toxin (Cat #D0564) and tetanus toxoid (#582231) were from Calbiochem.

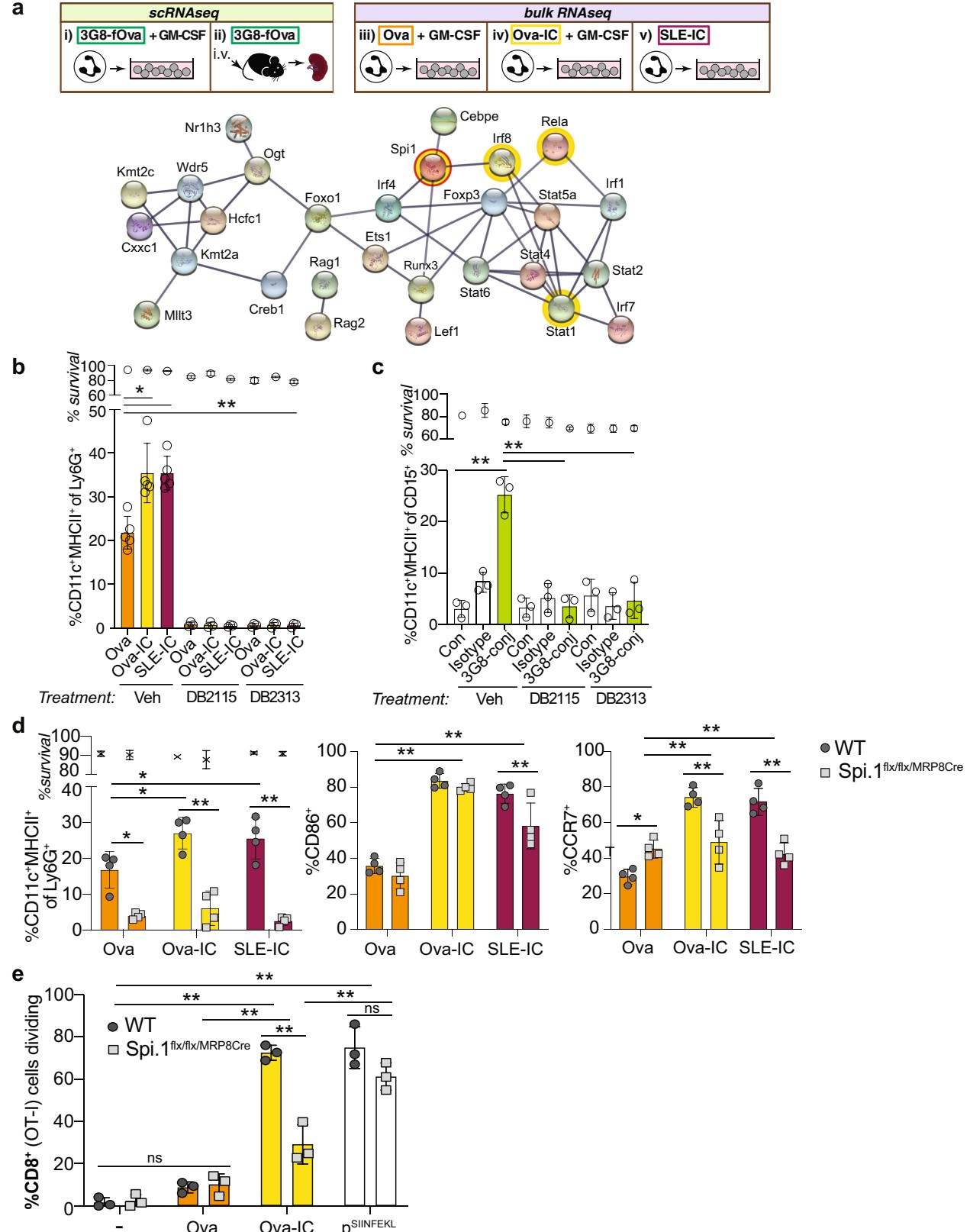

**Generation of model immune complexes.** To generate immune complexes, solutions containing 2 mg/ml of anti-Ova, anti-NIP-Ova, or anti-BSA antibodies were added to 0.2 mg/ml of Ova, FITC-Ova, NIP-Ova, or BSA (mixture 1) or 0.1 mg/ml of Ova, FITC-Ova, NIP-Ova, or BSA in PBS (mixture 2) at a 1:1 ratio and incubated overnight at 37 °C to generate IC. The excess antibody in mixtures 1 and 2 results in the appearance of a precipitate (insoluble immune complexes). Equal

volumes of mixtures 1 and 2 are combined to generate a solution of ICs. 10 μl of this solution is added per 1 ml of neutrophil suspensions.

**Sera, IgG isolation, and generation of immune complexes with RNP.** Peripheral blood was drawn into BD Vacutainer™ Venous Blood Collection Tubes SST. Tubes were centrifuged at 2500 × g for 30 min. Serum was aliquoted and frozen at −80 °C.

**Fig. 9 PU.1 is a master regulator of neutrophil to nAPC conversion. a** Schematic of indicated samples analyzed by scRNA seq (i-ii) and bulk RNA seq (iii-v): Isolated neutrophils from 2A3B/γ−/− mice treated with 3G8-fOva and cultured with GM-CSF (i), splenocytes harvested from 3G8-fOva-injected 2A3B/γ−/− mice (ii), isolated wild-type neutrophils treated with Ova (iii) or Ova-IC (iv) and cultured with GM-CSF, or SLE-IC and cultured without GM-CSF (v). STRING network of transcriptional factors potentially regulating genes associated with trajectory gene score. Highlighted in yellow are transcription factors of interest in the *PU.1*, and *NfκB(RelA)/Stat/Irf8* networks. **b** Percent of 2A3B/γ−/− BMNs treated with Ova or Ova-IC, or SLE-IC that acquire CD11c and MHCII after 3 days when cultured with vehicle or PU.1 inhibitors (DB2115, DB2313). **c** Percent of human peripheral blood neutrophils treated with isotype or 3G8-conjugate acquiring CD11c and HLA-DR when cultured with vehicle or PU.1 inhibitors. **d** Percent of WT and Spi.1^flx/flx/MRP8cre (PU.1 deficient) neutrophils treated with Ova, Ova-IC, or SLE-IC (as in **b**) acquiring CD11c and MHCII and further positive for CD86 and CCR7. **e** Percent proliferation of Cell Trace Violet-labeled CD8+ (OT-1) T cells co-cultured with WT and Spi.1^flx/flx/MRP8Cre nAPCs generated without (−) or with Ova or Ova-IC (plus GM-CSF), or nAPCs (−) pulsed with Ova-SIINFEKL peptide (pSIINF). Data are mean ± s.e.m. One-way analysis of variance and Dunnett's multiple comparison test. *$p < 0.05$, **$p < 0.005$.

To purify IgG, serum samples were incubated with Protein G high-capacity agarose beads (Genesee Scientific, #20–538). Antibody was eluted with 0.1 M Glycine (pH 2.7) and the samples were neutralized immediately with the neutralization buffer (1 M Tris-HCl, pH 9). The samples were then dialyzed against PBS overnight and concentrated using 100 kDa centrifugal filters (Millipore, UFC910024). To generate SLE-IgG/RNP immune complexes, solutions containing 2 mg/ml of purified SLE IgG or normal human IgG were mixed with equal volumes of 0.3 mg/ml of Sm/RNP complexes (#SRC-3000, Immunovision) in PBS (without Ca, Mg) and incubated for 4 h at 37 °C. 20 μl of immune complex solutions were added to 1 ml of neutrophil suspension. RNP was used at 3 μg/ml. SLE-IgG or normal human IgG alone were used at 20 μg/ml.

**Isolation of bone marrow derived murine neutrophils**. Mice were euthanized and femurs and tibias were flushed with ice-cold RPMI 1640 using a 25-gauge needle attached to a 20 cc syringe and clumps were dispersed by gently passing the cell suspension through a 1 ml pipet tip several times. Collected cells were filtered through a 70 μm strainer into a 50 ml Falcon tube, pelleted and resuspended in PBS containing 2% FBS and 1 mM EDTA at $1 \times 10^8$ cells/ml. Neutrophils were isolated by negative selection using the EasySep Mouse Neutrophil Enrichment Kit #19762 (STEMCELL Technologies). Aliquots of the isolated cells were taken to determine neutrophil count and viability by trypan blue exclusion. Viability was also determined by FACS using the fluorescent viability dye Fixable Viability Dye eFluor 780 (ThermoFisher). Purity was assessed by flow cytometry using antibodies described in Supplementary Table 3 and 6. Mice were treated i.p. with 10 ng/kg murine G-CSF for 3 consecutive days prior to isolation of bone marrow to increase neutrophil yields in select experiments: role of model ICs, Nip-Ova/anti-Ova, BSA-anti-BSA, SLE-IC in nAPC generation; Ex vivo generated nAPCs adoptively transferred for intravital microscopy; analysis of OT-I and OT-II proliferation in BMNs from WT,2A, 3B and γ−/− mice. Isolated murine neutrophils were resuspended in RPMI medium (RPMI 1640 with L-glutamine and 25 mM HEPES, supplemented with 10% fetal calf serum and penicillin/streptomycin (50 U/ml penicillin and 50 μg/ml streptomycin).

**Treatment and in vitro culture of bone marrow derived murine neutrophils**. Neutrophils were incubated with the following for 2 h at 37 °C, 5% CO₂ in falcon tubes with gentle rocking: (1) 10 μl/ml model immune complexes, generated as described in "Generation of model immune complexes", (2) 1.5 μg/ml antigen alone at concentrations contained in their respective ICs, (3) 20 μg/ml antibody alone, (4) 20 μg/ml SLE patient IgG or normal human IgG plus RNP containing samples, (5) 30 μg/ml FITC-IgG isotype control or FcγRIIIB (3G8)-FITC-Ova. Cells were then pelleted at $300 \times g$, washed with PBS and plated at $2 \times 10^6$ cells/well in 6-well plates in 2 ml of RPMI medium. For treatment with normal or SLE sera, 50 μl/ml SLE patient sera or normal human sera were added to the media of neutrophils in culture. Only cultures of neutrophils treated with FITC-IgG isotype control or FcγRIIIB (3G8)-FITC-Ova, model antigen ICs or their constituents (conditions 1–3) were supplemented with 10 ng/ml of GM-CSF. In select experiments, PU.1 inhibitors (700 nM DB2115 or 700 nM DB2313) were added to neutrophils in culture. In some experiments neutrophils were pretreated with an inhibitor of the actin cytoskeleton (20 μM Cytochalasin D) or lipid rafts (5 mM Methyl beta Cyclodextrin-MβCD) or vehicle control (DMSO) for 30 min at 37 °C and washed thoroughly. After 3 days of culture, non-adherent and adherent cells were harvested using Accutase (STEMCELL Technologies), washed with PBS and resuspended in FACS buffer for flow cytometric analysis using antibodies described in Supplementary Tables 3 and 6. For in vivo injections, only adherent cells harvested with Accutase were used. For co-culture of CD45.1 and CD45.2 cells, bone marrow neutrophils from CD45.1 mice (B6.SJL-*Ptprc^a Pepc^b*/BoyJ) were treated with Ova-IC. Bone marrow neutrophils from CD45.2 mice (the common isoform, expressed by C57Bl/6 wild-type mice) were loaded with Ova or anti-Ova as described above. The cells were washed thoroughly and co-cultured at a ratio of 1:1 (CD45.1:CD45.2). After 3 days in culture, non-adherent and adherent cells were harvested using Accutase (STEMCELL Technologies), washed with PBS and resuspended in FACS buffer for flow cytometric analysis gated on CD45.1 and CD45.2 cells.

**Isolation and in vitro culture of murine splenic classical dendritic cells and monocyte-derived cDCs**. For splenic classical DCs, wild-type mice were implanted s.c. with a Flt3L expressing B16F10 melanoma. At day 12, spleens were harvested and cDCs were isolated using CD11c microbeads (Miltenyi Biotec) according to manufacturer's instructions. For bone marrow, monocyte-derived DCs, monocytes were isolated using the Monocyte Isolation kit (Stem Cell Technologies) and cultured for 6 days in media (RPMI1640 plus10%FBS, 20 mM Penicillin/Streptomycin) supplemented with GM-CSF (20 ng/ml) and IL-4 (20 ng/ml). Isolated cells were stained for CD11c and MHCII (Supplementary Tables 3 and 6) and analyzed by flow cytometry. Dead cells were excluded from the analysis based on fixable viability dye fluorescence.

**Analysis of phagocytosis and reactive oxygen species (ROS) generation**. Phagocytosis of $10^6$ freshly isolated neutrophils and adherent nAPCs derived from anti-Ova, Ova-IC, and SLE-IC were assessed using a *E. coli* and IgG Phagocytosis Assay kit (IgG FITC) (Cayman Chemical, Ann Arbor, MI) following manufacturer's instructions. Briefly, cells were incubated in RPMI medium with a 1:4 dilution of FITC labeled, inactivated *E. coli* or 1:200 dilution of rabbit IgG coated, FITC labeled latex beads for 2 h. Trypan blue solution was added to quench surface-bound *E. coli* or IgG complex. Cells were then gently washed with assay buffer twice and FITC uptake was measured by flow cytometry. For analysis of ROS generation, $10^6$ freshly isolated neutrophils and adherent nAPCs derived from Ova-IC and SLE-IC suspended in PBS without $Ca^{2+}/Mg^{2+}$, were incubated with serum opsonized *E. coli* (MOI:200) or 100 μg/ml zymosan followed by luminol (50 μM) in PBS with $Ca^{2+}/Mg^{2+}$. ROS was measured as production of light over time (expressed in relative light units, RLU), which was continuously monitored at indicated time points using a 6-channel bioluminat LB-953 luminometer (Berthold).

**Analysis of FcγRIIIB surface expression and 3G8 binding**. To assess surface expression of FcγRIIIB (CD16), clone REA589 was used. The cells were incubated with (1) 30 μg/ml FITC-IgG isotype control or FcγRIIIB (3G8)-FITC-Ova or unconjugated CD16 (Clone: 3G8) or (2) Ova, Ova-IC or SLE-IC. The cells were washed thrice, blocked with TruStain FcX antibody (Biolegend) for 10 min at 4 °C and stained with 1:50 dilution of CD16 (clone REA589) antibody (Miltenyi Biotec) for 30 min at 4 °C. Cells were washed thoroughly and assessed by flow cytometry. To detect the integrity of binding of 3G8 to FcγRIIIB after inhibitor treatment, neutrophils were treated with Cytochalasin D or MβCD or vehicle for 30 min at 37 °C and washed twice with PBS. Cells were then incubated with unconjugated anti-human CD16 antibody Clone 3G8 (Stem Cell technologies) at 1:50 or Isotype IgG1κ for 30 min at 4 °C. Cells were washed twice and incubated with Alexa Fluor 647 (Thermo Fisher Scientific) labeled goat anti-Mouse IgG (H + L) cross-adsorbed secondary antibody at a dilution of 1:10,000 for 30 min at 4 °C. Cells were washed and analyzed by flow cytometry.

**Isolation of peripheral blood human neutrophils**. To isolate human neutrophils, peripheral blood was drawn into tubes containing trisodium citrate, citric acid, and dextrose (Vacutainer ACD Solution A, BD). Autologous serum, used for culture of cells was obtained by drawing blood into BD Vacutainer™ Venous Blood Collection Tubes SST, followed by centrifugation at $2500 \times g$ for 30 min and serum collection. All blood donors provided written informed consent. For normal human blood samples, 10mls human blood was supplemented with GM-CSF (10 ng/ml) for 30 min at 37 °C followed by addition of 30 μg FITC-IgG isotype control or FcγRIIIB (3G8)-FITC-Ova conjugate for 2 h at the indicated concentrations. Blood was then incubated with Hetasep (STEMCELL Technologies) according to manufacturer protocols to deplete red blood cells and enrich leukocytes. Neutrophils were isolated from the leukocyte-rich plasma layer using a Easysep Neutrophil enrichment kit (STEMCELL Technologies). Neutrophil purity was evaluated using CD15, CD11b, CD66b, and lineage markers (CD3, CD19, and CD56) Supplementary Tables 4 and 6.

**Treatment and culture of peripheral blood human neutrophils.** Treatment of blood from human volunteers was as follows. 10 ml blood samples were incubated with GM-CSF for 30 min and then 30 µg 3G8-fOva or isotype control (FITC-IgGκ) for 2 h. Neutrophils were isolated and placed in culture with media plus GM-CSF. PBMCs were isolated as described below and frozen in freezing medium (40% RPMI1640 + 50% FBS + 10% DMSO) and preserved for 2 days at −80 °C. After 2 days, the cells were harvested from isotype and 3G8-conjugate treated samples and co-cultured with autologous CD3 T cells isolated from frozen PBMCs (see method below) on IFNγ ELISpot microplates (R&D Systems, #EL285) at a 1:2 ratio. To examine neutrophil to nAPC conversion, neutrophils were placed in RPMI media, which was supplemented with 10% autologous serum and penicillin/ streptomycin (50 U/ml penicillin and 50 µg/ml streptomycin) and 20 ng/ml GM-CSF. After 48 h, cells were harvested using Accutase and evaluated by flow cytometry as described in Supplementary Tables 4 and 6. In select experiments, 700 nM of PU.1 inhibitors, DB2115 or DB2313, or vehicle control (water) were added to neutrophils in culture. For studies with vaccine antigens, cultures were additionally incubated with 1 µg/ml diptheria toxoid or 1 µg/ml tetanus toxoid for co-cultures. Treatment of blood from patients with myeloid neoplasias and co-culture with T cells were as follows. 5 ml blood was treated with 30 µg 3G8-conjugate or isotype and cultured for 2 days as described above and an additional 10 µg of 3G8 conjugate/isotype was added to 1 ml culture during 2 days of culture to increase the percentage of nAPC generated. PBMCs were isolated by density gradient centrifugation with lymphoprep™ (Stemcell technologies, Canada).

**Isolation of human T cells.** PBMCs were isolated from peripheral blood using Lymphoprep (Stemcell Technologies, Vancouver, Canada) density gradient medium, aliquoted in 1 ml cryopreservation tubes at 5 million cells/ml and frozen. The tubes were thawed after 2 days to isolate CD3 T cells for co-culture studies. For isolation of CD3+ T cells, negative selection was performed using EasySep Human T cell isolation kit (#17951, Stemcell Technologies, Vancouver, Canada). Viable cells were checked for CD3 expression.

**Human IFNγ ELISPOT assay.** After 16 h of co-cultures on IFNγELISpot microplates (R&D Systems, #EL285), samples were processed according to the manufacturer's protocol and results were quantitated using an ELISpot reader (CTL ImmunoSpot® S6 Fluorescent Analyzer).

**Flow cytometry.** Flow cytometry was performed on a FACSCanto II or LSRFortessa-12-color analyzers or FACSSymphony. FCS (flow cytometry standard format) 3.0 data file was used to export data that was analyzed using FlowJo (Mac version 10.5). Compensation controls were created for each fluorochrome. BD multicolor compensation beads and cells were used to set up compensation for the individual fluorochromes. For all experiments, cells were stained with the Fixable Viability Dye eFluor 780 (ThermoFisher) to gate out dead cells. Forward and side scatter gates were used to discriminate doublets and debris (FSC-A, FSC-H, SSC-A × SSC-H). Matched isotypes were used as controls and negative gating was based on FMO (fluorescence minus one) strategy. Only viable cells were included for the studies. For surface staining, single cell suspensions in FACS buffer (PBS supplemented with 2% FCS and 2 mM EDTA) were incubated with mouse BD Fc block or human TruStain FcX for 20 min at 4 °C. Samples were incubated with the indicated fluorochrome-conjugated antibodies for 30 min at 4 °C, washed with PBS and fixed with 1% paraformaldehyde.

**Flow cytometric analysis of murine bone marrow-derived neutrophils and nAPCs.** For phenotypic and functional analysis of bone marrow-derived neutrophils and nAPC, antibodies for the following cell surface markers were used: CD11c, MHC-II, Ly6G, CD80, CD86, and CCR7 (see Supplementary Tables 3 and 6 for details). Within the viable population of lineage negative (CD3, NK1.1, and CD19), CD11c+, and MHC-II+ events were gated and Ly6G expression was analyzed. The CD11c+MHC-II+ and Ly6G+ population was further analyzed for expression of co-stimulatory molecules CD80, CD86, and the migratory marker CCR7. The CD11c gates were set based on FMO controls and CD80, CD86, and CCR7 were selected based on isotype and FMO controls. Displayed numbers are CD11c+, MHCII+, and Ly6G+ events expressed as %. The subset DC markers XCR1, CD103, and CD8a were analyzed on the CD11c+, MHC-II+, and Ly6G+ population (gating strategy in Supplemental S1). For uptake analysis, FITC and Ly6G gates were set based on isotype control. Anti-Ly6G was coupled with APC fluorochrome. For OT-I and OT-II experiments, anti-CD3, -CD4, and -CD8 antibodies were used. Singlets of CD3+, CD4+, or CD8+ viable cells were gated for further analysis. Histograms are from one representative sample. Display populations of CellTrace Violet+ events (dark gray) were set based on isotype controls. Light gray histograms indicate day 0 staining of CellTrace Violet+ events. For intracellular staining, fixed cells were stained for surface markers and permeabilized with BD Perm/Wash Buffer (BD Biosciences). Cells were then stained with anti-Foxp3, -T-bet, and Ki-67 antibodies Cells were washed again with permeabilization buffer and analyzed by flow cytometry.

**Flow cytometric analysis of human blood neutrophils and dendritic cells.** To evaluate cell markers in freshly isolated human neutrophils, CD15, CD11b, CD66b,

and lineage markers (CD3, CD19, and CD56) were used. Dead cells were excluded using fixable viability dye and viable cells were gated for lineage negative markers. Cells were checked for CD15 and CD66b expression. For analysis of nAPC markers after culture of human neutorphils, viable cells were analyzed for expression of CD11c and HLA-DR. Double positive cells were further checked to evaluate co-stimulatory markers CD80, CD86, and migratory marker CCR7 (see Supplementary Tables 4 and 6). To evaluate cell markers in circulating leukocytes in whole blood of SLE patients and paired normal controls, 100 µL of blood was incubated with antibodies for the following surface markers: CD10, CD15, CD11c, MHCII, CD80, CCR7 (see Supplementary Tables 5 and 6). Within the viable population of lineage negative (CD3, CD19, and CD56), CD11c+ and MHCII+ events were gated and CD10 and CD15 expression was analyzed. The CD11c+MHCII+, CD10+, and CD15+ population was further analyzed for CD80 and CCR7 markers. The DC subset markers Clec9a was also evaluated in the CD11c+MHCII+CD10+CD15+ population. Red blood cells were lysed using lysis buffer (BD Pharm Lyse) and the samples were analyzed by flow cytometry in an LSR Fortessa cytometer (BD bioscience).

**Live cell imaging of mouse blood neutrophils.** Live cell imaging was performed using a CellAsic Onix2 microfluidics cell trap plate (M04T-01- EMD Millipore) (100 um² traps) mounted on a widefield inverted Nikon Ti microscope enclosed in a custom-made environmental chamber heated to 37 °C. Pre-mixed 5% $CO_2$ was perfused into the cells through the gas line in the CellAsic Onix2 microfluidics device. The microscope was equipped with a Plan Apo 100×/1.45 objective, a Nikon linear encoded motorized stage, an Andor Zyla 4.2 plus sCMOS monochrome camera and a Lumencor Spectra X light engine. The acquisition software controlling the microscope was Nikon NIS Elementens AR 4.30. Phototoxicity was minimized by reducing light intensity using an ND8. The signal from YFP-reporter and SYTO™ deep red-stained nucleus were collected using Chroma ET filter cubes 49003 and 49006, respectively. Neutrophils isolated from the peripheral blood of naive CD11c-YFP mice were stained with SYTO™ Deep Red nucleic acid stain (Invitrogen). SYTO stained neutrophils were introduced into the microfluidics cell trap plate and stimulated with 20ug/ml SLE-IC for 1.5 h. The cells were washed with RPMI1640 + FCS for 30 min and perfused with RPMI1640 + FCS + SYTO™ deep red nucleic acid stain at 3 kPa flow rate for the duration of the experiment. Transmitted light images were collected every 2 min in brightfield to track individual cells whereas fluorescence images in YFP-reporter and SYTO™ channels were collected every 30 min to minimize phototoxicity. Focus was maintained by the software-built steps in range autofocusing routine using transmitted light. A z-stack was collected at each timepoint.

**Confocal imaging of mouse blood neutrophils.** Isolated peripheral blood neutrophils stained with SYTO™ Deep Red nucleic acid stain (Invitrogen) were stimulated with 20 µg/ml SLE-IC for 1.5 h and placed in culture. Z-stack images were taken at Day 0 and at Day 2 were collected using a DMI6000 (Leica) with a 63X NA1.3 Plan-Apo glycerol immersion objective, CSUX Yokogawa spinning disc (Andor), Borealis illumination system (Andor), and Zyla Plus camera (Andor), controlled by MetaMorph 7.8 (Molecular Devices).

**Cytospins of human neutrophils.** Cytospins were prepared of purified human neutrophils (day 0) and 24 h after treatment with 3G8-fOva or isotype control and cultured in GM-CSF. Cytospins were stained with Wright-Giemsa.

**Analysis of FITC-Ova, FITC-Ova/anti-Ova, and anti-FcγRIII (3G8)-FITC-Ova uptake in vitro.** For murine neutrophil uptake experiments, cells from the indicated genotypes were cultured with FITC-Ova or FITC-Ova-anti-Ova model immune complexes generated as detailed above. At the indicated times, cells were collected, stained with anti-Ly6G, quenched with trypan blue, and subjected to FACs analysis to detect Ly6G+FITC+ cells. For human neutrophil uptake measurements, blood was incubated at 37 °C with FITC-IgG isotype control or anti-FcγRIIIB (3G8)-FITC-Ova antibody conjugate (8.5–12.5 µg/ml) with gentle agitation on a rocking table for the indicated times. Samples were then treated with ammonium chloride solution (STEMCELL Technologies #07800) for 15 min on ice to lyse red blood cells, washed with PBS, and subjected to FACS analysis with the indicated antibodies: anti-CD10, -CD15 -CD16 (REA-589) -CD16(3G8) and -CD32 (IV.3) (Supplementary Table 4).

**Adoptive transfer of CD45.1 nAPC into CD45.2 recipients and spleen harvest.** Neutrophils from CD45.1 mice were incubated with Ova-ICs to generate nAPCs as described above. After 3 days, adherent cells were collected, For ex vivo studies, half the nAPC sample was labeled with FITC and injected i.v. into CD45.2 mice. After 3 days, the spleen was harvested and subjected to a FACs sort for CD11c+MHCII+ cells and additionally sorted on FITC + CD45.1 positive cells (presumptive nAPCs) or CD45.2 cells (presumptive cDCs). The recovered nAPCs and cDCs were incubated with Cell trace violet labeled CD4+ (OTII) or CD8+ (OTI) T cells at a 1:2 ratio. After 3 days, samples were subjected to FACs analysis to examine T cell proliferation as described. For in vitro studies, the other half of the nAPC sample and CD11c positive cells isolated from spleen using CD11c+ beads (Miltenyi Biotec) were plated with Cell trace violet labeled CD4+ (OTII) or CD8+ (OTI)

T cells at a 1:2 ratio. After 3 days, samples were subjected to FACs analysis to examine T cell proliferation.

**Adoptive transfer of neutrophils into MHCII deficient mice and immunization with 3G8-fOva**. Neutrophils were freshly isolated from bone marrow of naive FcγRIIA + FcγRIIIB/γ$^{-/-}$ mice and $1 \times 10^7$ cells were injected i.v. into MHCII deficient recipient mice at day −3 and day −2. Five hours post injection of neutrophils, 3G8-fOva conjugate or 3G8 (control) was given i.v. At day 0, mice were injected i.v. with CellTrace Violet-labeled CD4$^+$ (OTII) T cells. After 3 days, lymph nodes (brachial, inguinal, renal, mediastinal, popliteal, axillary), spleen and liver were harvested and T cell proliferation was evaluated.

**Analysis of nAPC accumulation in organs after 3G8 i.v. injection**. 3G8-fOva conjugate or 3G8 (unconjugated) was given with or without 5 μg GM-CSF intravenously. After 3 days blood, lymph nodes (brachial, inguinal, renal, mediastinal, popliteal, axillary), spleen, lungs and liver were harvested and percent nAPCs expressing CD11c$^+$MHCII$^+$CD80$^+$CD86$^+$CCR7$^+$Ly6G$^+$ was evaluated.

**Blood and spleen analysis for detection of Ag uptake and neutrophil-DC markers**. γ$^{-/-}$, 3Bγ$^{-/-}$, and 2A3Bγ$^{-/-}$ mice were injected retro-orbitally with 30 μg of 3G8 (anti-FcγRIIIB)-FITC-Ova. Blood samples were taken at the indicated times and subjected to red blood cell lysis using ACK lysis buffer, washed with PBS and analyzed by FACS with antibodies for the following markers: Ly6G, CD11c, CD11b, MHCII, CD80, and CCR7 (nAPC analysis) or Ly6C, CD115, CD16(REA-589) and CD32 (uptake). Spleens were harvested after 3 days and processed as described in the organ harvest section.

**CD8$^+$ (OTI) and CD4$^+$ (OTII) T cell proliferation assays in vitro**. T-cells from the Ovalbumin specific TCR transgenic OT-I and OT-II mice were isolated from spleen and lymph nodes by positive selection. Cells were enriched with microbeads conjugated to anti-mouse CD8a (Ly-2) (Miltenyi) for enrichment of CD8$^+$ T-cells or anti-mouse CD4 (L3T4) (Miltenyi) for CD4$^+$ T-cell enrichment. Final preparations contained 85–90% CD8$^+$ or CD4$^+$ T-cells. $0.5 \times 10^6$ bone marrow neutrophils from the indicated genotypes were loaded with Ova, Ova-IC or vehicle control as described earlier, seeded in 96-well microplates in RPMI medium (10% FCS) supplemented with GM-CSF and cultured for 3 days to generate nAPC. At day 3, adherent cells were counted and replated at equal numbers for all samples. As a positive control, nAPC (with vehicle control, i.e. no Ova) were pulsed with 1 μg/ml of Ova$_{257–264}$ 24 h prior to the addition of T cells. To assess T cell proliferation, OT-I and OT-II T cells were resuspended in PBS ($10^7$ cells/ml) containing 0.1% bovine serum albumin (Sigma) and incubated with CellTrace Violet (Invitrogen) at 5 μM for 20 min at 37 °C. The reaction was stopped with 10% FCS and the cells were washed twice with cold PBS and resuspended in RPMI supplemented with 10% FCS, L-glutamine, and pen/strep. nAPC were co-cultured with $2 \times 10^5$ CellTrace Violet-labeled OT-I or OT-II cells. After 3 days for OT-II and 5 days for OT-I, cells were washed with PBS and analyzed by flow cytometry for CellTrace Violet dye dilution.

**CD8$^+$ (OTI) and CD4$^+$ (OTII) T cell proliferation assays in vivo**. Enriched OT-I and OT-II cells were labeled with CellTrace Violet dye as described for in vitro assays. Cells were resuspended in PBS and injected i.v. ($3 \times 10^6$ cells/mouse) into recipient mice (day 0) that had received an i.v. injection of 30 μg of FcγRIIIB-FITC-Ova conjugate at day −10. On day 4, mice were euthanized and cells from the spleen were harvested and stained for flow cytometry analysis. For MHCII deficient mice, the mice were injected retro-orbitally with 30 μg of 3G8 (anti-FcγRIIIB)-FITC-Ova conjugate or 3G8 antibody on day-3 and day-2. The mice also received 2A3Bγ$^{-/-}$ neutrophils on day -3 and day -2. CellTrace Violet dye labeled OT-II T cells were injected i.v on day 0. On day 3, mice were euthanized and cells from lymph nodes, spleen and liver were harvested and analyzed using flow cytometry.

**In vivo CTL assay**. Recipient mice were injected with 30 μg of FcγRIIIB-FITC-Ova at day −7. At day 0, target splenocytes were generated by harvesting spleens from WT naive mice and incubating half the cells with ("pulsed") half without ("unpulsed") 1 μg/ml Ova$_{257–264}$ (SIINFEKL Analytical Biotechnology Services, Boston) for 1 h at 37 °C in RPMI supplemented with 10% FCS. The cells were washed and resuspended in PBS and the pulsed and unpulsed cells were labeled with 6 μM CellTrace Violet (Thermofisher, C34557) (pulsed) or 1 μM CellTrace Violet (unpulsed) respectively for 20 min at 37 °C. The cells were washed and resuspended in PBS, and $1 \times 10^6$ pulsed and $1 \times 10^6$ unpulsed cells were co-injected i.v. into the indicated recipient mice. 16 h later, spleens from recipient mice were harvested and assessed for loss of pulsed targets by flow cytometry. The following equations were used to determine the actual value of cell specific lysis:[102] %Specific cytotoxicity $= [1 - \{(x/y)/(a/b)\}] \times 100$ where $x =$ CellTraceViolet$^{int/high}$ event number in T Ag-primed mouse, $y =$ CellTraceViolet$^{low}$ event number in T Ag-primed mouse, $a =$ CellTraceViolet$^{int/high}$ event number in naive mouse, and $b =$ CellTraceViolet$^{low}$ event number in naive mouse. For neutrophil depletion, mice were intraperitoneally administered 350 μg anti-Ly6G antibody (clone 1A8, BioXCell, NH) or isotype control at day −2 and −1 before the addition of

FcγRIIIB-FITC-Ova every 2 days thereafter for the remainder of the experiment. Depletion of Ly6G+ cells was confirmed and monitored in peripheral blood by flow cytometry.

**Adoptive transfer of nAPC or immunization with anti-FcγRIIIB-FITC-Ova prior to B16F10-Ova challenge**. Neutrophils isolated from WT or MyD88$^{-/-}$TRIF$^{-/-}$ mice were incubated with PBS (vehicle), Ova or Ova-anti-Ova ICs for 2 h, washed and cultured as described under "Neutrophil treatments". After 3 days, non-adherent cells were gently removed and the remaining adherent cells were detached with Accutase. $1 \times 10^6$ adherent cells were injected intravenously into Isoflurane-anesthetized mice 7 days prior to subcutaneous inoculation of B16F10-Ova cells. For immunization studies, 100 μl of PBS solution containing the 3G8 (anti-FcγRIIIB)-FITC-Ova (30 μg) or FITC-Ova at an equal molar ratio of Ova present in the conjugate (12 μg), was injected i.v. into mice 6 h after an intraperitoneal injection of 5 μg of GM-CSF. Four additional daily injections were given. Ten days after 3G8-fOva conjugate or FITC-Ova injection, mice were challenged with a subcutaneous injection of $2 \times 10^5$ B16F10-Ova cells as detailed in the B16F10-Ova tumor challenge, organ harvest and flow cytometric analysis section. For analysis of tissue accumulation of adoptively transferred nAPCs, neutrophils were incubated with Ova-ICs to generate nAPCs as described. After 3 days, adherent cells were harvested and labeled with Cell trace fluorescent dye (Invitrogen) according to the manufacturer's instructions. $5 \times 10^6$ nAPCs were injected i.v. into mice with tumor implanted 6 days prior.

**B16F10-Ova tumor challenge and T cell depleting antibodies**. B16F10 cells expressing soluble Ova in vitro in DMEM/high glucose supplemented with 10% fetal calf serum and maintained at sub-confluent density. Mice were anesthetized with an intraperitoneal injection of a ketamine/xylazine cocktail (90 mg/kg ketamine, 10 mg/kg xylazine), shaved and injected in the flank subcutaneously with $2 \times 10^5$ tumor cells in 100 μl HBSS. Tumors were measured with a caliper every 1–2 days once palpable in any one group (long diameter and short diameter) and tumor volume was calculated using an ellipsoid formula (1/2XDxd[2]) where "D" and "d" are the longer and shorter diameter respectively. For T cell depletion, beginning 1 day prior to injection of B16F10 implantation, mice were treated with intraperitoneal injection of an initial dose of 200 μg/mouse of anti-CD4 (clone GK1.5, BioXCell) or anti-CD8 (clone 2.43, BioXCell) antibodies in PBS, followed by dosing with 100 μg/mouse every 7 days throughout the course of the experiment.

**Organ harvest, T cell depletion, and flow cytometric analysis**. Organs and tumors were surgically removed and processed within 30 min of removal. Briefly, on ice, all excess fat was removed. The organs were gently dissociated in FACS buffer (PBS supplemented with 2% FCS and 2 mM EDTA) by shearing the tissue on a 70 μm nylon cell strainer (FisherBrand) using a 3 ml syringe plunger. Tumors were minced/digested in Collagenase type I and dissociated using gentle MACS Dissociator 1X and lymphocytes were purified on a Percoll (GE Healthcare) gradient and resuspended in FACS buffer. Cells from spleen and liver were subjected to red blood cell lysis using ACK lysis buffer solution (Lonza Cat 10-548E) for 2 min at room temperature, washed once with PBS and resuspended in FACS buffer. Blood was collected from the retroorbital plexus of isoflurane-anesthetized mice using micro-hematocrit capillary tubes into EDTA tubes (final 5 mM). Samples were subjected to red blood cell lysis using ACK lysis buffer for 10 min at room temperature, washed with PBS and resuspended in FACS buffer. For flow cytometric analysis, cells were analyzed for expression of CD4, CD8, CD44, CD62L, MHCI$^{tet}$ (H-2 K$^b$ Ova Tetramer (Ova$^{257–264}$)), CD69, CD25, PD-1, Foxp3, Ki-67, and T-bet.

**Delayed-type hypersensitivity (DTH) responses**. Ova-IC or Ova fed neutrophils were cultured in the presence of GM-CSF. 3 days later, non-adherent cells were discarded, adherent cells were harvested using Accutase and $1 \times 10^6$ nAPC were injected i.v. into wild type recipient mice. 7 days after immunization, 50 μg of soluble Ova in 100 μl PBS was injected into the right, rear foot pad and an equal volume of PBS was injected in the left, rear foot pad as a control. Paw swelling was measured on day 0 before Ova injection and 24 h after Ova or PBS challenge using digital calipers. The magnitude of the DTH response was calculated as the thickness after and before Ova or PBS challenge. For histological analyses of the DTH response, paws were collected, fixed in formalin solution, neutral buffered, 10% (Sigma, HT-50-14-120 ml) and processed for H&E staining using standard histopathological methods. The Inflammatory score was determined by an observer blinded to the identity of the samples.

**Cytokine detection and analysis**. Supernatants from nAPCs generated in vitro from Ova-IC or Ova was collected and analyzed for cytokine and chemokine levels using the pro-inflammatory focused 32-plex array (Eve Technologies, Calgary, AB).

**Intravital microscopy of popliteal lymph node**. Approximately $5 \times 10^5$ nAPCs generated with Ova-IC as described for the DTH model were resuspended in 25 μl of PBS and injected into the right hind footpad of wild-type C57Bl/6 mice at day 0.

$(10–10.5) \times 10^6$ isolated OT-1 β-actin-GFP cells in 200 µl PBS were injected i.v. 30 min before nAPC injection. At day 3 after nAPC injection, mice were anesthetized with a mixture of ketamine (50 mg kg$^{-1}$) and xylazine (10 mg kg$^{-1}$) injected intraperitoneally. The jugular vein of the anesthetized mice was cannulated for intravenous delivery of Qtracker 655 Vascular dye and additional anesthetics as needed. The right popliteal lymph node was prepared microsurgically and positioned on a custom-made microscope stage for intravital microscopy. During microsurgery, attention was given to spare blood vessels and afferent lymph vessels. The exposed lymph node was submerged in normal saline, covered in a glass coverslip with a thermocouple placed next to the lymph node to monitor local temperature which was maintained at 37 °C. Two-photon intravital imaging was performed with an upright microscope (Prairie Technologies) and a water-immersion ×20 objective (0.95 numerical aperture). A MaiTai Ti:sapphire laser (Spectra-Physics) was tuned to 870 nm and 900 nm for two-photon excitation and second-harmonic generation. Images of xy sections (512 × 512 pixels) were acquired every 1.5–2 s. for 5 min (analysis) or 10 min (videos) with electronic zoom varying from ×1 to ×3. Emitted light and second-harmonic signals were directed through 450/80-nm, 525/50-nm, and 630/120-nm bandpass filters and detected with non-descanned detectors. Files were saved as multiple TIF images and imported into FIJI[103] for analysis or Imaris software (Bitplane) for export as videos.

**Preparation of samples for bulk RNAseq**. Bone marrow neutrophils were isolated from wild-type mice by negative selection using the EasySep Mouse Neutrophil Enrichment Kit and incubated with Ova, Ova-IC or SLE-IC for 2 h as described. Cells were washed and cultured with GM-CSF for the Ova and Ova-IC samples and without GM-CSF for the SLE-IC treated cells. After 3 days, adherent cells were harvested and total RNA was extracted using RNeasy Mini kit (Qiagen, Madison WI) following the manufacturer's protocol. Day 0 untreated neutrophils were harvested immediately after isolation. RNA quantity and purity for all samples were measured using a NanoDrop™ Spectrophotometer (Thermo Fisher Scientific Inc). The KAPA mRNA HyperPrep Kit (Sigma-Aldrich) was used to generate libraries that were then sequenced using the Illumina *NextSeq 500* System. Single-end reads were mapped to Gencode version vM20 mouse assembly using ARMOR to generate transcripts per million (TPM) matrices through SALMON version 0.10.2. For quality control, we first defined common genes as genes detected with at least one mapped fragment in 90% of the samples. We then computed the percentage of those common genes (22,443 genes) detected in each sample. All samples had >97% of common genes detected, passing quality control.

**Preparation of samples for single-cell RNAseq and sorting**. Three recipient 2A3Bγ$^{-/-}$ mice were injected with 30 µg of 3G8-conjugate at day −3. On day 0, spleens from 3G8-FITC(f)-Ova injected mice and WT control mice were harvested separately. Before sorting, splenic cells from 2A3Bγ$^{-/-}$ mice were stained with 0.7 µg of FACS antibodies, for Lin− markers (CD3, B220, CD19, NK1.1, Thy1.1, Thy1.2, and Ter119), CD11c, MHCII, CD11b, and Ly6G and the fixable viability stain 780 (See supplementary Tables 4 and 6). The singlet population was enriched by gating on live cells. This population was further gated for Lin-negative cells to remove T cells, B cells, and NK cells. The remaining Lin-negative population was gated for FITC (i.e., 3G8-fOva) positive and Ly6G$^+$ CD11b$^+$ positive cells. CD11c and MHCII positive cells were sorted using a BD FACSAriaIII cell sorter as sort 1 and the double negative population was sort 2. The sorted cells were individually hashed with different hashing antibodies. WT control spleens were gated for Lin-negative cells and Ly6G positive cells were sorted. The Ly6G$^+$ cells were hashed separately. Neutrophils from 2A3Bγ−/− mice were harvested on day 0 and in vitro Ova-IC generated nAPCs were harvested on days 1, 2, and 3 and hashed separately. Both in-vivo and in-vitro hashed samples were mixed in equal proportion and subjected to 10× Chromium controller. Sample populations were stained with anti-mouse Total-Seq A Hashtag antibodies #1–#12. Approximately 3000 of each population were then pooled together and a total of 36,000 cells were loaded onto the 10× Chromium controller.

**Bulk RNAseq gene expression quantification and quality control**. The Kapa mRNA library preparation (KAPA mRNA HyperPrep Kits) was used to generate libraries that were then sequenced on the NextSeq 500. Single-end reads were mapped to Ensembl version GRCm36.p6 transcripts using kallisto[104,105] to generate transcripts per million (TPM) matrices. For quality control, we first defined common genes as genes detected with at least one mapped fragment in 90% of the samples. We then computed the percentage of those common genes (17,927 genes) detected in each sample. All samples had >98% of common genes detected, passing quality control.

**Single-cell RNAseq gene expression quantification and quality control**. Libraries were generated according to the 10× Genomics Chromium Single Cell 3′ protocol and sequenced using a NextSeq 500 sequencer (Illumina). Cellranger v3.0.2 was used to process the raw BCL files. First, raw BCL files were demultiplexed using cellranger mkfastq to generate FASTQ files with default parameters. These FASTQ files were aligned to the mm10 genome and gene/antibody reads were quantified simultaneously using cellranger count. The final raw matrix consisted of barcode specific UMI counts for each gene/hashtag. Starting with raw

unfiltered cellranger output matrices, barcodes with greater than 500 unique genes detected and less than 5% mitochondrial UMI passed RNA-seq quality control. To remove barcodes with ambiguous hashing identity, two hashing UMI quality metrics were calculated: (i) the proportion of hashing UMI for the most abundant hashing antibody and (ii) the ratio of the number of UMI for the second most abundant hashing antibody to the number of UMI for the most abundant hashing antibody. Cells with >80% most abundant hashing antibody UMI and a ratio of second to most abundant hashing antibody of less than 5% passed hashing quality control and were available for downstream analysis.

**Single-cell clustering, visualization, and cell type identification**. To process the single cell data, raw gene × cell matrices were first normalized, then subsetted on the top 3581 variable genes based on standardized log dispersion. The gene expression matrix was scaled and centered by mean 0 variance 1 on the variable genes and then cosine distance was calculated. PCA analysis was performed on these variable genes and 50 PCs were calculated Visual inspection of an elbow plot revealed that the top 15 PCs captured most of the deviation in the data. These PCs were harmonized using Harmony to remove batch effects based on sample differences. These harmonized PCs were used to cluster the cells based on Louvain[106] graph-based clustering. UMAP[107,108] dimensionality reduction was performed for dimensionality reduction. Finally, we annotated the clusters based on a panel of curated cell-type specific marker genes combined with differential expression analysis and hashtag abundances. These cell type identifications were maintained throughout all further analyses.

**Bulk PCA and visualization**. PCA analysis was performed on scaled expression of the genes whose mean expression was in the top 90th percentile of all genes. 10 PCs were calculated and visualized.

**Trajectory analysis and trajectory score**. To characterize cell differentiation trajectory, we use Monocle 3[109–111]. A principal graph was learned on the UMAP projection of the in vitro cells using the learn graph () function. These cells were then ordered using the order cells () function in order to generate a pseudo-time axis. The following linear model was then fit using poisson regression to counts ($X$) of gene $i$ in cell $j$ found along the discovered trajectory defined by pseudo-time $P$ with $nUMI$ as an offset term:

$$X_{i,j} \sim \beta_0 + \beta_1 P_j + offset(nUMI_j)$$

The set of genes with $p < 0.00005$ (1246 genes) defined the set of genes correlated with pseudo-time and were used to calculate trajectory gene scores. These gene scores were calculated by multiplying the fitted beta from the above model for gene $i$ with the expression of that gene in cell $j$. The sum of these values across all genes within each cell generated a score applied to the in vivo data along with the bulk data. A full list of these genes can be found in Supplementary Data 3.

**Maturation score**. To characterize maturation levels of Nt.1 and Nt.5 subsets, a maturation score was calculated by adding the expression of a set of 49 maturation-associated genes provided in a recent paper exploring neutrophil heterogeneity[63]. We subsetted the normalized expression matrix on these genes and then scaled the resulting matrix. These scaled expression values were multiplied by an indicator variable (1 or −1) representing positive or negative logFC in the respective cluster and summed across all cells in each cluster. The full list of genes may be found in Supplementary Data 3.

**Pseudo-time gene expression correlation**. To show changes in gene expression of certain markers and how these changes correlated with pseudo-time, we binned the in vitro cells that fit the trajectory identified above into 25 bins. Within each bin, the mean and standard deviation of the normalized expression of each marker across all cells was determined and plotted vs. pseudo-time. The dots representing the mean normalized expression value within the bin were colored according to the most abundant cell type in that bin.

**GO ontology pathway analysis**. Pathway analysis was performed using the KEGG (Kyoto Encyclopedia of Genes and Genomes) database[112–114]. Top 200 genes by auc were used as input into the find enriched pathways function from the KEGG Profile package[115]. Ribosomal genes were excluded from the top 200 genes and disease related pathways were removed from the final list of pathways to consider.

**BART TF analysis and STRING database visualization**. The genes that correlated with pseudo-time were entered into the BART web-browser, which uses the MARGE-cistrome function to predict potential regulators of the input gene list. These regulators were taken and visualized using the STRING database. In the network produced by the STRING database, the edges were limited to those with high confidence scores (>0.700). The colors of nodes were later changed to highlight the regulatory modules identified in the networks.

**Statistics**. Statistical analyses were performed using Graphpad prism 8 (La Jolla, CA), STATA 13 (StataCorp. 2013. College Station, TX) and JMP10 software (SAS Institute, Inc, USA). All the data included in the studies are expressed as mean ± SEM. For group analysis, one-way ANOVA with Dunnett's multiple comparison was used. If only two groups were compared, a two-tailed students t-test with Bonferroni correction was used. The statistical significance of the differences for SLE patient and SLEDAI scores was determined by a nonparametric test using Mann–Whitney analysis. Correlation was analyzed using Spearman test. $*P < 0.05$ and $**P < 0.005$ was considered significant. For statistical analysis of tumor growth, the change in volume over time from day 7 to day 14 was estimated in each animal controlling for group and the repeated measures of the same animal through use of a mixed-effects regression model with a random subject effect. The model considered separate intercepts and slopes for animals in each of the four groups. An additional model considered possible non-linear relationships of volume with time in each animal through inclusion of a quadratic effect in time. For estimation, this effect was centered at the mean time of 12.5 days. Models were fitted using maximum likelihood, and likelihood ratio tests were used to test whether slopes differed across groups and whether the inclusion of quadratic terms improved model fit. Models were fitted using the Mixed routine in STATA version 13. A model adding the quadratic term was not significantly better than the model assuming linearity over time with separate slopes per group.

**Reporting summary**. Further information on research design is available in the Nature Research Reporting Summary linked to this article.

## Data availability

All data generated or analyzed during this study are included in this published article (and its Supplementary information files). Sequence data that support the findings of this study have been deposited in the Gene Expression Omnibus (GEO) database with primary accession code GSE173569. All reasonable requests for unique materials will be provided under an institutional Materials Transfer Agreement. Source data are provided with this paper.

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

## Acknowledgements

We thank the MicRoN (Microscopy Resources on the North Quad) Core, Harvard Medical School, in particular Dr. Paula Montero Llopis, for experimental design, assistance, and support of this work and Saki Mihori, Dept. of Pathology, Brigham and Women's Hospital for technical assistance with live cell image analysis. Drs. Martin LaFleur and Arlene Sharpe (Evergrande Center for Immunological Diseases, Harvard Medical School, Boston, MA) provided the B16F10-Ova cells, Dr. Richard S. Blumberg (Dept of Medicine, Brigham and Women's Hospital) provided anti-NIP antibody, Drs. Gregory Poon and David Boykin (Georgia State University, Atlanta, GA) provided the PU.1 inhibitors. Dr. Vasileious Kyttaris (Division of Rheumatology, Beth Israel Deaconess, Boston, MA) provided SLE sera. Dr. Robert Anthony (Department of Medicine, Massachusetts General Hospital, Boston, MA) isolated IgG from normal and SLE sera. Dr. Akira Shizuo (Department of Host Defense, Research Institute for Microbial Diseases, Osaka University, Suita Osaka, Japan) provided MyD88/TRIF$^{-/-}$ mice. This work was supported by National Institutes of Health R01HL065095 (T.M.), R01AI152522 (T.M.), 1R01AR063759 (S.R.), U01HG009379 (S.R.), the Lupus Distinguished Innovator Award (Lupus Research Alliance) (T.M.), CONACyT 261539 (F.R.S.), the Harvard Ludwig Institute (T.M. and J.C.A.) and internal funds provided by Mass General Brigham, Innovation Sponsored Programs (T.M.).

## Author contributions

V.M. and X.C. performed and analyzed all the studies in the paper except the RNA analysis and studies in lupus patient samples. J.M. and F.Z. completed the RNA analysis, F. Rosetti (Mexico) and I.M.S. provided SLE sera and completed the studies in blood samples from lupus patient and normal volunteers, K.O. contributed to the work in FcγR humanized mice and P.X.L. conducted the intravital microscopy studies. F. Rosenbauer (Germany) provided the mice with conditional PU.1 deletion in neutrophils, J.C.A. conceptualized the research and R.M.S. provided the myeloid neoplasia samples, U.V.A. provided valuable advice and resources, A.H.L. provided valuable advice and conducted the histological analysis of samples and S.R. supervised the RNA analysis. T.N.M. conceptualized the research and designed and supervised the experiments, V.M. designed the experiments, T.M., J.C.A., V.M., J.M., and X.C. wrote the manuscript. All authors edited the manuscript.

## Competing interests

The authors declare no competing interests.
