## [Peer Review File · Nature Communications]

REVIEWER COMMENTS

Reviewer #1 (Fc receptor signaling, cancer therapy.) (Remarks to the Author):

The authors show that treatment of mouse neutrophils with immune complexes in the presence of GM-CSF results in induction of a DC phenotype exemplified by expression of CD11c, MHC-II, CD80, CD86 and CCR7. This effect was dependent on the γ chain subunit, which associates with activatory Fc γ Rs, but not MyD88 or TRIF and thus independent of TLR signalling. Human FCGR1IA or FCGR1IIB were able to substitute for gamma chain deficiency. These CD11c and MHCII⁺ neutrophils were more frequent in SLE patients than in normal donors and furthermore there was a correlation of DC marker expression with increased disease severity scores. The authors provide evidence that immune complexed ovalbumin is internalised by neutrophils and presented to naïve TCR transgenic CD4 and CD8 T cells in vitro and in vivo and this approach could also mobilise endogenous T cells to elicit immunity against OVA-expressing B16 melanoma cells. The authors also provide a strategy using an anti-FCGR1IIB in a humanised mouse model, to deliver antigen to neutrophils and demonstrate that this can lead to successful priming of T cell responses. Interestingly, the authors provide preliminary data to suggest that neutrophils from 1 primary myelofibrosis and 2 AML patients can be stimulated with anti-FCGR1IIB conjugate to activate autologous T cells and attribute this to enhanced presentation of endogenous antigens. Using scRNAseq the authors show that the induced neutrophil cell population is heterogeneous comprising a subset that shares a number of genes with DCs. Given that GM-CSF and IgG immune complexes contribute to conversion of neutrophils to APCs, the authors tried to identify shared driver mechanisms. The authors identified PU.1 from RNAseq data as a potential transcription factor involved in the transition of neutrophils into DC-like APCs. Pharmacological PU.1 inhibitors as well as conditional deletion of PU.1 in neutrophils resulted in a reduction in the generation of DC-like neutrophils in vitro without affecting the induction of CD86. The data in the first part of the manuscript describing conversion of neutrophils into professional APCs including the ability to cross-present Ag to CD8 T cells is largely known, although previous studies focussed on the role of cytokines. In contrast, identification of PU.1 as a regulator of the generation of this population is novel. It remains unclear if these neutrophils express the full array of costimulatory ligands which are associated with DC activation. Similarly their ability to secrete IL-12 and induce Th1 differentiation is not known. A side by side comparison with resting and activated DCs using a titration of antigenic peptides/proteins would have been useful as this could provide a direct comparison of the T cell stimulatory effects on a cell per cell basis. Another drawback is the lack of a model to demonstrate the role of the conversion process during a natural immune response, for example during infection, or if this is a phenomenon that is primarily associated with immune pathology? Finally, the reader is left wondering about the mechanism through which FCGR1IIB, a GPI-anchored receptor, mediates these effects, as well as the role of GM-CSF in this context. The authors suggest that their anti-FCGR1IIB-FITC-OVA conjugate comprises 1 IgG:2 FITC-OVA molecules, but characterisation of the protein is not shown. If this claim is correct then this conjugate is likely to be different from an immune complex and the implication, therefore, is that the bivalent engagement of FCGR1IIB by the conjugate is sufficient for the conversion of neutrophils into APCs. Another pertinent question is whether induction of DC-like features by anti-FCGR1IIB can be dissociated from antigen delivery.

Reviewer #2 (Fc receptor/TLR signaling, Ag presentation.) (Remarks to the Author):

The manuscript by Mysore et al. describes a differentiation of neutrophil to acquire APC-like feature when stimulated with IC in the presence of GM-CSF. The authors termed this antigen presenting neutrophil as nAPC and showed that either human Fc γ R1IA or Fc γ R1IIB can mediate the differentiation, which is accompanied by morphological changes in the nucleus. They found SLE patients have more

CD11c, MHCII expressing neutrophils and their expression of CD80 and CCR7 correlated with SLE clinical severity score. They differentiated human neutrophils to nAPC using anti-CD16 antibody 3G8-conjugate and GM-CSF. Functionally, these nAPC are capable to present antigen and stimulate CD4 and CD8 proliferation. Similarly, human differentiated neutrophils stimulated autologous T cells. When injected to mice expressing B16F10 melanoma, Ova-IC nAPC protected mice from the tumor. The CD11c MHCII expressing neutrophils were observed in vivo in FcγRIIIB/IIA transgenic animals after injection of 3G8-conjugate. FcγRIIA/IIIB transgenic animals immunized with 3G8-Ova together with GM-CSF boost protected animal against B16F10-tumor. Finally, the authors used both single cell and bulk RNAseq of nAPC to gain insight of signaling pathways important for the neutrophil differentiation and found PU.1 was necessary.

Overall, the manuscript provides a systematic approach to show the importance of nAPC as antigen presenting cells to stimulate adaptive immunity and potentially important for anti-tumor immunity. The strength of the manuscript is that it provided animal data and potential clinical relevance of differentiated neutrophils. The manuscript is data heavy.

However, the concept of the neutrophil differentiation into antigen presenting cells in the presence of GM-CSF has been shown by Matsushima et al Blood 2013. The difference here is that the current manuscript emphasizes the contribution of IC in the neutrophil conversion. The conclusion of IC induces neutrophil into APC appears over stated as Matsushima et al showed GM-CSF alone can drive neutrophil into APC. It is possible IC treatment resulted in cytokine production that further enhanced GM-CSF in neutrophil differentiation. That is IC amplified the conversion and resulting in more nAPCs, but GM-CSF induced phenotypic conversion of neutrophils. The current results are consistent with GM-CSF being the primary differentiation cytokine, but augmented with IC treatment. For example, the authors showed Ova-IC + 3 days GM-CSF resulted in higher expression of CD11c than anti-Ova or Ova treatment. This could very well be a kinetic effect as Matsushima et al showed 6 days GM-CSF culture resulted in higher CD11c expression than 4 days. The result of 3G8-fOva alone had no effect on the growth of B16F10-Ova melanoma, but 3G8-fOva+GM-CSF reduced tumor growth is also consistent with GM-CSF is the main neutrophil differentiation factor (Figure 3g). If IC is more important than GM-CSF in suppressing tumor growth, why not to use 3G8-Ova to boost instead of GM-CSF. It is also not clear if all control animals received GM-CSF injection. Similarly, Ova-nAPC showed partial tumor protection compared to Ova-IC nAPC when transferred to B16F10 tumor expressing mice (Figure 2f).

Similar morphological changes from polymorphonuclear to mononuclear neutrophils (shown in Fig 1) was shown by Matsushima et al (Blood 2013) in GM-CSF treated cells. Likewise, the ability of differentiated neutrophils to present antigen to CD4 and CD8 T cells and stimulate their proliferation were also reported by Matsushima et al.

Regarding to the role of FcγRIIIB in neutrophil function, it has been shown that FcγRIIIB plays important role in neutrophil response to IC, particularly in IC-mediated neutrophil secretion of reactive oxidants (Fossati et al. Arthritis & Rheumatism 2002).

The other question is whether FcγRIIIB is uniquely important in this process compared to FcγRI and FcγRIIA. The statements on FcγRIIA in Fig 1g and Suppl Fig 1q are misleading as in these cases 3G8 is not the right antibody. Authors stated: "neutrophils from $\gamma^-/-$ mice expressing human FcγRIIIB acquired DC markers when treated with 3G8-conjugate and GM-CSF (Figure 1g), changes that were largely absent in $\gamma^-/-$ neutrophils lacking FcγRIIIB or expressing only FcγRIIA". Here, FcγRIIA specific antibody (IV.3) not 3G8 should be used to address if FcγRIIA also contributed to CD11c expression. Similarly, the statement of faster internalization of IIIB than IIA in 3G8-conjugate treated human neutrophils (Suppl. fig.1q) is misleading as IIA internalization should be measured in the

presence of IV.3. In fact, the 3G8 treatment on human neutrophils should be also done using IV.3. As shown in Figure 1c, similar results (Fig 1g-h) should be obtained through FcgRIIA. Namely, if FcgRIIIB is not unique in driving neutrophil differentiation to nAPC, the significance of FcgRIIIB in therapy is less clear.

As the concept of neutrophils differentiation into APC to stimulate T cells has been demonstrated before, the manuscript needs to provide novel mechanistic understanding of the nAPC generation instead of simply broaden the GM-CSF treatment to IC+GMCSF. It is not clear if FcRg chain is involved as FcgRIIIB does not interact with signaling molecules. If FcRg is not necessary, how did Ova-IC signal the neutrophils ?

In several places, the authors made statement of TLR is not required for Ova-IC mediated nAPC generation and their stimulation of T cells is out of context. TLR is understandably less relevant in IC stimulation in general as IC signaling involves src,erk pathway not MyD88 . To address if TLR signaling can lead to neutrophil differentiation to APC-like. The authors need to use LPS stimulation not Ova-IC. 3G8 (without Ova conjugation) injection to FcgRIIIB transgenic animals is a better control than 3G8-Ova injection to g-/- animal as g-/- animals failed to generate nAPC due to lack of FcgRIIIB rather than lack of IC stimulation. The 3G8 control should be included in Fig 3a. Similarly, unconjugated 3G8 should be a control for Fig 1g.

Reviewer #3 (Neutrophil biology, autoimmune disease.)(Remarks to the Author):

This is an interesting manuscript from the Mayadas group that proposes that a subset of neutrophils have the ability to acquire APC capabilities in response to immune complex stimulation with a putative role in anti-tumor responses. Conceptually, this is an important advance but given that various observations are contrary to the current dogma of various aspects of neutrophil biology, ruling out certain confounders or alternative explanations to the observations is particularly important. There are also several areas that need better characterization of phenotype/function of this subset.

Major comments:

1. Overall, the manuscript is structured in a manner that makes it difficult to read and follow, with a lot of jargon and abbreviations that distract from the findings. Authors are encouraged to rewrite this in a more structured/clear manner with better detail on the justifications of their experiments/models and findings.

2. I have some concerns about the general in vitro protocol used. The authors take neutrophils from peripheral blood, stimulate for 2h, then culture the cells for 3 days. Very long time, especially since in some experiments (Figs 1e and 3a) the cells adopt the nAPC phenotype quicker and neutrophils tend to die within a few hours in culture. There is insufficient detail, or this reviewer found it hard to find the details in the paper, on what percentage of these neutrophils in culture die and what percentage survive and differentiate into "APC_like cells". Careful viability assays, ideally with some sort of tracking, would be ideal to get a better estimate of the percentage of initial neutrophils being plated that : a) survive over a 3 day period; b) acquire APC-like capabilities based on cell markers; c) any evidence of proliferation in these cells? Are these more immature neutrophils that retain plasticity or is this bona fide transdifferentiation from a terminally differentiated state of the neutrophil into other cell? This is not clear, despite the multiple and complex experimental approaches. If these cells indeed acquire a survival advantage, how does this happen?

2. In the work with human cells, the authors stimulate in whole blood, then purify the neutrophils. The

rationale for this is not clear in the text. Are they attempting to look at indirect effects from other cells? In this regard, this needs to be clarified as other cells may interfere with experiment, even if it may be more reflective of the in vivo situation.

3. How physiological is the short 2h pulse? The experiments with SLE sera are likely to promote significant cell death at that time point and this is not clearly addressed.

4. The authors need to better address the finding that the nAPC phenotype correlates with a change in morphology. This is quite unclear in the text. Have these cells become bona fide DCs? Or is this change in phenotype just reflective of NET formation that has been shown by various groups to be induced by immune complexes? Some of the pictures are certainly reminiscent of this phenomenon, given the protrusions, potential evidence of extracellular debris and the /contraction/expansion of the nucleus at specific time points. If that's the case, could this be a late event when these "APC-like" markers become displayed during NET formation? Or is the change in nuclear morphology associated to a de-differentiation state? This is to me one of the most unclear aspects of the paper that needs to be better defined.

5. The scRNA is well done and interesting. One thing stands out that needs to be addressed. The authors use sorted cells, so the hAPCs have the appropriate surface markers. They conclude that hAPC2 cluster corresponds to the functional cells based on expression of APC genes and such. The PU.1 connection is also based on this cluster. But what is hAPC2? Do they express the same surface markers? Are they derived from neutrophil Nt.1? What drives the transition into hAPC2 and what would the functionality be? It appears to be of similar size to hAPC1. This needs to be better addressed and explained.

7. I was unclear on why the authors first gate on aCD11c HLA positive cells to then look at neutrophil markers in human blood. I would have expected to do it the opposite way? Gate on neutrophils first and then look at how many express APC markers to better understand the prevalence of these cells in circulation in immune-mediated diseases and healthy people. Since the paper functionality is mostly on cancer, what are the levels of these cells in various cancers and association with prognosis? While this may be beyond the scope of this paper, it would certainly enhance the implications.

8. Do these cells maintain neutrophil-related functions? This would be important to answer the question on whether they still represent bona fide neutrophils versus a transdifferentiation/dedifferentiation phenomenon. At the least the authors can look at degranulation and/or phagocytosis or ROS synthesis.

9. The Eruslanov group has published extensively on a subset of tumor-associated neutrophils with APC capabilities. This reviewer thinks it would be very important to compare this subset to the subset described in this paper. These neutrophils have been described as APC-like "hybrid neutrophils," which originate from CD11b(+)CD15(hi)CD10(-)CD16(low) immature progenitors, are able to cross-present antigens, as well as trigger and augment anti-tumor T cell responses. Are these the same cells? A thorough comparison and discussion is warranted here. Similarly, how does this APC-neutrophil fit into the LDG literature in autoimmunity and are there any overlaps here based on the gene expression data already published on these cells?

NOTE: In the manuscript, all new experimental data is in red.

REVIEWER COMMENTS

Reviewer #1 (Fc receptor signaling, cancer therapy):

The authors show that treatment of mouse neutrophils with immune complexes in the presence of GM-CSF results in induction of a DC phenotype exemplified by expression of CD11c, MHC-II, CD80, CD86 and CCR7. This effect was dependent on the γ chain subunit, which associates with activatory Fc γ Rs, but not MyD88 or TRIF and thus independent of TLR signaling. Human FCGR1IA or FCGR1IIB were able to substitute for gamma chain deficiency. These CD11c and MHCII⁺ neutrophils were more frequent in SLE patients than in normal donors and furthermore there was a correlation of DC marker expression with increased disease severity scores. The authors provide evidence that immune complexed ovalbumin is internalised by neutrophils and presented to naïve TCR transgenic CD4 and CD8 T cells *in vitro* and *in vivo* and this approach could also mobilise endogenous T cells to elicit immunity against OVA-expressing B16 melanoma cells. The authors also provide a strategy using an anti-FCGR1IIB in a humanised mouse model, to deliver antigen to neutrophils and demonstrate that this can lead to successful priming of T cell responses. Interestingly, the authors provide preliminary data to suggest that neutrophils from 1 primary myelofibrosis and 2 AML patients can be stimulated with anti-FCGR1IIB conjugate to activate autologous T cells and attribute this to enhanced presentation of endogenous antigens. Using scRNAseq the authors show that the induced neutrophil cell population is heterogeneous comprising a subset that shares a number of genes with DCs. Given that GM-CSF and IgG immune complexes contribute to conversion of neutrophils to APCs, the authors tried to identify shared driver mechanisms. The authors identified PU.1 from RNAseq data as a potential transcription factor involved in the transition of neutrophils into DC-like APCs. Pharmacological PU.1 inhibitors as well as conditional deletion of PU.1 in neutrophils resulted in a reduction in the generation of DC-like neutrophils *in vitro* without affecting the induction of CD86.

1) The data in the first part of the manuscript describing conversion of neutrophils into professional APCs including the ability to cross-present Ag to CD8 T cells is largely known, although previous studies focused on the role of cytokines. In contrast, identification of PU.1 as a regulator of the generation of this population is novel.

We would like to address the question of novelty as this is a strong point of our manuscript. Each of the points below have been made clearer in the expanded manuscript. We would like to point out that, to date, whether nAPCs made with cytokines and soluble antigen can cross-present antigen to CD8 and whether mature rather than immature neutrophils can convert to nAPCs is still debated. A more thorough discussion of what is known and not known is now included in the Introduction section.

1) Immune complex binding to Fc γ Rs represents a distinct stimulus for neutrophil to APC conversion *in vitro* (a) and *in vivo* (b):

a) *In vitro*: SLE immune complexes alone in the absence of GM-CSF drive neutrophil conversion to nAPC in a Fc γ R dependent manner (**Figure 1b,c**).

b) *In vivo*: Anti-Fc γ RIIIB-antigen conjugate (in the absence of GM-CSF) leads to the generation of nAPCs in blood, secondary lymphoid organs, liver and lung. In new data, we show that *i.v.* injected GM-CSF does not lead to appreciable conversion and when injected with the conjugate does not enhance conversion compared to conjugate alone (**Figure 4b**).

2) Mature blood neutrophils undergo conversion to nAPC *in vitro* and *in vivo* (in mice) upon Fc γ R engagement. This is unlike previous studies, which largely reported on the acquisition of DC-like features by immature, immediate precursors of end stage neutrophils (e.g. band neutrophils).

3) Functionally, nAPCs generated with antibody-antigen complexes far exceed nAPCs generated with uncomplexed antigen in their ability to activate naïve CD4 and cross-present antigen to naïve CD8 T cells, and are functionally equivalent to classical DCs (cDCs) *in vitro*.

To date, among APCs, only cDCs, robustly activate immunologically naïve CD4 and CD8 T cells (whose activation threshold significantly exceeds that for memory T cells) and cross-present antigen to CD8, a process critical for anti-tumor immunity. It is in this context that our findings are very novel as detailed below.

Note: nAPC were generated with Ova-anti-Ova (Ova-IC-nAPC), Ova alone (Ova-nAPC) or no antigen. All were generated in the presence of GM-CSF.

a) *In vitro*, Ova-IC-nAPCs stimulated the proliferation of greater than 80% of naïve CD4 T cells (**Figure 2a**). This was 5-10 fold higher than that observed for CD4 T cells incubated with Ova-nAPC, the condition used in previously published studies. Likewise, Ova-IC-nAPC stimulated the proliferation of greater than 80% of CD8 T cells (**Figure 2b**), a canonical read-out of cross-presentation. In contrast, Ova-nAPC failed to stimulate CD8 T cells: The low T cell proliferation observed with Ova-nAPC was similar to that observed with nAPCs that had no antigen (**Figure 2b**). Importantly, the level of CD4 and CD8 T cell proliferation induced by Ova-IC-nAPCs was comparable to that observed for IL-4/GM-CSF monocyte derived cDCs and splenic cDCs (**Figure 2c,d**).

b) Ova-IC-nAPCs secrete several T cell immunomodulatory cytokines (e.g. IL-1 β , IL-15, IL-23) at levels that were significantly higher than for Ova-nAPCs and, in fact, cDCs (**Figure 2e**).

4) *In vivo*, only nAPCs generated with antibody-antigen complexes activate naïve CD4 and cross-present to CD8 T cells.

First, in a side-by-side comparison, we show that adoptively transferred Ova-IC-nAPCs but not Ova-nAPCs provide CD8 dependent anti-tumor immunity and induce the generation of endogenous, antigen specific CD8 T cells (**Figure 2h**). Second, the i.v. injection of an anti-Fc γ RIIIB-Ova conjugate (3G8-fOva) in the absence of GM-CSF generates nAPCs that promote CD4 (**Figure 4c**) and CD8 (**Figure 4d**) T cell proliferation, activate endogenous naïve CD8 T cells for robust target cell killing (**Figure 4e**) and inhibit growth of Ova-expressing melanoma (**Figure 5a**) with associated generation of endogenous Ova-specific T cells (**Figure 5b**).

5) The pioneer transcription factor PU.1 promotes neutrophil to nAPC conversion.

As the reviewer acknowledges, another novel aspect of our studies is our finding that the pioneer transcription factor PU.1 is required for neutrophil to nAPC conversion (**Figure 6**).

2) It remains unclear if these neutrophils express the full array of costimulatory ligands which are associated with DC activation. Similarly their ability to secrete IL-12 and induce Th1 differentiation is not known. A side by side comparison with resting and activated DCs using a titration of antigenic peptides/proteins would have been useful as this could provide a direct comparison of the T cell stimulatory effects on a cell per cell basis.

We did not query the full array of co-stimulatory ligands on nAPCs associated with DC activation but rather focused on nAPC functionality and compared it to cDCs as detailed in point 1). We show that the levels of CD4 and CD8 T cell proliferation by Ova-IC-nAPC were comparable to splenic cDCs (Flt3DC) and IL-4/GM-CSF activated monocyte derived cDC (**Figure 2a-d**). These results infer that the IC-induced nAPCs have the full array of co-stimulatory molecules. DC derived cytokines represent an important signal for promoting T cell survival, differentiation and function. We performed ELISAs on nAPC supernatants to show that Ova-IC-nAPC released ng quantities of T cell immunomodulatory cytokines including IL-1 β , TNF and IL-15 that are 5 to 3,500 fold higher than those for nAPCs generated with Ova/GM-CSF and, in many cases, higher than those observed for cDCs (**Figure 2e**).

3) Another drawback is the lack of a model to demonstrate the role of the conversion process during a natural immune response, for example during infection, or if this is a phenomenon that is primarily associated with immune pathology?

It is not possible currently to determine if nAPCs play a significant role in natural protective immune responses since there is currently no method to selectively deplete nAPCs. However, we do show that they are part of the natural history of SLE, a prototypical IC-mediated disease, and that they can enhance responses to vaccine antigens (in vaccinated human volunteers) and tumor antigens (in B16F10 melanoma model and patients with myeloid neoplasia). These data indicate their potential relevance to immune diseases and to therapeutic vaccinations.

4) Finally, the reader is left wondering about the mechanism through which FCGR11B, a GPI-anchored receptor, mediates these effects, as well as the role of GM-CSF in this context.

To explore how the GPI-linked FcγRI1B may signal neutrophil to DC conversion, we first evaluated whether monomeric 3G8 alone is sufficient or whether 3G8-fOva, which is a heterogeneous species containing 3G8 and fOva at different ratios (unpublished data), is required. We found that only 3G8-fOva but not 3G8 treatment of FcγRI1B/γ^{-/-} neutrophils promotes receptor internalization (**Figure 3b**) and subsequent neutrophil to nAPC conversion to (**Figure 3c**) compared to isotype control. The endocytic machinery is well recognized to promote receptor signaling and gene transcription¹⁻⁴ and our previous work demonstrated that FcγRI1B and FcγRI1A internalize ICs via a lipid-raft, actin and cdc42 regulated pathway. Cytochalasin D and MβCD that disrupt the actin cytoskeleton and lipid rafts, respectively, prevented 3G8-fOva induced FcγRI1B internalization (**Figure 3b**) and reduced subsequent nAPC generation to levels seen with 3G8 or isotype control (**Figure 3c**) without effecting binding of 3G8-fOva to the surface (**Suppl. Figure 3b**). Similar results were obtained when using Ova-IC or SLE-ICs, (**Figure 3d-e**). nAPCs generated with GM-CSF alone (in presence of Ova control) were unaffected by the inhibitors (**Figure 3e**). Thus, endocytosis is an analogous proximal step driving 3G8-fOva and IC induced neutrophil to nAPC conversion. However, we cannot rule out the possibility that the pharmacological inhibitors of endocytosis effect other pathways necessary for neutrophil differentiation to nAPCs.

We also assessed the potential role of GM-CSF in FcγRI1B induced nAPC conversion *in vivo*. By 72hr after 3G8-fOva injection, nAPCs generated *in vivo* accumulated in lymph nodes, spleen, and the non-lymphoid organs, liver and lung (**Figure 4b**), similar to the reported tropisms of blood cDCs. In contrast, 3G8 alone did not induce significant accumulation of nAPCs in any organs. Notably, GM-CSF treatment of mice given 3G8-fOva did not further increase the frequency of nAPCs. However, it did result in a small but significant increase in the frequency of nAPCs in some tissues of mice given 3G8 alone (**Figure 4b**) albeit this was significantly lower compared to mice given 3G8-fOva alone.

5) The authors suggest that their anti-FCGR11B-FITC-OVA conjugate comprises 1 IgG:2 FITC-OVA molecules, but characterisation of the protein is not shown. If this claim is correct then this conjugate is likely to be different from an immune complex and the implication, therefore, is that the bivalent engagement of FCGR11B by the conjugate is sufficient for the conversion of neutrophils into APCs.

We postulate that the conjugate has its effects by bivalent engagement of FcγRI1B as anti-FcγRI1B (3G8) alone does not promote conversion (Point 4) and western blot analysis of 3G8-fOva reveals a heterogeneous mixture containing 3G8 and Ova at different ratios (unpublished data). Notably, five independent preparations of the antibody-antigen conjugate gave reproducible results. As the ICs (Ova-anti-Ova and SLE-IgG-RNP, which also remain uncharacterized) and the conjugate induced neutrophil to nAPC conversion that was associated with FcγRI1B internalization and prevented by inhibitors of the actin

cytoskeleton and lipid rafts, we can infer that Fc γ RIIIB-antigen conjugate and IC modalities trigger analogous, proximal mechanisms for neutrophil to nAPC conversion..

The degree or nature of cross-linking required to signal neutrophil to nAPC conversion is an interesting area for future investigation and would require careful fractionation of the 3G8-fOva and IC preparations followed by evaluation of the activity of each fraction.

6) Another pertinent question is whether induction of DC-like features by anti-FCGRIIIB can be dissociated from antigen delivery.

We presume that the reviewer is asking whether anti-Fc γ RIIIB alone can induce nAPC. Our study shows that anti-Fc γ RIIIB (3G8) alone is unable to induce neutrophil to nAPC conversion (see response to point 5). This suggests that bivalent receptor cross-linking, as occurs with soluble ICs and likely the anti-Fc γ RIIIB-Ova conjugate is required to signal neutrophil transition to nAPCs.

Reviewer #2 (Fc receptor/TLR signaling, Ag presentation.)

The manuscript by Mysore et al. describes a differentiation of neutrophil to acquire APC-like feature when stimulated with IC in the presence of GM-CSF. The authors termed this antigen presenting neutrophil as nAPC and showed that either human FcγRIIA or FcγRIIIB can mediate the differentiation, which is accompanied by morphological changes in the nucleus. They found SLE patients have more CD11c, MHCII expressing neutrophils and their expression of CD80 and CCR7 correlated with SLE clinical severity score. They differentiated human neutrophils to nAPC using anti-CD16 antibody 3G8-conjugate and GM-CSF. Functionally, these nAPC are capable to present antigen and stimulate CD4 and CD8 proliferation. Similarly, human differentiated neutrophils stimulated autologous T cells. When injected to mice expressing B16F10 melanoma, Ova-IC nAPC protected mice from the tumor. The CD11c MHCII expressing neutrophils were observed in vivo in FcγRIIIB/IIA transgenic animals after injection of 3G8-conjugate. FcγRIIA/IIIB transgenic animals immunized with 3G8-Ova together with GM-CSF boost protected animal against B16F10-tumor. Finally, the authors used both single cell and bulk RNAseq of nAPC to gain insight of signaling pathways important for the neutrophil differentiation and found PU.1 was necessary.

1) However, the concept of the neutrophil differentiation into antigen presenting cells in the presence of GM-CSF has been shown by Matsushima et al Blood 2013. The difference here is that the current manuscript emphasizes the contribution of IC in the neutrophil conversion. The conclusion of IC induces neutrophil into APC appears over stated as Matsushima et al showed GM-CSF alone can drive neutrophil into APC

Please see response to Reviewer 1 (point 1), who brought up similar concerns.

2) It is possible IC treatment resulted in cytokine production that further enhanced GM-CSF in neutrophil differentiation.

To address this possibility, we exploited neutrophils expressing CD45.1 or CD45.2, which can be distinguished by antibodies that recognize each isoform. We co-cultured Ova or anti-Ova pretreated CD45.2⁺ neutrophils with Ova-IC-pretreated CD45.1⁺ neutrophils in media supplemented with GM-CSF followed by the analysis of nAPC markers in each group. A significantly greater percent of CD45.1⁺ Ova-IC- versus CD45.2⁺ Ova- or CD45.2⁺ anti-Ova treated neutrophils acquired nAPC markers (**Supp. Fig 1j**). Thus, ICs enhance the frequency of neutrophil differentiation to nAPCs compared to GM-CSF alone in a cell autonomous manner.

3) That is IC amplified the conversion and resulting in more nAPCs, but GM-CSF induced phenotypic conversion of neutrophils. The current results are consistent with GM-CSF being the primary differentiation cytokine, but augmented with IC treatment.

See response to 1) and 2)

4) For example, the authors showed Ova-IC + 3 days GM-CSF resulted in higher expression of CD11c than anti-Ova or Ova treatment. This could very well be a kinetic effect as Matsushima et al showed 6 days GM-CSF culture resulted in higher CD11c expression than 4 days.

As noted in response to point 1), a major difference is that Matsushima et al primarily used immature neutrophils for the majority of their study while we used mature blood and bone marrow derived neutrophils. More importantly, functionally, we show that Ova-IC generated nAPCs were far superior to nAPCs generated with uncomplexed antigen, Ova (conditions of previous studies) in promoting CD4 T cell proliferation and was the only condition under which CD8 T cell proliferation was observed.

The result of 3G8-fOva alone had no effect on the growth of B16F10-Ova melanoma, but 3G8-fOva+GM-CSF reduced tumor growth is also consistent with GM-CSF is the main neutrophil differentiation factor

(Figure 3g). If IC is more important than GM-CSF in suppressing tumor growth, why not to use 3G8-Ova to boost instead of GM-CSF. It is also not clear if all control animals received GM-CSF injection.

In the B16F10 model, all control animals received GM-CSF, yet only FcγRIIIB expressing mice given 3G8-fOva (AAC) had reduced tumor growth. As noted in Reviewer 1 (point 1), in new studies, we show that GM-CSF does not increase 3G8-fOva induced nAPC generation *in vivo*. Also, Ova-specific cytotoxic T cells (CD62L^{low}CD44^{hi} CD8⁺/Tetramer⁺) were generated in B16F10-Ova bearing FcγR humanized mice versus wild-type mice (control) preimmunized with 3G8-fOva in the absence of GM-CSF (2A3B/γ^{-/-}, 0.75 +/- 0.1%; WT, 0.11 +/- 0.02%) albeit it was lower compared to 3G8-fOva immunized mice given GM-CSF (2A3B/γ^{-/-}: 2.12 +/- 0.50%). This indicates that GM-CSF does not directly affect the ability to generate nAPCs *per se* but we cannot rule out that it regulates the survival or function of nAPCs and/or T cells. It is possible that consecutive injections of 3G8-fOva, rather than the single injection administered may overcome the need for GM-CSF, a possibility that will be addressed in future experiments.

Similarly, Ova-nAPC showed partial tumor protection compared to Ova-IC nAPC when transferred to B16F10 tumor expressing mice (Figure 2f).

We apologize if the figure is misleading, but there is actually no statistically significant difference between Ova-nAPC treated and untreated mice (**Figure 2h**). The asterisks represent the statistical significance between Ova-IC versus Ova and untreated animals. Consistent with this finding, the number of endogenous T cells recognizing Ova (as detected with an MHC-I tetramer recognizing Ova-SIINFEKL peptide specific CD8⁺ T cells) in Ova-nAPC treated animals is 10-20 fold lower than for Ova-IC-nAPC treated mice (**Figure 2h**).

5) Similar morphological changes from polymorphonuclear to mononuclear neutrophils (shown in Fig 1) was shown by Matsushima et al (Blood 2013) in GM-CSF treated cells.

Matsushima et al 2013 and Oehler et al 1998 did report morphological changes but both used immature immediate precursors of end-stage neutrophils treated with GM-CSF over 6-9 days and relied on static images. We used mature peripheral blood neutrophils treated with SLE-ICs (without GM-CSF) and conducted live cell imaging of CD11c-YFP neutrophils, which revealed the rapid kinetics of neutrophil to nAPC conversion (within 11hrs). The dramatic conversion of a nucleus that is polymorphonuclear to one that is mononuclear in appearance suggests that the neutrophil transdifferentiates into nAPCs and demonstrates that SLE-IC alone in the absence of GM-CSF can induce neutrophil to nAPC conversion. In addition, the live cell imaging answered potential criticisms of data relying solely on static images as the latter cannot rule out that progenitors contaminating neutrophil preparations expand over a 3 day culture period or that neutrophils preferentially die leaving behind “contaminating” mononuclear cells.

Likewise, the ability of differentiated neutrophils to present antigen to CD4 and CD8 T cells and stimulate their proliferation were also reported by Matsushima et al.

Please note that Matsushima et al, stated in the manuscript that GM-CSF treated immature neutrophils only supported “modest CD8 T cell proliferation”. In our study, a head-to-head comparison of nAPC generated from mature neutrophils treated with Ova+GM-CSF or Ova-ICs+GM-CSF clearly showed that naïve CD4 T cell proliferation in the latter was markedly higher than the former (**Figure 2a**) and only Ova-ICs+GM-CSF induced nAPC cross-presented antigen to CD8 T cells (**Figure 2b**). This demonstrates that nAPCs generated with antigen-antibody complexes that engage FcγRs and not uncomplexed antigen generate fully functional nAPCs. The Ova-IC nAPCs induced CD4 and CD8 T cell proliferation that was comparable to classical, monocyte derived DCs and splenic cDCs (**Figure 2c,d**).

6) Regarding to the role of FcγRIIIB in neutrophil function, it has been shown that FcγRIIIB plays important role in neutrophil response to IC, particularly in IC-mediated neutrophil secretion of reactive oxidants

(Fossati et al. Arthritis & Rheumatism 2002). The other question is whether FcγRIIIB is uniquely important in this process compared to FcγRI and FcγRIIA.

Neutrophil functions have been assigned to this receptor albeit most reports have not been able to delineate whether the FcγRIIIB is sufficient for the queried function because of the presence of FcγRIIA. Our humanized mice, expressing FcγRIIIB and/or FcγRIIA on neutrophils in the absence of any other FcγRs allows us to assign a role specifically to FcγRIIIB in the process being studied. We do not know which of the effector functions ascribed to FcγRIIIB potentially play a role in neutrophil to nAPC conversion. In new studies, we show that FcγRIIIB endocytosis plays an important role in neutrophil to nAPC conversion (see Reviewer 1, point 4-6).

Engagement of FcγRIIIB alone is sufficient although it is not uniquely important. Like FcγRIIIB/γ^{-/-} neutrophils, FcγRIIA/γ^{-/-} neutrophils also convert to nAPCs after engaging ICs. With respect to FcγRI it is not constitutively expressed on human neutrophils and in our humanized FcγR mice, IIA and IIB transgenes are expressed in mice lacking the γ-chain and therefore lacking all murine FcγRs including FcγRI. We don't have a humanized mouse model for human FcγRI and therefore cannot evaluate the role of this receptor in neutrophil to nAPC conversion.

7) The statements on FcγRIIA in Fig 1g and Suppl Fig 1q are misleading as in these cases 3G8 is not the right antibody. Authors stated: "neutrophils from γ^{-/-} mice expressing human FcγRIIIB acquired DC markers when treated with 3G8-conjugate and GM-CSF (Figure 1g), changes that were largely absent in γγ^{-/-} neutrophils lacking FcγRIIIB or expressing only FcγRIIA". Here, FcγRIIA specific antibody (IV.3) not 3G8 should be used to address if FcγRIIA also contributed to CD11c expression.

In the experiment referred to by the reviewer was not conducted to conclude that FcγRIIA is not important. Rather, neutrophils from the FcγRIIA expressing mice were used as a negative control to show specificity of the 3G8-conjugate. We anticipate engaging FcγRIIA with an anti-IV.3-Ova conjugate would promote neutrophil to nAPC conversion given that, *in vitro*, IC treatment of FcγRIIA/γ^{-/-} neutrophils promotes neutrophil to nAPC conversion in the absence of FcγRIIIB.

One of the goals of our studies was to potentially generate a therapeutic, as appreciated by the reviewer. For this, FcγRIIIB is a superior target than FcγRIIA as the former is abundantly and selectively expressed on neutrophils while the latter is also expressed on human platelets and other leukocyte subsets including NK cells. That said, if this platform were to be advanced for therapeutic purposes, a FcγRIIIB selective antibody will need to be generated, as 3G8 recognizes not only recognizes FcγRIIIB but also FcγRIIA, which is expressed on multiple cell types.

8) Similarly, the statement of faster internalization of IIB than IIA in 3G8-conjugate treated human neutrophils (Suppl. fig.1q) is misleading as IIA internalization should be measured in the presence of IV.3. In fact, the 3G8 treatment on human neutrophils should be also done using IV.3. As shown in Figure 1c, similar results (Fig 1g-h) should be obtained through FcγRIIA. Namely, if FcγRIIIB is not unique in driving neutrophil differentiation to nAPC, the significance of FcγRIIIB in therapy is less clear.

Please see the response above.

9) As the concept of neutrophils differentiation into APC to stimulate T cells has been demonstrated before, the manuscript needs to provide novel mechanistic understanding of the nAPC generation instead of simply broaden the GM-CSF treatment to IC+GMCSF.

The novelty of our studies is detailed in Reviewer 1, Point 1.

10) It is not clear if FcRγ chain is involved as FcγRIIIB does not interact with signaling molecules. If FcRγ is not necessary, how did Ova-IC signal the neutrophils ?

In our humanized mouse model, Fc γ -chain is not involved in signaling as our mice are on a $\gamma^{-/-}$ background. Please see response to Reviewer 1 (point 4-6) for a description of new experiments that suggest a potential mechanism by which Fc γ RIIIB mediates neutrophil to nAPC conversion.

11) In several places, the authors made statement of TLR is not required for Ova-IC mediated nAPC generation and their stimulation of T cells is out of context. TLR is understandably less relevant in IC stimulation in general as IC signaling involves src,erk pathway not MyD88 . To address if TLR signaling can lead to neutrophil differentiation to APC-like. The authors need to use LPS stimulation not Ova-IC.

The rationale for evaluating MyD88/TLR deficient mice was not to interrogate the contribution of TLR signaling in neutrophil to nAPC conversion. Rather, it was to rule out the contribution of TLR engaged by potential LPS/endotoxin contamination of our reagents in Fc γ R dependent neutrophil to nAPC conversion and nAPC immunogenicity as TLR agonists are known to be required to induce cDC immunogenicity.

3G8 (without Ova conjugation) injection to Fc γ RIIIB transgenic animals is a better control than 3G8-Ova injection to $g^{-/-}$ animal as $g^{-/-}$ animals failed to generate nAPC due to lack of Fc γ RIIIB rather than lack of IC stimulation. The 3G8 control should be included in Fig 3a. Similarly, unconjugated 3G8 should be a control for Fig 1g.

We apologize that unconjugated 38G was not included in original Figure 1g (now Figure 3a, an *in vitro* assay), and Figure 3a (now Figure 4a, an *in vivo* assay). However, it is now included in other *in vitro* (**Figure 3b,c**) and *in vivo* (**Figure 4b,c**) studies and show that it does not lead to Fc γ RIIIB internalization or nAPC generation.

Reviewer #3

1. Overall, the manuscript is structured in a manner that makes it difficult to read and follow, with a lot of jargon and abbreviations that distract from the findings. Authors are encouraged to rewrite this in a more structured/clear manner with better detail on the justifications of their experiments/models and findings.

The paper has been re-written and separated into sections with headings to improve readability.

2. I have some concerns about the general *in vitro* protocol used.

a) The authors take neutrophils from peripheral blood, stimulate for 2h, then culture the cells for 3 days. Very long time, especially since in some experiments (Figs 1e and 3a) the cells adopt the nAPC phenotype quicker and neutrophils tend to die within a few hours in culture. There is insufficient detail, or this reviewer found it hard to find the details in the paper, on what percentage of these neutrophils in culture die and what percentage survive and differentiate into "APC_like cells".

We show that only 10-15% of cells die (this information is noted as % survival above the majority of the graphs). Normal neutrophils can survive *in vivo* for several days, depending on the study, (Hidalgo et al *Trends Immunol* 2019) while *in vitro*, their survival is significantly increased in the presence of serum and appropriate survival factors. The method of isolation that minimizes activation is also key to retaining viability of isolated neutrophils *in vitro*.

b) Careful viability assays, ideally with some sort of tracking, would be ideal to get a better estimate of the percentage of initial neutrophils being plated that: a) survive over a 3 day period; b) acquire APC-like capabilities based on cell markers; c) any evidence of proliferation in these cells?

Survival and percent of cells acquiring APC-like markers are presented in each graph. Also, in all our flow cytometric analyses, only viable cells are evaluated for APC-like markers. We evaluated Ki67 expression, a proliferation marker, and saw no proliferation in the cultures through day 3, which is when we harvest our cells for all our phenotypic and functional analyses. This is included in the text of the result section.

c) Are these more immature neutrophils that retain plasticity or is this bona fide transdifferentiation from a terminally differentiated state of the neutrophil into other cell? This is not clear, despite the multiple and complex experimental approaches. If these cells indeed acquire a survival advantage, how does this happen?

Our live cell imaging of peripheral blood neutrophils provides the clear evidence that segmented, mature neutrophils directly convert to mononuclear, CD11c+ nAPCs, which suggests transdifferentiation. We show that this is a remarkably rapid process that occurs within 1 hrs of stimulation with SLE-IC (with not GM-CSF present).

2. In the work with human cells, the authors stimulate in whole blood, then purify the neutrophils. The rationale for this is not clear in the text. Are they attempting to look at indirect effects from other cells? In this regard, this needs to be clarified as other cells may interfere with experiment, even if it may be more reflective of the *in vivo* situation.

Our rationale was to mimic a physiological relevant treatment strategy as we envision that the conjugate would be injected intravenously. This regimen also minimizes activation of human neutrophils, which are more susceptible to activation than murine neutrophils. That is, the addition of anti-Fc γ RIIB-antigen to purified human neutrophils, followed by wash steps, resulted in neutrophil clumping and loss of viability, something that was not observed with murine neutrophils. Treating human neutrophils in whole blood before their isolation avoided this problem.

3. How physiological is the short 2h pulse?

We believe it is physiological as antigen is normally presented for short duration and in limited concentration. The 2hr pulse was also designed to restrict antigen uptake and processing to neutrophils before their conversion to nAPC as we were interested in answering the question of whether neutrophils can process antigen and load peptide on MHC following endocytosis of soluble protein, while still being a neutrophil as this would suggest a pathway for antigen processing in neutrophils that is distinct from its degradative routes. This could not have been appreciated in previous studies; where reported, antigen was present during the entire time of neutrophil conversion to APCs.

The experiments with SLE sera are likely to promote significant cell death at that time point and this is not clearly addressed.

We do not observe significant cell death following treatment with SLE sera or SLE-ICs compared to untreated neutrophils. The percent survival is noted above each graph that evaluates nAPC conversion. The relative lack of cell death is likely because cells are only pulsed for 2hrs with SLE sera or SLE-ICs before being placed in culture.

4. The authors need to better address the finding that the nAPC phenotype correlates with a change in morphology. This is quite unclear in the text. Have these cells become bona fide DCs? Or is this change in phenotype just reflective of NET formation that has been shown by various groups to be induced by immune complexes? Some of the pictures are certainly reminiscent of this phenomenon, given the protrusions, potential evidence of extracellular debris and the /contraction/expansion of the nucleus at specific time points. If that's the case, could this be a late event when these "APC-like" markers become displayed during NET formation? Or is the change in nuclear morphology associated to a de-differentiation state? This is to me one of the most unclear aspects of the paper that needs to be better defined.

Our live cell imaging demonstrates that mature neutrophils stimulated with SLE-ICs start to acquire CD11c and, at the same time, a mononuclear morphology. The morphological change is also seen by confocal microscopy and, in human neutrophils, by immunohistochemistry. NETosis would result in SYTO DNA dye extruded from the cells. By live cell imaging, extracellular DNA was observed in less than 5% of cells that were not associated with neutrophils that had acquired CD11c, a bonafide DC marker. As with our other *in vitro* studies, the SLE-ICs are washed away after a 2hr incubation in the microfluidic device, which avoids persistent stimulation and wide-spread activation of neutrophils.

5. The scRNA is well done and interesting. One thing stands out that needs to be addressed. The authors use sorted cells, so the hAPCs have the appropriate surface markers. They conclude that hAPC2 cluster corresponds to the functional cells based on expression of APC genes and such. The PU.1 connection is also based on this cluster. But what is hAPC2? Do they express the same surface markers?

Throughout the manuscript, nAPCs are regularly identified via surface expression of CD11c and MHC II surface markers using flow cytometry. In the single-cell experiment, a sorted population of CD11c+MHCII+ cells make up the cells in both nAPC1 and nAPC2 clusters and thus correspond to the cells explored throughout Figure 6.

In the original manuscript, we explored differences between nAPC1 and nAPC2 subsets by conducting a KEGG pathway analysis on the differentially expressed genes (DEGs) between the two clusters (**Supp. Fig.6f**). These DEGs were determined using a wilcoxon rank sum test and the top 200 genes were selected by AUC. In the revised manuscript, to more robustly define the differences between nAPC1 to nAPC2, we grouped all nAPC1 and nAPC2 single cells together into pseudo-bulk profiles and conducted differential gene expression analysis between the two (**Suppl. Table S4**). KEGG pathway analysis of the top 200 DEGs by p-value for each subset of nAPCs revealed upregulation of many cytokine/chemokine-related pathways among nAPC1 cells such as TNF signaling, IL-17 signaling, and NFκB signaling. nAPC2 cells on the other hand, show upregulation of pathways potentially directly related to nAPC conversion and function such as

FcγR mediated phagocytosis, protein processing and antigen processing and presentation (**Supp. Fig.6g**). These pathways are in agreement with the original results from the KEGG pathway analysis presented in the manuscript. Yet, we posit that the pseudo-bulk analysis is more robust and resistant to incorrectly identifying differential expression among genes that are globally highly expressed. We also applied gene set enrichment analysis (GSEA) to the full list of 4374 variable genes between nAPC1 and nAPC2 cells, which further supports the KEGG pathway analysis results (**Supp. Fig.6g**). Altogether these results suggest key differences between the functionality of nAPC1 and nAPC2 subsets and predicts that nAPC2s represent more fully functional antigen presenting cells.

These analyses have been added to the supplement and the following sentences were added to the manuscript: “Direct comparison of nAPC1 and nAPC2 subsets revealed upregulation of genes within the antigen processing and presentation pathway, as well as pathways related to protein processing, and endocytosis among nAPC2 cells (**Suppl. fig.6g, Suppl Table S4**).”

Are they derived from neutrophil Nt.1? What drives the transition into hAPC2

We hypothesize that Nt.1 cells are progenitors of nAPC1 based on transcriptional similarity between these two clusters. As we note in the manuscript, the nAPC1 subsets show greater transcriptional similarity to Nt.1 and Nt.5 cells than to nAPC2 subsets (**Supp. Fig.4c**). Meanwhile, nAPC2 subsets show greater similarity to Nt.4 cells than to other Nt subsets or to nAPC1 subsets (**Supp. Fig.4c**). Transcriptional similarity alone does not prove that one subset is derived from the other. However, given the nature of our in vitro experiment, which captures cell states along a time course (day 0-day 3), these transcriptional similarities suggest that Nt.1 cells are the likely progenitors of nAPC1s, while Nt.4 cells are likely progenitors of nAPC2s as reflected in our trajectory (**Supp Fig. 4c**).

To definitively answer the reviewer’s question regarding the origin of nAPC1 cells, further experiments are required. For example, once a sorting strategy is established for separating Nt.1 and Nt.4 subsets, their ability to generate nAPC1 versus nAPC2 could be evaluated and the transcription factors driving these processes for could be identified.

.....and what would the functionality be? It appears to be of similar size to hAPC1. This needs to be better addressed and explained.

To directly explore the functional differences between nAPC1 and nAPC2, further experiments are required. First, a sorting strategy must be established by identifying a unique set of surface markers to separate the two subsets. This is challenging as it may not just be the absence or presence, but rather the surface expression level of a given receptor. Then, functional assays are needed to compare the ability of these subsets to promote T cell activation. We plan to pursue these interesting studies in the future but, in our view, they are beyond the scope of this paper, which establishes a new pathway of nAPC generation and demonstrates their biological relevance.

7. I was unclear on why the authors first gate on aCD11c HLA positive cells to then look at neutrophil markers in human blood. I would have expected to do it the opposite way? Gate on neutrophils first and then look at how many express APC markers to better understand the prevalence of these cells in circulation in immune-mediated diseases and healthy people.

We wanted to address the abundance of nAPCs within the total DC population so we gated first on DCs. In fact, we found that the frequency of CD11c⁺MHCII⁺ DCs was similar in normal and SLE patient samples, while the CD11c⁺MHCI⁺ expressing neutrophil markers differed. We also reasoned that since nAPCs are much less frequent than neutrophils in human peripheral blood, gating first on a restricted cell population would yield more accurate results.

Since the paper functionality is mostly on cancer, what are the levels of these cells in various cancers and association with prognosis? While this may be beyond the scope of this paper, it would certainly enhance the implications.

Singhal et al., 2016 (*Cancer Cell*, 30:120-135) reported neutrophils with antigen-presenting cell features in tumor tissue taken from early-stage human lung cancer patients. This study was referenced in our discussion. However, there was no correlation with prognosis. To our knowledge, there are no published studies that correlate nAPC generation with prognosis in cancer or any other diseases. Our study in SLE patient blood is novel in that we do show that the frequency of mature nAPCs in the blood of patients with SLE, an immune complex mediated disease, correlates with clinical disease scores (**Figure 1h**).

8. Do these cells maintain neutrophil-related functions? This would be important to answer the question on whether they still represent bona fide neutrophils versus a transdifferentiation/dedifferentiation phenomenon. At the least the authors can look at degranulation and/or phagocytosis or ROS synthesis.

In new experiment **Figure 1f-g**, we observed similar levels of phagocytosis of IgG opsonized beads and *E.coli* by Ova-IC-nAPCs, SLE-IC-nAPCs and freshly isolated neutrophils. Reactive oxygen species generation following *E.coli* or zymosan treatment was also similar.

9. The Eruslanov group has published extensively on a subset of tumor-associated neutrophils with APC capabilities. This reviewer thinks it would be very important to compare this subset to the subset described in this paper. These neutrophils have been described as APC-like "hybrid neutrophils," which originate from CD11b(+)CD15(hi)CD10(-)CD16(low) immature progenitors, are able to cross-present antigens, as well as trigger and augment anti-tumor T cell responses. Are these the same cells? A thorough comparison and discussion is warranted here. Similarly, how does this APC-neutrophil fit into the LDG literature in autoimmunity and are there any overlaps here based on the gene expression data already published on these cells?

Our human neutrophils are CD15 and CD10 high, express CD66b and are CD16 (FcγRIIIB) high and are isolated from peripheral blood. Therefore, these are fully differentiated cells rather than immature neutrophil precursors described by Singhal et al. Our scRNAseq analysis was performed in murine neutrophils thus precluding a direct comparison with human LDGs. In future studies, we plan to conduct scRNAseq on human nAPCs and compare their profiles with LDGs and other neutrophil subsets.

REVIEWER COMMENTS

Reviewer #2 (Remarks to the Author):

While the revised manuscript is clearly improved, there are still controls missing and overstatements of their results.

The section title: "Immune complex generated nAPCs phagocytose targets and generate reactive oxygen species that is equivalent to neutrophils." should be changed to "nAPCs retain the ability to phagocytose and to generate reactive oxygen species of neutrophils" . "Immune Complex" is not needed as GM-CSF generated nAPC phagocytose the same (Figure 1f).

The statement: "Importantly, the frequency of nAPCs expressing these markers correlated with SLE clinical severity (SLEDAI) scores(Figure 1h, Suppl. fig.1o)," is over stated. Only the expression of CD80 resulted in good correlation to SLEDAI. The CD80, CCR7 and Clec9A expressions were measured for 15 individuals but only 13 were used in SLEDAI correlation plot. Here the SLEDAI correlation plots need to include all 15 individuals, consistent with the expression plots.

The conclusion of "Thus, in a side-by-side comparison, nAPCs generated by engaging FcγRs with Ova-IC were far superior to nAPCs made with uncomplexed Ova in stimulating CD4⁺ T cells despite the presence of GM-CSF in both sets of samples and was the only condition under which crosspresentation to CD8 T cells was observed." is overstated. The proliferation assay showed that Ova-nAPC has the same capacity as Ova-IC nAPC to stimulate T cell proliferation when pulsed with Ova peptide, suggesting the two nAPC only differ in their antigens presented. Namely, Ova-IC nAPC has antigenic peptide loaded to MHC through Fc receptor-mediated internalization of Ova where Ova-nAPC does not internalize Ova and lacks antigen bound MHC. An intrinsically superior nAPC would mean a genuine difference in their marker expressions. However, the differences (Figure 1) between Ova-IC and Ova nAPC appears to be the numbers with more nAPC generated in the presence of Ova-IC than Ova. The difference between Ova-IC and Ova nAPC are primarily in antigen presentation related functions that can be explained by their internalization difference.

The proper controls in Figure 5a and 5b are missing. The control for 3G8-fOva treated B16F10-Ova tumor bearing 2A3B/γ^{-/-} mice should be 2A3B/γ^{-/-} mice without treatment, which is missing. Likewise, 3B/γ^{-/-} mice should be treated with 3G8-fOva to compare with the untreated 3B/γ^{-/-} control. The control mice for 3G8-fOva treatment needs to be in the same strain as there is a significant strain variation in tumor growth (e.g. the difference between 3B/γ^{-/-} and wt+fOva mice). If the results of 3G8-fOva treated 2A3B/γ^{-/-} mice and the WT+fOva mice are compared, the benefit of 3G8-fOva may not be significant.

Reviewer #3 (Remarks to the Author):

The authors have done a good job responding to the various queries by the reviewers. It would have been good to further strengthen the in vitro studies by doing longer term cultures (longer than 3 days) and assess the phenotype then but overall I am satisfied with this revision.

RESPONSE TO REVIEWER #2

Please note that yellow highlighted regions represent textual changes made to the manuscript. These changes are also highlighted in yellow in the manuscript.

@@@Referee #2 reports@@@

While the revised manuscript is clearly improved, there are still controls missing and overstatements of their results.

The section title: “Immune complex generated nAPCs phagocytose targets and generate reactive oxygen species that is equivalent to neutrophils.” should be changed to “nAPCs retain the ability to phagocytose and to generate reactive oxygen species of neutrophils” . “Immune Complex” is not needed as GM-CSF generated nAPC phagocytose the same (Figure 1f).

We have changed the section title as suggested.

The statement: “Importantly, the frequency of nAPCs expressing these markers correlated with SLE clinical severity (SLEDAI) scores(Figure 1h, Suppl. fig.1o),” is overstated. Only the expression of CD80 resulted in good correlation to SLEDAI.

This is not accurate. As shown in Figure 1h, the correlation for all three markers, CD80, CCR7 and Clec9A was statistically significant ($p < 0.001$, 0.013 and 0.004 respectively). Lupus is a very heterogeneous disease so showing a correlation with SLEDAI scores that is statistically significant is noteworthy. Suppl Fig 1o is the gating strategy for the FACs analysis.

The CD80, CCR7 and Clec9A expressions were measured for 15 individuals but only 13 were used in SLEDAI correlation plot. Here the SLEDAI correlation plots need to include all 15 individuals, consistent with the expression plots.

This is not accurate. All 15 points were used for the correlation plots. There is overlap in some of the dots and are therefore not readily distinguishable from each other.

The conclusion of “Thus, in a side-by-side comparison, nAPCs generated by engaging FcγRs with Ova-IC were far superior to nAPCs made with uncomplexed Ova in stimulating CD4 6 T cells despite the presence of GM-CSF in both sets of samples and was the only condition under which crosspresentation to CD8 T cells was observed.” is overstated. The proliferation assay showed that Ova-nAPC has the same capacity as Ova-IC nAPC to stimulate T cell proliferation when pulsed with Ova peptide, suggesting the two nAPC only differ in their antigens presented. Ova-peptide is a well described positive control for these assays. As written in the manuscript, “nAPCs generated in all conditions promote CD8⁺ T cell proliferation when pulsed with Ova SIINFEKL peptide (Figure 2b), which directly binds MHC I and bypasses the need for antigen processing and co-stimulatory molecules”. In other words, the SIINFEKL Ova-peptide, which is the cognate epitope for the TCR transgenic OT-1 T cells, bypasses this step.

Namely, Ova-IC nAPC has antigenic peptide loaded to MHC through Fc receptor-mediated internalization of Ova where Ova-nAPC does not internalize Ova and lacks antigen bound MHC.

As stated in the manuscript, there is only a two-fold increase in uptake of Ova-IC versus Ova, which does not explain the 5-10 fold difference in T cell proliferation between the two groups. “The significantly greater T cell stimulation by Ova-IC versus Ova nAPCs cannot be attributed to differences in antigen internalization, as neutrophil uptake of Ova-IC was only twice that of Ova (Suppl. fig.2d).”

An intrinsically superior nAPC would mean a genuine difference in their marker expressions. However, the differences (Figure 1) between Ova-IC and Ova nAPC appears to be the numbers with more nAPC generated in the presence of Ova-IC than Ova.

This is not accurate. Figure 1a, right panel, shows that CD11c+MHCII+Ly6G+ cells generated with Ova-IC (yellow bar) have a 5-7 fold increase in CD80, CD86 and CCR7 compared to CD11c+MHCII+Ly6G+ cells generated with Ova (grey) or anti-Ova (orange) alone. Therefore, in addition to the generation of additional CD11c+MHCII+ nAPCs, the Ova-IC leads to a markedly higher upregulation of T cell co-stimulatory markers and the migration marker, CCR7.

The difference between Ova-IC and Ova nAPC are primarily in antigen presentation related functions that can be explained by their internalization difference.

This is not the case for the reasons outlined in the responses above.

The proper controls in Figure 5a and 5b are missing. The control for 3G8-fOva treated B16F10-Ova tumor bearing 2A3B/ $\gamma^{-/-}$ mice should be 2A3B/ $\gamma^{-/-}$ mice without treatment, which is missing. Likewise, 3B/ $\gamma^{-/-}$ mice should be treated with 3G8-fOva to compare with the untreated 3B/ $\gamma^{-/-}$ control.

This is a valid point but it is logistically difficult to get all the mice age and gender matched for the multiple groups (total of 32 mice) evaluated in Figure 5a,b. The reason is that 2A3B/ $\gamma^{-/-}$ cannot be maintained as a homozygous colony as the 3B transgene is X-linked and thus requires mating of 2A3B/ $\gamma^{-/-}$ with 3B/ $\gamma^{-/-}$ to generate 3B/ $\gamma^{-/-}$ and 2A3B/ $\gamma^{-/-}$ mice that can be used in the same experiment. We have evaluated 2A3B/ $\gamma^{-/-}$ in a separate experiment and they exhibit tumor growth that is similar to the 3B/ $\gamma^{-/-}$ mice.

The control mice for 3G8-fOva treatment needs to be in the same strain as there is a significant strain variation in tumor growth (e.g. the difference between 3B/ $\gamma^{-/-}$ and wt+fOva mice).

There are no "strain" differences *per se* as all the mice are on a C57Bl/6 background. The difference between 3B/ $\gamma^{-/-}$ and wt+fOva mice are that the latter are immunized with Ova, which will lead to an anti-tumor response driven by endogenous antigen presenting cells, such as classical dendritic cells. Likewise, $\gamma^{-/-}$ mice given the 3G8-fOva will not make nAPCs as they lack the 3B receptor but will be immunized with the Ova that is conjugated to the 3G8. Our data show that the anti-Fc γ RIIIB(3G8)-fOva generation of nAPCs is more effective at reducing tumor growth compared to just immunizing with Ova alone (i.e. Wt+fOva and $\gamma^{-/-}$ +3G8-fOva).

This data is now better described in the manuscript as follows:

The reduction in tumor growth in 3G8-fOva immunized mice expressing Fc γ RIIIB and Fc γ RIIA was also significant compared to the more modest decrease in tumor volume observed in two sets of GM-CSF treated controls: wild-type mice given fOva, which will lead to an anti-tumor immune response driven by endogenous APCs, such as cDCs and $\gamma^{-/-}$ mice given 3G8-fOva that do not make nAPCs as they lack the Fc γ RIIIB but nonetheless, like wild-type plus fOva, will mount an immune response to fOva (contained in the 3G8-fOva conjugate) (Figure 5a).

If the results of 3G8-fOva treated 2A3B/ $\gamma^{-/-}$ mice and the WT+fOva mice are compared, the benefit of 3G8-fOva may not be significant

The 2A3B/ $\gamma^{-/-}$ and WT+fOva (as well as the $\gamma^{-/-}$ +3G8-fOva) were compared and the difference was significant ($p < 0.005$) as shown in Figure 5a with detailed statistics in Suppl Figure 5a (the legend for Suppl Figure 5a legend is pasted below). Furthermore, we analyzed the spleens for Ova-peptide specific effector CD8 T cells (CD62loCD44hi) using MHC1-tetramers and markers for CD8 effector, CD4 and Treg cells. This analysis also showed the advantage of 2A3B/ $\gamma^{-/-}$ +3G8-fOva versus WT+fOva and $\gamma^{-/-}$ +3G8-fOva.

"a) Regression analysis was used to compare slope curve co-efficient (tumor growth rate) between the different groups. A Prism regression analysis was used and tested for whether slopes and intercepts differ. Overall comparison was done using a one-way ANOVA and Dunnett's multiple comparison. Trendlines for tumor volume of all groups, created in Excel, are shown."

RESPONSE TO REVIEWER #3

@@@Referee #3 reports@@@

The authors have done a good job responding to the various queries by the reviewers. It would have been good to further strengthen the in vitro studies by doing longer term cultures (longer than 3 days) and assess the phenotype then but overall I am satisfied with this revision.

We thank the reviewer for appreciating our responses to the queries by the reviewers.